# A machine learning-based perspective on deep convective clouds and their organisation in 3D. Part II: Spatial-temporal patterns of convective organisation

Sarah Brüning[1] and Holger Tost[1]

[1]Institute for Physics of the Atmosphere, Johannes Gutenberg University Mainz, Johann-Joachim-Becher-Weg 21, Mainz, 55128, Rhineland-Palatinate, Germany

**Correspondence:** Sarah Brüning (sbruenin@uni-mainz.de)

**Abstract.** This series of papers explores spatio-temporal patterns of convective cloud occurrence and organisation. We use a machine learning-based method to extrapolate a contiguous 3D cloud field of 2D satellite data. In Part 2, we focus on convective organisation in tropical West Africa between March and August 2019, examining how it relates to the 3D properties of convective clouds and their core structures. We quantify organisation using three indices (SCAI, COP, ROME) to capture different aspects of spatial cloud clustering. Our findings highlight how cloud properties may interact with organisation. Hence, strong organisation may occur with larger cloud areas, lower cloud tops and core heights, and shorter lifespans compared to the average convective system. In contrast, weak organisation may be associated with smaller clouds, fewer cores, but similarly shorter lifespans. We find an increasing frequency of convective organisation in the northern hemisphere during boreal summer months, likely linked to the northward migration of the Intertropical Convergence Zone (ITCZ). From March to May, patches of strong convective organisation emerge along the African coastlines and over the southern Atlantic Ocean. Between June and August, hotspots shift inland, particularly across the Sahel and wider West African plains. Notably, oceanic regions show slightly stronger organisation overall. However, overlapping regions of strong and weak organisation may complicate the interpretation of regional statistics. While the machine learning-based 3D perspective helps bridge observational gaps in the representation of cloud structures, the inherent complexity and variability of convective organisation highlight the need for continued investigation.

## 1 Introduction

Atmospheric convection plays an essential role in the climate system through its contribution to weather and climate variability (Brune et al., 2020). In the tropics, we observe convective clouds forming as spatially connected structures of extensive size (Houze, 1977). These mesoscale convective systems (MCSs) are one of the main drivers for the transport of heat and moisture through the atmosphere. Furthermore, they affect the hydrological and radiative variability on Earth (Hartmann et al., 1984).

The spatial clustering of convective systems - also known as convective organisation - may promote the occurrence of severe weather events such as hail and floods (Becker et al., 2021). However, a robust assessment of the connection between convective organisation and extreme weather, in particular in a future climate under global warming, expresses the need for further research.

Although the term convective organisation has become increasingly popular in climate research, it is often used vaguely. Mapes and Neale (2011) broadly summarise organisation as "non-randomness in meteorological fields in convecting regions". This definition induces a clustering of deep convective cells which is ubiquitous in the atmosphere, particularly in the tropics. However, the underlying mechanisms remain insufficiently understood (Muller and Bony, 2015). While convective organisation is difficult to quantify in observational data, idealised model configured in radiative-convective equilibrium (RCE) could demonstrate a large-scale clustering of convective clouds which is known as self-aggregation of convection (e.g, Nakajima and Matsuno (1988); Held et al. (1993); Wing et al. (2017)). Following Bretherton et al. (2005), it occurs on a timescale between days and weeks and describes the transition of an approximately random distribution of convective cells into convecting and non-convecting regions that grow upscale over time. Convective aggregation is driven by either internal dynamics, like cold pools and radiative feedback, or external forces, such as the land-sea-breeze (e.g., Haerter et al. (2019), Coppin and Bony (2015), Dauhut et al. (2016)). Self-aggregation increases with the size and proximity of convective clouds and affects the radiative feedback, large-scale circulation, and moisture distribution in the vicinity of a cloud cluster (Hartmann et al., 1984). For instance, an idealised model setup shows that an aggregated state consists of a single moist region surrounded by dry regions. Moreover, the feedback between convection, surface fluxes, and radiation further drives aggregation (Tobin et al., 2012). Research shows that self-aggregation may increase with a warming climate (Wing et al., 2020). However, there remain uncertainties connected to a large model spread (Bläckberg and Singh, 2022).

Despite these insights derived from models, identifying and quantifying convective organisation in observational data persists to be a challenge. This may be due a high variability in the quality and quantity of observations. In response, previous studies have developed various metrics aiming towards a deeper understanding about the underlying physical mechanisms. The indices analyse the spatial distribution of the clouds within a defined area to estimate the strength of convective organisation (Pscheidt et al., 2019). For instance, they help differentiate a regularly distributed, randomly distributed, or organised cloud field by using morphological attributes such as the number of clouds, their nearest-neighbour distances, size, shape, pattern, and timing (Pendergrass, 2020; Retsch et al., 2020).

Providing timely forecasts and a robust climate risk assessment requires even more a correct representation of convective organisation. While satellite observations has shown that organisation within the tropics may increase overall with extreme precipitation (Semie and Bony, 2020), we have limited knowledge about convective organisation on a regional level. In this study, we aim to provide a deeper understanding of the relationship between cloud properties and convective organisation on this regional scale, comparable to the work of Bao et al. (2024). The area of interest (AOI) covers West Africa and the tropical Atlantic Ocean between $30°$ N–$30°$ S and $30°$ W–$30°$ E and lies within the Inter-Tropical Convergence Zone (ITCZ). Here, the environmental conditions favour the development of deep convective clouds, which are often associated to heavy rain (Takahashi et al., 2023). A heterogeneous landmass distribution in the northern and southern hemispheres and land-ocean

contrasts may affect the development of convection (Zipser et al., 2006). Over the tropical Atlantic Ocean, a weaker large-scale forcing may induce lower cloud tops and less intense rain rates than over continental Africa (Futyan and Genio, 2007). The rainfall variability between the individual regions of the AOI substantially depends on the moisture availability and thermal gradients (Berthou et al., 2019). Overall, the West African monsoon (WAM) dominates the West African climate. A strong temperature gradient between the warm Sahara and the colder waters of the Gulf of Guinea drives the WAM (Fontaine and Philippon, 2000). Stronger convection generally leads to an increase in heavy rain, a larger detrainment, and a slightly smaller thick anvil emissivity. For instance, Stubenrauch et al. (2023) found a distinct annual cycle of convective organisation connected to seasonal shifts of the convective cloud properties.

In Part 1 of this sequence of papers, we derived contiguous trajectories of convective clouds and their deep convective core regions (hereafter: cores) in 15-minute intervals for a six-month period between March to August 2019 (Brüning and Tost, 2025). In this study, we examined cloud and core properties of tropical convection and the life-cycle of single-core and multi-core convective clouds. In this paper, we aim to complement the findings by an in-depth analysis of spatio-temporal patterns of convective organisation. Moreover, we aim to investigate the connection between convective organisation and cloud properties within the AOI. For this purpose, we quantify convective organisation at each point in time by employing three organisation indices. The goal is to derive spatial patterns of organisation and compare their spatio-temporal variability (Biagioli and Tompkins, 2023). Our study employs convective cloud trajectories derived from a 4D time series of contiguous 3D radar reflectivities, which we predict from a machine learning (ML)-based extrapolation of 2D satellite data (Brüning et al., 2024). We employ an object-based algorithm to detect and track convective clouds in the predicted radar reflectivity field. This perspective allows a simultaneous coverage of the horizontal (cloud and core area) and vertical (cloud and core height) properties in the AOI, including remote oceanic regions over the Atlantic Ocean. Our aim is to showcase how convective organisation is distributed in the AOI within the six-month period. Furthermore, we strive to quantify how differences in the cloud and core properties are connected to a weak or strong convective organisation.

We have divided this article into five further sections. In Sect. 2, we describe the dataset used in this study. Section 3 presents an overview of metrics employed to quantify convective organisation. Section 4 contains an overview of the results comprising the spatio-temporal variability of organisation indices and cloud properties. Section 5 relates our key findings to other studies. Moreover, we discuss some limitations we encountered and evaluate the role of the ITCZ and other environmental drivers for the development of tropical organisation. Finally, Sect. 6 contains a summary and the main conclusions.

## 2 Data

To quantify convective organisation over tropical West Africa, we use a ML-based 3D cloud mask build on the 3D cloud reconstruction method described in Brüning et al. (2024) and the convective cloud detection framework by Brüning and Tost (2025). The following section outlines the workflow for producing the 3D radar reflectivity dataset, detecting convective clouds and cores, and extracting cloud properties (Figure 1).

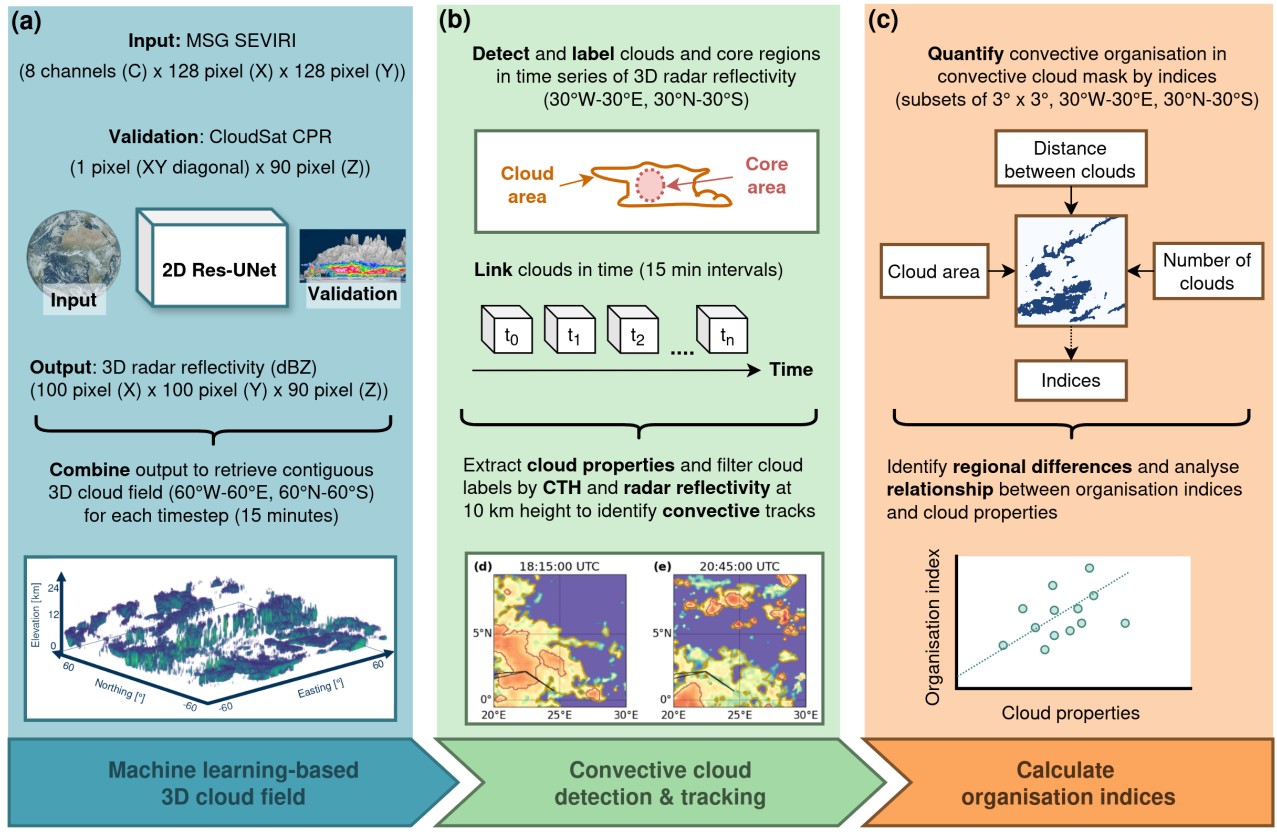

**Figure 1.** Overview of the workflow for this study. In **(a)**, we show how to derive a contiguous 3D cloud field from 2D data by a machine learning-based extrapolation (Brüning et al., 2024). For this purpose, 2D satellite imagery from the MSG SEVIRI sensor is fed into a 2D Res-UNet and trained to predict a 3D image of radar reflectivities validated against vertical cross sections of the CloudSat CPR. The predictions cover 100 x 100 pixels along 90 vertical bins of 240 m. These patches are combined to cover an area between 60° W–60° E and 60° N–60° S. In **(b)**, an object-based algorithm is employed to detect convective clouds and their cores within the predicted 3D radar reflectivity field. The temporal resolution of the data is 15 minutes. Through each point in time, we link identified cloud objects and filter the trajectories by the cloud top height (CTH), cloud base height (CBH), and number of cores to identify possible convective tracks (Brüning and Tost, 2025). In the current study **(c)**, we aim to quantify convective organisation by calculating organisation indices that are based on the area, distance, and number of objects in each cloud mask. The indices are calculated in 15-minute intervals for a period between March–August 2019. The results are used to analyse regional differences of convective organisation and to describe the relationship between convective organisation and cloud properties.

**Table 1.** Overview of MSG SEVIRI channels used to predict 3D radar reflectivities in this study.

| Channel | Wavelength ($\mu m$) | Description | Spatial resolution at nadir | Retrieval at nighttime |
|---------|----------------------|-------------|----------------------------|------------------------|
| IR3.9 | 3.48–4.36 | Near infrared window | 3 km | Yes |
| WV6.2 | 5.35–7.15 | Upper-troposphere water vapour | 3 km | Yes |
| WV7.3 | 6.85–7.85 | Lower-troposphere water vapour | 3 km | Yes |
| IR8.7 | 8.30–9.10 | Mid infrared window | 3 km | Yes |
| IR9.7 | 9.38–9.94 | Ozone sensitivity | 3 km | Yes |
| IR10.8 | 9.80–11.80 | Clean longwave window | 3 km | Yes |
| IR12.0 | 11.00–13.00 | Dirty longwave window | 3 km | Yes |
| IR 13.4 | 12.40–14.40 | CO2 sensitivity | 3 km | Yes |

## 2.1 Satellite data

To identify, track, and analyse convective clouds, we employ a machine learning (ML) algorithm that generates time series of 3D radar reflectivity fields based on 2D satellite observations, as described in Brüning et al. (2024). The input data are derived from the Spinning Enhanced Visible and Infrared Imager (SEVIRI) onboard the Meteosat-11 (MSG) satellite (Schmetz et al., 2002). The AOI is situated near the nadir of SEVIRI, which is positioned above the Equator at 0° longitude. SEVIRI captures multispectral imagery across 12 channels in the visible, near-infrared, and thermal-infrared ranges. Eleven of these channels offer a temporal resolution of 15 minutes and a spatial resolution of 3 km, while one high-resolution visible channel provides 1 km resolution at nadir. From these, we use eight channels to train our ML model (Table 1).

We employ vertical cross-sections of radar reflectivity from the 94-GHz Cloud Profiling Radar (CPR) onboard the polar-orbiting CloudSat satellite to validate our ML-based predictions. The CPR is an active radar instrument, which transmits microwave pulses toward Earth to detect vertical profiles of cloud hydrometeors. It has a vertical resolution of 240 m (distributed across 125 bins) and a horizontal resolution of 1.4 km across-track and 1.8 km along-track (Stephens et al., 2008). Our study employs data from the level-2 2B-GEOPROF product. While the CPR has a reduced sensor sensitivity at high altitudes, thin ice clouds like cirrus may be underrepresented. Moreover, the radar may be affected by signal attenuation at low altitudes caused by the topography (Sassen and Wang, 2008). To address these limitations, we limit the data to contain 90 height levels ranging from 2.4 km to 24 km. To improve the ML model performance, we filter the radar reflectivities by the CloudSat cloud mask quality flag to reduce the number of noisy pixels (Marchand et al., 2008).

## 2.2 3D cloud field reconstruction

In the following section, we describe the methodology used to reconstruct a 3D cloud field, based on the framework developed by Brüning et al. (2024). Our approach utilises a ML algorithm built on the 2D Res-UNet architecture — a modified convolutional neural network specifically designed for image segmentation tasks (Ronneberger et al., 2015). The model is primarily

**Table 2.** Modifications applied in this study to the Res-UNet originally proposed in Brüning et al. (2024)

| Parameter | Original configuration | Modification |
|---|---|---|
| Number of input channels | 11 | 8 |
| Loss function | L2 | L1 |
| Nighttime predictions | No | Yes |
| Average RMSE | 3.05 | 2.99 |

trained to reconstruct vertical cross-sections of the CloudSat CPR using data from the MSG SEVIRI satellite. Due to the U-Net architecture, the model is capable of producing full 3D radar reflectivity volumes rather than just 2D slices.

The AOI for the reconstructed 3D cloud field spans from 60° W to 60° E and from 60° S to 60° N, corresponding to 2400 × 2400 pixels in the horizontal dimensions. MSG SEVIRI satellite imagery serves as input to the Res-UNet model, setting the horizontal resolution of the 3D data to 3 km × 3 km. Initially, we used 11 spectral channels covering the visible, near-

115 infrared, and thermal-infrared ranges. However, the visible channels were excluded in this study to enable daylight-independent predictions (Tables 1 and 2).

The training data consist of 128 × 128 pixel patches of MSG SEVIRI imagery, spatially and temporally aligned with Cloud-Sat overpasses. Each training sample includes a diagonal CPR cross-section. To address the resolution mismatch between MSG SEVIRI and CloudSat, the CPR data are downsampled to match the horizontal resolution of MSG SEVIRI pixels. To mitigate

the strong class imbalance between cloudy and cloud-free conditions, cloud-free samples are limited to a maximum of 10 % of the training data. The model is trained on nine months of data and validated on a separate three-month period. It is optimised to reconstruct CloudSat-like 3D reflectivity volumes with a horizontal resolution of 100 × 100 pixels and 90 vertical levels. Predicted radar reflectivity values range from –25 to 20 dBZ and maintain the 15-minute temporal resolution of the MSG SE-VIRI input. An L1 loss function (mean absolute error, MAE) is used during training to evaluate performance. Direct validation

is only possible for the diagonal cross-section, which constitutes about 10 % of the data points in each training sample. During the three-month test period, the modified daylight-independent model achieves a root mean square error (RMSE) of 2.99 dBZ, improving upon the original model's average RMSE of 3.05 dBZ (Table 2).

To achieve complete spatial coverage of the domain (60° W to 60° E and 60° S to 60° N), individual 3D output patches are stitched together to form a contiguous volume of 2400 × 2400 × 90 pixels (Figure 1, a). This approach may enable consistent

spatial coverage, especially over remote oceanic regions where active sensors are scarce (Prein et al., 2024). Visual inspection confirms that no artifacts are present at tile boundaries, indicating seamless reconstruction across the domain. However, model accuracy tends to decrease with increasing distance from the MSG SEVIRI nadir. Finally, the 3D radar reflectivity volumes are concatenated along the temporal axis to create a 4D cloud field, which is then used to detect and track convective clouds. For the purposes of this study, we crop the domain to 1200 × 1200 pixels, covering the region from 30° W to 30° E and 30° N to

30° S — effectively focusing on the area between the Tropic of Cancer and the Tropic of Capricorn.

## 2.3 Detection and tracking of convective clouds and cores

Convective clouds are detected and tracked using the *tobac* package (Sokolowsky et al., 2024), which supports an object-based analysis of 3D meteorological data. The detection framework - as described in Brüning and Tost (2025) - proceeds in three stages: the cloud detection and tracking, the core detection, and the classification of potentially convective clouds. We use the ML-based predictions of the radar reflectivity as input data for the detection framework. While radar reflectivity does not directly measure vertical velocity, it may provide information for detecting hydrometeors associated with convective cloud development (Luo et al., 2008). By merging the 3D data fields along the temporal dimension, we receive a 4D time series that is fed into the tracking algorithm to create continuous trajectories with a temporal resolution of 15 minutes.

We identify potential candidates of convective clouds within the 3D cloud field by applying a fixed radar reflectivity threshold of -15 dBZ. This threshold is used to distinguish hydrometeors from background noise in the radar reflectivity data (Marchand et al., 2008). Although moderately restrictive, this threshold is intended to capture the full spatio-temporal evolution of convective clouds throughout their life cycle, thereby supporting the formation of contiguous trajectories (Esmaili et al., 2016). To reduce noise, we first apply a smoothing Gaussian image filter with an effective scale of half a standard deviation (sigma = 0.5) on the 3D radar reflectivity field. Next, we compute the centroids of potential cloud structures using a weighted center-of-mass approach, where the weight of each point is determined by its reflectivity value above the -15 dBZ threshold. Each identified centroid is assigned a unique identifier, which is retained throughout the subsequent tracking and segmentation processes. We then apply a 3D watershed segmentation algorithm to delineate the volume of individual cloud structures associated with each centroid. The algorithm places markers at the detected centroids within a binary 3D volume, where all other grid points are set to zero. From these markers, the algorithm expands outward through the volume, assigning reflectivity-based pixels to the corresponding cloud until the –15 dBZ threshold is reached. This process produces a labeled 3D cloud mask. Subsequently, we analyse the morphology of each cloud to determine whether any structures might represent a merger of multiple cloud systems. Each cloud's shape is characterised using the best-fitting ellipse, and we compute the aspect ratio — that is, the ratio of the major to the minor axis length. If the major axis is more than 75 % longer than the minor axis, we split the identified cloud into separate objects for further analysis. We track the labeled 3D cloud objects over time by linking them based on their estimated movement speed. At each 15-minute interval, we predict the expected position of a cloud object using its velocity from previous time steps. To streamline this linking process, we define a maximum search radius between time steps, within which only cloud objects are considered potential matches. When new clouds form, we assign them the average velocity of nearby clouds to estimate their likely movement (Heikenfeld et al., 2019). We require a minimum area overlap of 50 % to determine similarity between clouds across consecutive 15-minute intervals.

We aim to detect convective cores for each cloud object at every time step throughout its life cycle. For this purpose, we use the previously generated labeled 3D cloud mask. Core centroids are identified by locating local maxima in a combined metric that incorporates both smoothed radar reflectivity and the vertical extent of a contiguous potential core layer. Specifically, we calculate the mean radar reflectivity for each vertical cloud column, and determine the height of the core layer by counting the number of pixels with reflectivity values greater than 0 dBZ located above 5 km altitude. To fill isolated gaps in otherwise

**Table 3.** Cloud and core properties derived from the contiguous convective cloud trajectories.

| Feature type | Feature name | Definition |
|---|---|---|
| Cloud | Cloud area | Area of the cloud ($km^2$) |
| | Cloud top height (CTH) | Height of the cloud (km) |
| | Lifetime | Lifetime of the cloud trajectory (h) |
| | Surface type | Value of land-sea mask |
| Core | Number of cores | Number of identified convective core regions |
| | Core area | Average area of convective cores ($km^2$) |
| | Core height | Depth of the core in the vertical column (km) |

vertically continuous cores, we expand the threshold from 0 dBZ to -5 dBZ in columns that contain at least one pixel exceeding 0 dBZ (Luo et al., 2008; Igel et al., 2014). We then combine both indicators —average reflectivity and potential core vertical depth — for each pixel associated with a cloud label, resulting in a 2D layer where we search for local maxima. If at least one local maximum is detected, the corresponding locations are considered candidate core centroids. If no local maxima are found — for example, if no columns contain pixels above 0 dBZ at altitudes higher than 5 km — the cloud is recorded as having zero

cores for that time step. Otherwise, we use a 3D watershed segmentation algorithm to delineate the core volumes surrounding each centroid, allowing for multiple cores to exist within a single cloud at the same time.

## 2.4 Extraction of cloud properties

We use the labelled cloud masks to extract cloud and core properties at each point in time. Moreover, we compute average properties across the cloud's lifetime to derive distinct key properties that may characterise the trajectory. These properties

include the cloud lifetime, cloud area, cloud top height (CTH), number of cores, and mean core area and height (Table 3). The cloud area is computed from the column-wise maximum horizontal extent of the 3D cloud mask, while CTH is derived from the vertical extent. For the cloud lifetime, we extract the time (in hours) between the first and last detection of each trajectory of the labelled pixels. Each cloud track is classified as either marine (sea) or continental (land) using a binary land-sea mask. For this purpose, we determine the most frequent (modal) surface type across all grid points along the cloud trajectory. While

this method does not capture changes in surface type throughout the cloud's life-cycle, it may provide insights on the effect of the most frequently occurring surface type. For clouds with one or more cores, we count the maximum number of cores associated to the trajectory. Moreover, the core area and height are derived from the column-wise maximum horizontal extent and vertical extent of the previously identified cores, similar to the cloud area and CTH.

## 2.5 Filter convective cloud trajectories

We filter the cloud trajectories to exclude possibly non-convective tracks from the analysis. For that purpose, we employ three criteria occurring for at least a single timestep of 15 minutes: (a) One or more core regions, (b) radar reflectivity of higher than 0 dBZ at 10 km height, and (c) minimum CTH of 10 km and maximum CBH of less than 5 km. While we do not require the convective clouds to have a CTH higher than 10 km at every time step during their trajectory, we discard trajectories that never reach the CTH threshold. After filtering the dataset, we receive 375,000 uniquely labeled 3D cloud objects, each associated

with a continuous time trajectory and structural information about cloud and core properties (Figure 1, b).

For further analysis, we exclude cloud tracks detected for a single time step of 15 minutes. This results in a refined dataset of 354,073 convective cloud trajectories between March and August 2019. In Fig. 2, we showcase the spatio-temporal distribution of the cloud trajectories. Most clouds are located between 5° S and 20° N, with peak activity from 5°–10° N (Figure 2, a). Approximately 75 % of cloud tracks occur over ocean, with land-based tracks comprising the remaining 25 % (Figure 2, b).

Compared to the land-sea distribution of grid points across the AOI, we observe a 10 % shift toward ocean for detected clouds (Brüning and Tost, 2025). Most trajectories contain a single convective core (70 %), while the proportion of multi-core systems declines with increasing core count (Figure 2, c). Cloud frequency is higher in March–May (MAM) than in June–August (JJA) (Figure 2, d). The diurnal variability is less pronounced than these monthly differences along the period (Figure 2, e). We observe a high proportion of clouds have a lifetime between 0–3 h (42 %) or 3–6 h (37 %). Hence, about 80 % of the cloud

tracks last for less than 6 h. The proportion of cloud tracks with a longer lifetime is considerably lower (Figure 2, f).

While this framework enables a seamless tracking of convective systems along the ML-based 4D time series, it remains subject to several limitations. The predicted data display a ML-based extrapolation of the received CloudSat CPR reflectivities. Hence, they include uncertainties connected to the ML model, such as the blurriness of predictions induced by the loss function which optimizes towards the mean. We receive few information on thin ice clouds due to a reduced sensitivity of the CloudSat

CPR to ice clouds in high altitudes (Sassen and Wang, 2008). Moreover, the detection framework rests on an object-based perspective to investigate atmospheric processes. We note the identified trajectories may underlie simplifications caused by an inherent subjectivity of the thresholds applied in the cloud detection step. Nevertheless, the approach may help to bring further insights into the structure and organisation of convective clouds.

## 3 Method

### 3.1 Quantifying convective organisation

Convective organisation describes the contrast between convective cells randomly distributed in space and time from those clustering together inducing a stronger convective organisation (Pendergrass, 2020). While there exist various organisation indices to quantify the spatial clustering, each index alone may not sufficiently characterise convective organisation (Stubenrauch et al., 2023). Instead, all indices have specific limitations, such as a sensitivity to the mean cloud area or to the number

of individual objects. In response, we chose a combination of three organisation indices (SCAI, COP, ROME). All indices are

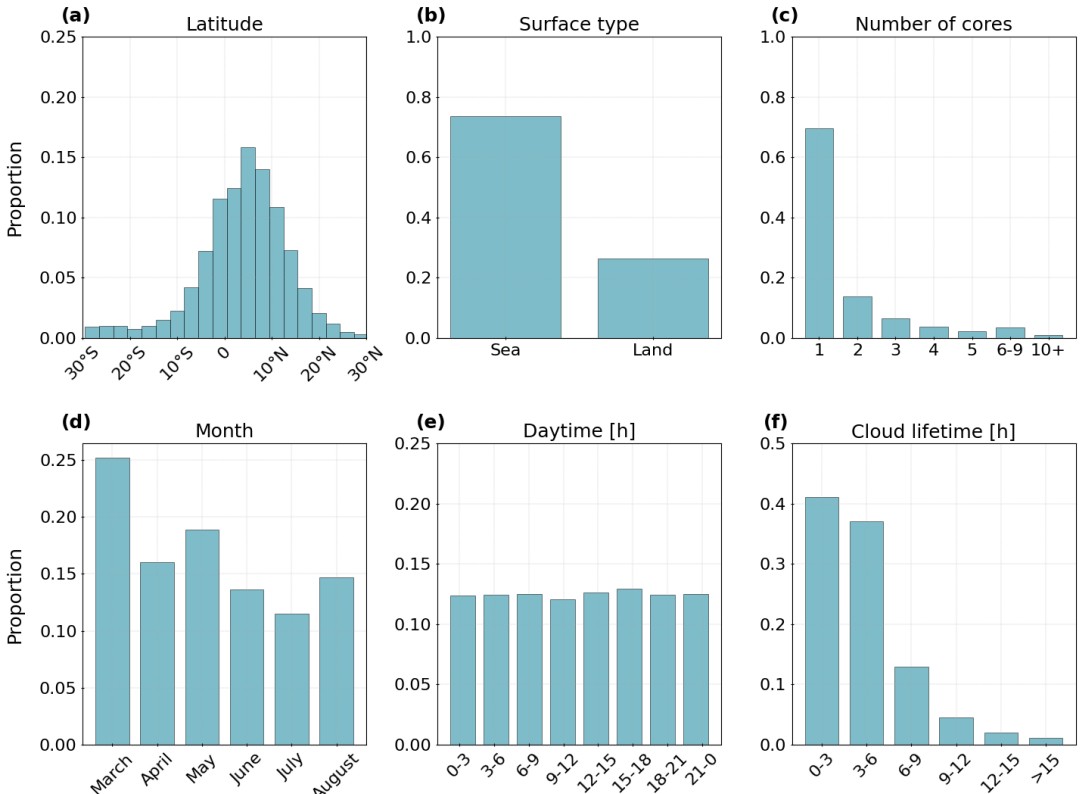

**Figure 2.** Summary of cloud tracks retrieved between March–August 2019 (n = 354,073). We show the spatial and temporal distribution of the data based on **(a)** the latitude grouped in 3° intervals between 30° S and 30° N, **(b)** the surface type derived from a land-sea mask, **(c)** the number of cores, **(d)** the month, **(e)** the daytime, and **(f)** the cloud lifetime.

designed to work on 2D data. Their input is a binary field, in this case the cloud mask derived in Sect. 2.3, representing the location of labeled convective objects (Semie and Bony, 2020). We calculate the three organisation indices for the AOI between 30°W–30°E and 30°N–30°S at each timestep of 15 minutes (Figure 1, c).

The first index is the simple-to-compute and straightforward Simple Convective Aggregation Index (SCAI). SCAI describes the ratio of the degree of convective disaggregation to a potential maximal disaggregation within a domain (Tobin et al., 2012). The index is unitless and inversely proportional to the number of grid boxes. SCAI compares the number of objects in the domain ($N$) and the geometric mean distance ($D_0$) between the centroid positions of all possible object pairs to the possible maximum number of objects that can exist in the domain ($N_{max}$) and the characteristic domain size ($L$).

$$\text{SCAI} = \frac{N D_0}{N_{max} L} 1000. \tag{1}$$

SCAI is a unitless index between 0 and infinity whereas lower values point towards a stronger convective organisation. By design, calculating SCAI requires the presence of multiple cloud clusters.

The Convective organisation Potential (COP) was developed by White et al. (2018) as an adaptation of the $I_{org}$ index. It assumes objects that are larger and closer together are more likely to interact with each other. In contrast to SCAI, the index takes the cloud size into account. COP uses the number of objects ($N$), the area of the i-th object ($A_i$) and the j-th object ($A_j$), and the distance between the centroids of the i-th and the j-th object ($d_{ij}$). It adds the characteristic domain size ($L$) and the total image size ($L_2$). The index is defined by

$$\text{COP} = \frac{2}{N(N-1)} \sum_{i=1}^{N} \sum_{j=i+1}^{N} \frac{\sqrt{A_i/\pi} + \sqrt{A_j/\pi}}{d_{ij}} \tag{2}$$

which is the mean over all the possible pairs of the interaction potential. COP is a positive and unitless index between 0–1 whereas higher values indicate a stronger convective organisation. Larger and closer objects have a higher increase in COP than small and widespread objects (Pscheidt et al., 2019).

Additionally, we calculate the Radar Organisation MEtric (ROME). The index considers the average size, proximity, and size distribution of convective clouds. Initially, it was designed to analyse radar observations. However, it also worked well with other data (Bläckberg and Singh, 2022). The index assesses connections between pairs of continuous convective regions and assigns a weight to each pair that increases with their respective areas and decreases with their separation distance. The weight is equal to the area of the larger contiguous convective region plus a contribution from the smaller contiguous convective region that depends on the separation distance (Retsch et al., 2020). It employs the smallest distance between the edges of the i-th and the j-th object in the domain ($\tilde{d}_{ij}$) to define

$$\text{ROME} = \frac{2}{N(N-1)} \sum_{i=1}^{N} \sum_{j=i+1}^{N} \cdot \left[ A_{ij}^{(max)} + A_{ij}^{(min)} \cdot min\left(1, \frac{A_{ij}^{(min)}}{\tilde{d}_{ij}^2}\right) \right] \tag{3}$$

where $A_{ij}^{(max)} = \max(A_i, A_j)$ and $A_{ij}^{(min)} = \min(A_i, A_j)$. ROME is a positive index measured in units of area. Its value consists of a contribution from the mean area of contiguous convective regions and the distribution of sizes and interaction between different contiguous convective regions. The index is positive, with an increasing ROME value corresponding to a stronger aggregation.

While SCAI and COP are easy to compute, the calculation of ROME is less convenient. Since it has been designed to retrieve information from radar reflectivities, we include the index in our study. In contrast to SCAI and COP, ROME may also be computed when only a single object is present. As evaluated by, e.g., Mandorli and Stubenrauch (2024) and Biagioli and Tompkins (2023), each index has its own strengths and weaknesses. SCAI is insensitive to the size of the objects and mainly dominated by the variability in the number of clouds. However, it is less affected by shifts in time and space which induce high fluctuations of the index values, e.g, due to changes in the resolution of the input image or between two consecutive time steps. In contrast, the calculation of COP includes the object area. While COP correctly increases with the proximity and size, it is sensitive to noise caused in a domain with only a few objects. The index is correlated to the image resolution and shows a high variability for consecutive time steps. While ROME is more noise-safe and independent of the dataset resolution, it strongly connects to the object size. Compared to SCAI and COP, ROME shows a lower variability along consecutive time steps and it is less sensitive to the proximity of objects. Despite these limitations, we employ these indices that have been applied before

| (a) Grid-based | (b) Moving-window |

**Figure 3.** Visualisation of the moving-window approach used to calculate the organisation indices from the labeled 3D cloud mask. When using a fixed grid cell size **(a)**, clouds may be split at the borders leading to an enhanced small-scale value variability between the subsets. In this study, we employ a moving-window which iterates along the grid cells with a kernel size of $1° \times 1°$. At each iteration, we update the index value by calculating the mean between the index at the former and current subset **(b)**. In contrast to **(a)**, the indices are less influenced by a single grid cell and rather represent the average composed of all window locations.

in our studies to retrieve comparable results. However, building an adapted methodology for assessing convective organisation
may benefit future research.

### 3.2   Calculating grid-based organisation indices

To assess regional variability in convective organisation, we refrain from computing organisation indices over the entire domain. Instead, the AOI is partitioned into overlapping $3° \times 3°$ grid cells (e.g., Semie and Bony (2020); Tobin et al. (2012)). Given that the spatial extent and number of convective cloud elements affect the resulting index values, it may be beneficial to mitigate
artifacts arising from cloud systems intersecting grid boundaries. In response, we implement a moving-window approach. The initial window is anchored at the northwestern corner of the AOI ($27°–30°$ N, $27°–30°$ W) and is incrementally shifted by $1°$ in both the zonal and meridional directions (Figure 3). For each time step, the spatial organisation indices (SCAI, COP, and ROME) are computed within a $3° \times 3°$ window. To enhance statistical robustness and reduce sensitivity to window placement, we calculate a local mean across adjacent overlapping windows, assigning the averaged value to the central grid cell.
This approach may reduce boundary-related discontinuities and contribute towards a more stable representation of convective structure, particularly in regions where cloud systems span multiple windows (Jin et al., 2022).

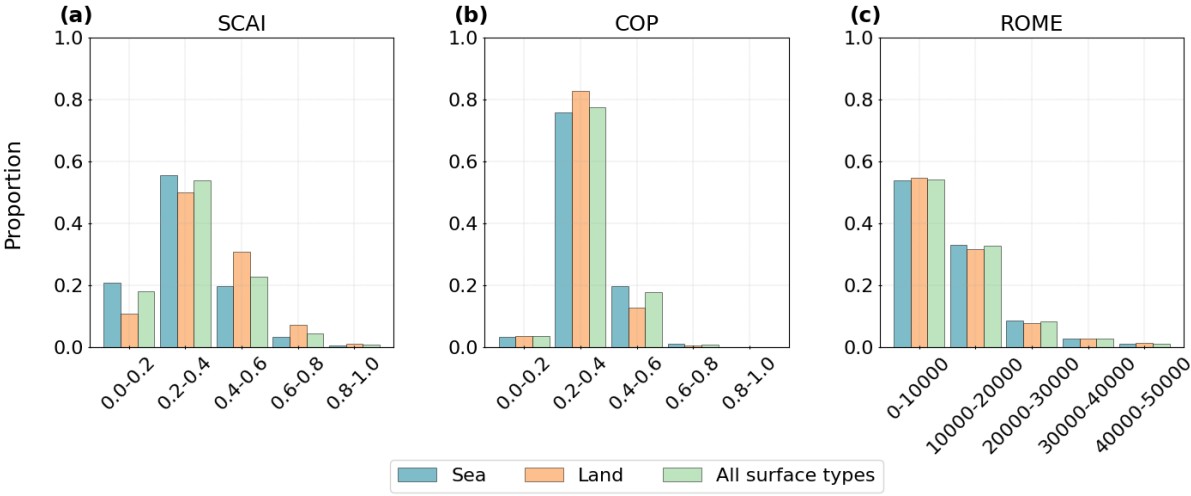

**Figure 4.** Distribution of convective organisation indices grouped by the surface type (for clouds over the sea, over land, and for all cloud tracks independent of the surface type). We see the frequencies of **(a)** SCAI, **(b)** COP, and **(c)** ROME.

## 4 Results

### 4.1 Distribution of organisation indices

This section analyses the spatial and temporal distributions of the three convective organisation indices: SCAI, COP, and ROME. Lower SCAI values (or higher COP and ROME values) are indicative of enhanced convective clustering, reflecting stronger spatial organisation. Conversely, high SCAI (low COP or ROME) values correspond to more scattered convective structures, implying weaker organisation (Biagioli and Tompkins, 2023).

Figure 4 (a) shows that SCAI values range between 0 and 1, with a peak concentration between 0.2–0.4. Oceanic regions have a slightly higher frequency of SCAI values lower than 0.4, whereas values higher than 0.4 are more common over land. This finding may suggest SCAI detects stronger convective organisation over water. COP values are mainly distributed between 0.2 and 0.6. Over the ocean, values above 0.4 are more frequent, whereas over land, lower values dominate — again pointing to stronger convective organisation over the ocean (Figure 4, b). ROME displays a right-skewed distribution, with most values falling below 20,000. Differences between land and ocean are minor compared to SCAI or COP (Figure 4, c). Overall, the results may indicate a marginally stronger convective organisation over oceanic regions, with ROME showing the weakest land–sea contrast.

Figure 5 compares the diurnal cycle, changes to core numbers, and latitudinal averages of the indices over land and ocean within the 30° S–30° N domain. For SCAI, we find predominantely lower values over land throughout the day. The diurnal cycle exhibits minima between 09:00–12:00 UTC and 21:00–00:00 UTC, particularly over land. SCAI increases between 00:00–06:00 UTC and 12:00–21:00 UTC (Figure 5, a). COP shows a weaker temporal variability than SCAI but with values

consistently suggesting higher organisation over the ocean (Figure 5, d). Diurnal variations in SCAI and COP reach up to 10 % of the indices' scales. ROME shows daytime (06:00–18:00 UTC) and nocturnal (00:00–03:00 UTC) peaks over land and mostly nocturnal peaks (21:00–06:00 UTC) over the ocean (Figure 5, g). Collectively, the indices indicate maximum convective organisation occurs over land in the afternoon and over the ocean in the night and early morning; minima occur at night over land and from noon to afternoon over the ocean. SCAI and ROME decrease with increasing numbers of convective cores

(Figure 5, b, h). For ROME, organisation decreases up to five cores but increases beyond six, particularly over land (Figure 5, h). COP, by contrast, remains largely unaffected by core number as it points out only a slight decrease of convective organisation with increasing core numbers and an increase for clouds with more than six cores (Figure 5, e). This finding suggests for SCAI a stronger convective organisation for higher core numbers, which opposes the results for COP and ROME. Latitudinally, all indices show stronger organisation near the equator, although the spatial variability differs for the three indices. As SCAI is

sensitive to object numbers, a higher frequency of detected clouds near the equator and less clouds near the borders of the AOI may contribute to the variability of the index (Figure 2, Figure 5, c). COP varies less with latitude, whereas we observe slightly higher values between 20° S–20° N (Figure 5, f). For ROME, we find the highest variability between latitudinal averages and surface types with peaks over land between 20° S and the equator, and over oceanic regions near the equator and between 20°– 30° S (Figure 5, i). Compared to other regions in the domain, the results show a considerably stronger convective organisation

over the southern Atlantic Ocean (30° S) for SCAI and ROME.

## 4.2   Spatial patterns and statistical relationships

Figure 6 presents the spatial distribution of the three organisation indices (SCAI, COP, ROME), along with associated cloud and core properties, interpolated onto a 3° × 3° grid and displayed as latitudinal cross-sections. Distinct regional patterns emerge across the AOI, highlighting potential links between convective organisation and cloud structure. Near the equator - particularly

over continental Africa - higher SCAI values may coincide with a smaller cloud area, elevated cloud top height (CTH), and taller convective cores. In contrast, lower SCAI values are found primarily over the Atlantic Ocean and in subtropical zones of northern and southern Africa (15°–30° N/S). These regions are characterized by larger cloud areas, a lower CTH and lower core heights (Figure 6, a, d, e, i). For the cloud lifetime, the number of cores, and the core area, we observe a less distinct connection. They show a high spatial variability along the AOI, whereas a longer cloud lifetime, a higher number of cores, and

a larger core area may be related to a smaller cloud area, higher CTH, and higher core height in near-equator regions (< 15° N/S) and to a larger cloud area, lower CTH, and lower core height near the tropics (> 15° N/S) (Figure 6, d–i). COP exhibits low spatial variability, with most values between 0.2–0.5 (Figure 6, b). ROME, in contrast, displays pronounced spatial differences: high values occur between 15°–30° N/S, particularly over the Atlantic Ocean and near African coastlines, and near the equator over the Gulf of Guinea and continental Africa (Figure 6, c). Over the Sahel, clouds tend to be large, with numerous, wide

but relatively shallow cores. Over the South Atlantic (15°–30° S), cloud systems exhibit large areas, long lifetimes, and a high number of cores. This pattern may reflect cloud clustering in the AOI may be influenced by oceanic circulation and adjacent landmasses (Atiah et al., 2023). Overall, regions with stronger convective organisation - indicated by low SCAI and high COP or ROME - tend to exhibit smaller clouds with low CTH and core heights. For the number of cores, the core area, and cloud

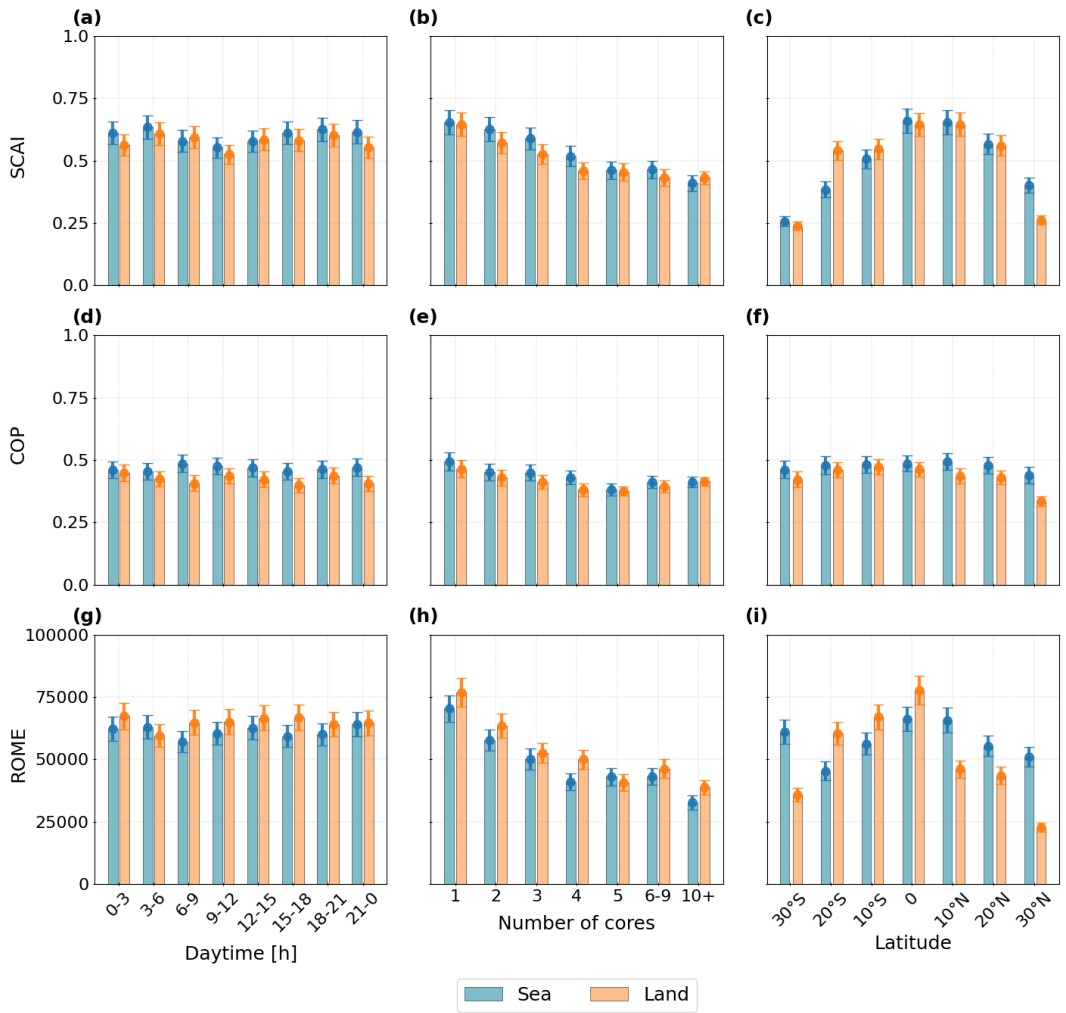

**Figure 5.** Comparison of the organisation indices **(a)**–**(c)** SCAI, **(d)**–**(f)** COP, and **(g)**–**(i)** ROME. The columns show the diurnal cycle (grouped in 3 h intervals), the number of cores, and the latitude (in $10°$ intervals) grouped by the surface type (land, sea). Vertical errorbars show the standard error of the mean.

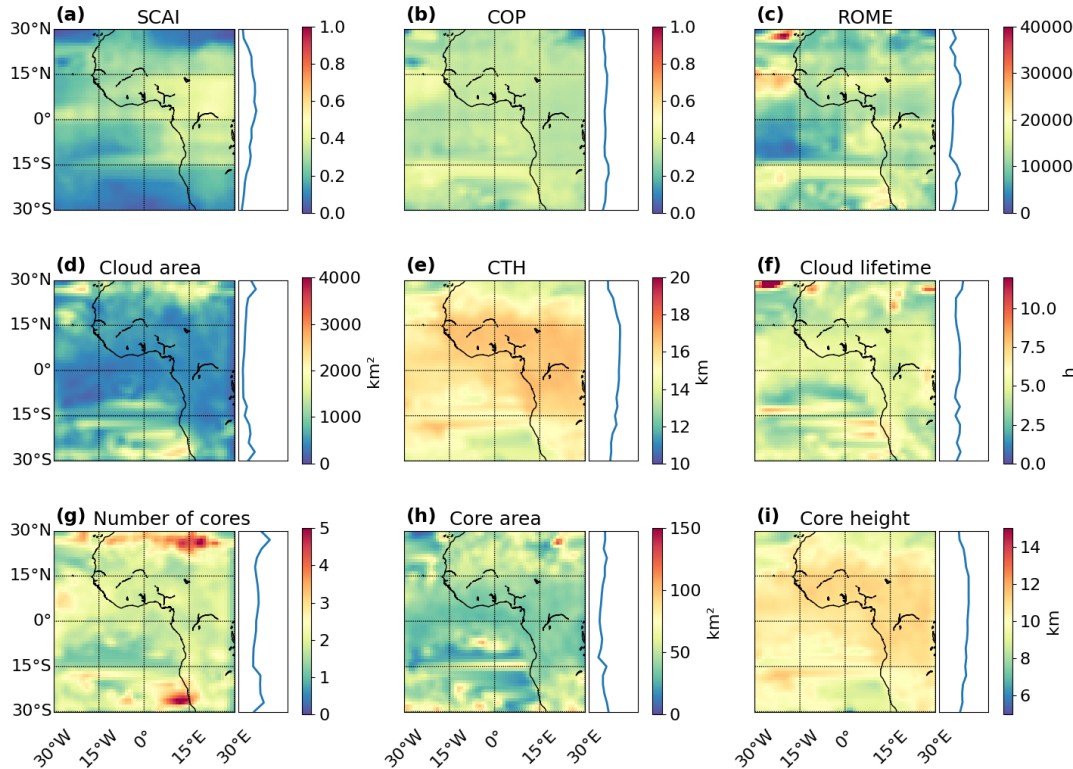

**Figure 6.** Mean values for **(a)** SCAI, **(b)** COP, **(c)** ROME, **(d)** the cloud area, **(e)** the CTH, **(f)** the cloud lifetime, **(g)** the number of cores, **(h)** the convective core size, and **(f)** the core height. The plot shows the spatial distribution in the AOI interpolated on a 3° x 3° grid (left) and the average for each latitude between 30° N and 30° S (n = 354,073).

lifetime, a higher regional variability may be apparent. These contrasts are most apparent between equatorial and subtropical regions.

To quantify the relationship between organisation indices and cloud properties, we compute Spearman's rank correlation coefficient R using data from all cloud tracks (Figure 7). The logarithmic distributions reveal a general skew toward low values for SCAI, ROME, cloud area, lifetime, number of cores, and core area. The correlation analysis shows that COP and ROME may be positively associated with cloud area, lifetime, CTH, number of cores, and core height (Figure 7, g–r). In contrast, SCAI

is negatively correlated with all of these properties except for CTH and the core height (Figure 7, a–f). For the core area, we see a weak negative correlation to all indices. The findings suggest that stronger convective organisation may be statistically linked to larger, longer-lived cloud systems, a higher CTH and core height, and more cores. Interestingly, these statistical relationships contrast with some spatial patterns in Fig. 6. For instance, while higher ROME values spatially co-occur with smaller clouds and shorter lifetimes in some regions, correlation coefficients suggest that, overall, organisation increases with cloud area and

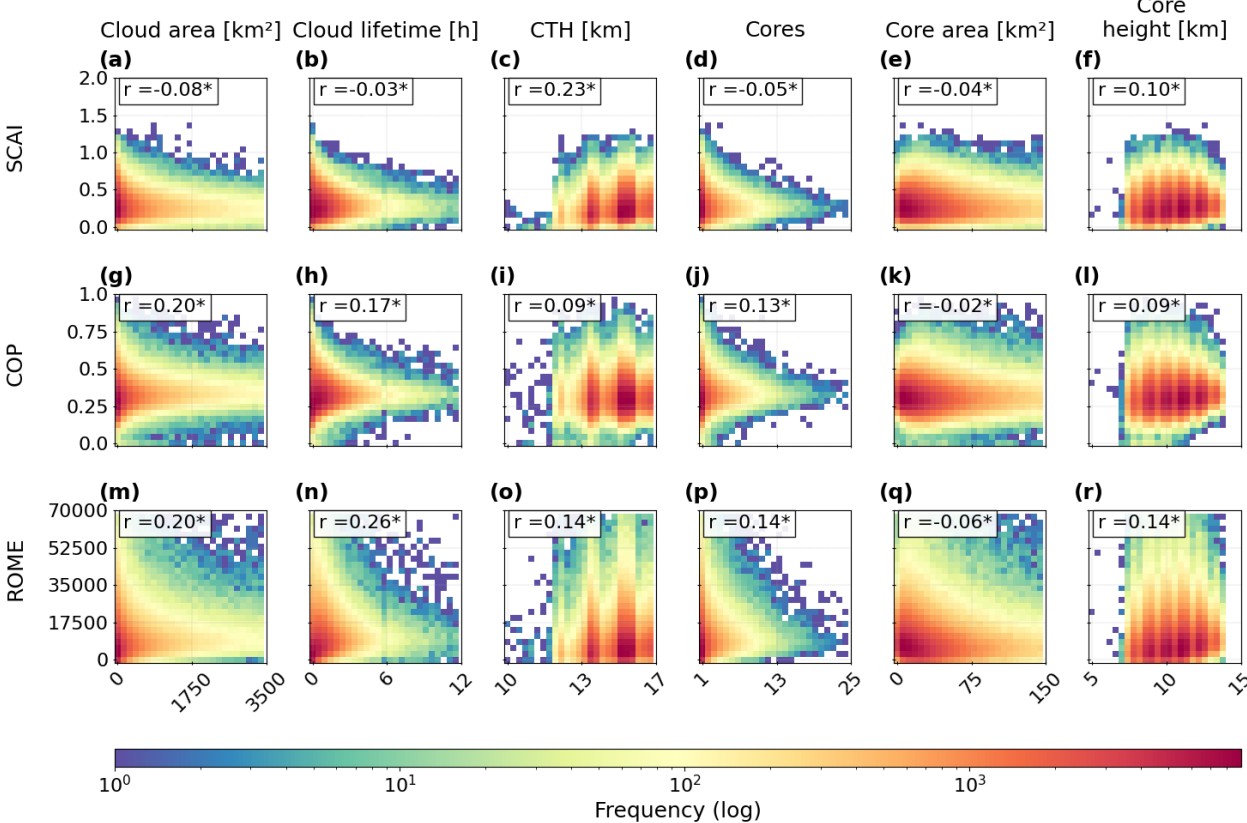

**Figure 7.** Histogram showing the logarithmic frequency distribution for the SCAI, COP, and ROME against the **(a, g, m)** the cloud area, **(b, h, n)** the cloud lifetime, **(c, i, o)** the CTH, **(d, j, p)** the number of cores, **(e, k, q)** the convective core size, and **(f, l, r)** the core height. In each histogram, we add the Spearman correlation coefficient R to quantify the strength of the relationship (n = 354,073) (significant with *: p < 0.1).

duration. However, most correlations are weak, with maximum coefficients around 0.26 between ROME and the cloud lifetime. They highlight the complex and regionally variable nature of these relationships.

### 4.3    Temporal variability of cloud properties and organisation indices

The previous analysis suggests the overall correlation between convective organisation indices and cloud/core properties is generally weak. In this section, we aim to capture changes in convective behaviour along the period that may help to explain

observed patterns. For this purpose, we filter the dataset into two subsets between March to May (MAM, n = 212,984) and June to August (JJA, n = 141,089). Here, we analyse monthly means over land and ocean (Figure 8). Overall, differences between land and ocean typically span up to 10 % of each index's dynamic range (Figure 4). For the monthly changes, most variables do not exhibit a linear trend. SCAI, COP, and the number of cores remain relatively stable, while the CTH and core height

vary non-monotonically (Figure 8, a, b, e, g, i). SCAI generally decreases over the ocean and increases slightly over land until
June, returning to near-March values by August (Figure 8, a). COP displays similar changes over land, while over the ocean, it
increases marginally throughout the period (Figure 8, b). ROME exhibits the strongest variability, increasing over both surface
types, especially over the ocean (Figure 8, c). Notably, average CTH, cloud lifetime, and core height are consistently higher
over land, whereas cloud and core areas are larger over the ocean, particularly from May to August (Figure 8, d–f, h–i). The
number of cores remains fairly constant across the time series (Figure 8, g). Over the ocean, we observe a steady increase in
cloud and core area and a decrease in CTH. Core height peaks in May and July, followed by a decline in August. Over land,
temporal changes are less pronounced, though the core area shows a slight dip until May and then rises again by August.

Figure 9 illustrates the mean differences between boreal spring (MAM) and boreal summer (JJA), calculated as MAM
minus JJA. The data are interpolated onto a $3° \times 3°$ grid and averaged along latitudes. While SCAI shows only a weak monthly
variability (Figure 8, a), we observe regional differences of up to $\pm$ 0.4 across the AOI. Notably, SCAI increases between
$15°–30°$ N and decreases south of $15°$ N, especially over the Gulf of Guinea and central Africa ($0°–15°$ S) (Figure 9, a).
COP tends to increase between $0°–15°$ N and decrease north of $20°$ N during JJA, although these changes are generally small,
remaining within $\pm$ 0.2. More pronounced decreases of up to -0.4 are seen south of $15°$ S, over the Sahel, and near the Canary
Islands (Figure 9, b). ROME shows small localised decreases during JJA across northern Africa, the Canary Islands, and coastal
southern Africa. In contrast, it increases between $15°$ N and $15°$ S, especially near the equator and around $15°$ S (Figure 9, c).
The spatial patterns of the cloud properties partly align with (cloud area, cloud lifetime) or oppose (CTH, core height) those
observed for the organisation indices. For instance, both cloud area and lifetime tend to increase in the Southern Hemisphere
during JJA, though the magnitude and intensity of these changes vary considerably across the AOI. Over northern continental
Africa and the Congo River basin, cloud area and lifetime decline from MAM to JJA (Figure 9, d, f). In contrast, the CTH
increases north of $15°$ N and decreases south of $15°$ S during JJA (Figure 9, e). The number of cores reveals a less consistent
pattern, with a high spatial variability. Increases are observed during JJA over the Atlantic Ocean, the West African coast,
northern continental Africa, and the equatorial rainforests. Conversely, declines are noted over coastal areas north of $15°$ N
and south of the equator (Figure 9, g). Similarly, the core area displays a rather fragmented spatial pattern across the AOI, with
slightly larger values in the Northern Hemisphere and a particular increase south of $20°$ S during JJA (Figure 9, h). The core
height broadly follows the same pattern as CTH, rising north of $15°$ N and declining south of $15°$ S. Additionally, core heights
increase between $0°–10°$ S in boreal summer (Figure 9, i). Observed increases of the cloud area, core area, and cloud lifetime
may coincide with a reduction in CTH, core height, and core number. However, there appear spatial and temporal variations
which may reflect the influence of, e.g., local circulations and land–sea contrasts on convective development across the AOI.

We evaluate how the relationships between organisation indices and cloud/core properties evolve along the two seasonal
subsets by comparing the correlation coefficients between MAM and JJA (Table 4). Overall, SCAI maintains negative correla-
tions with cloud properties, while COP and ROME remain positively correlated. The direction of correlation does not change
along the period, though some coefficients vary in strength. From boreal spring to summer, correlations between SCAI and
cloud properties increase slightly - except for the CTH and core height. Correlations between COP and cloud properties pre-
dominantly increase, whereas the differences are lower than for SCAI. For ROME, we see an increase for the correlation to

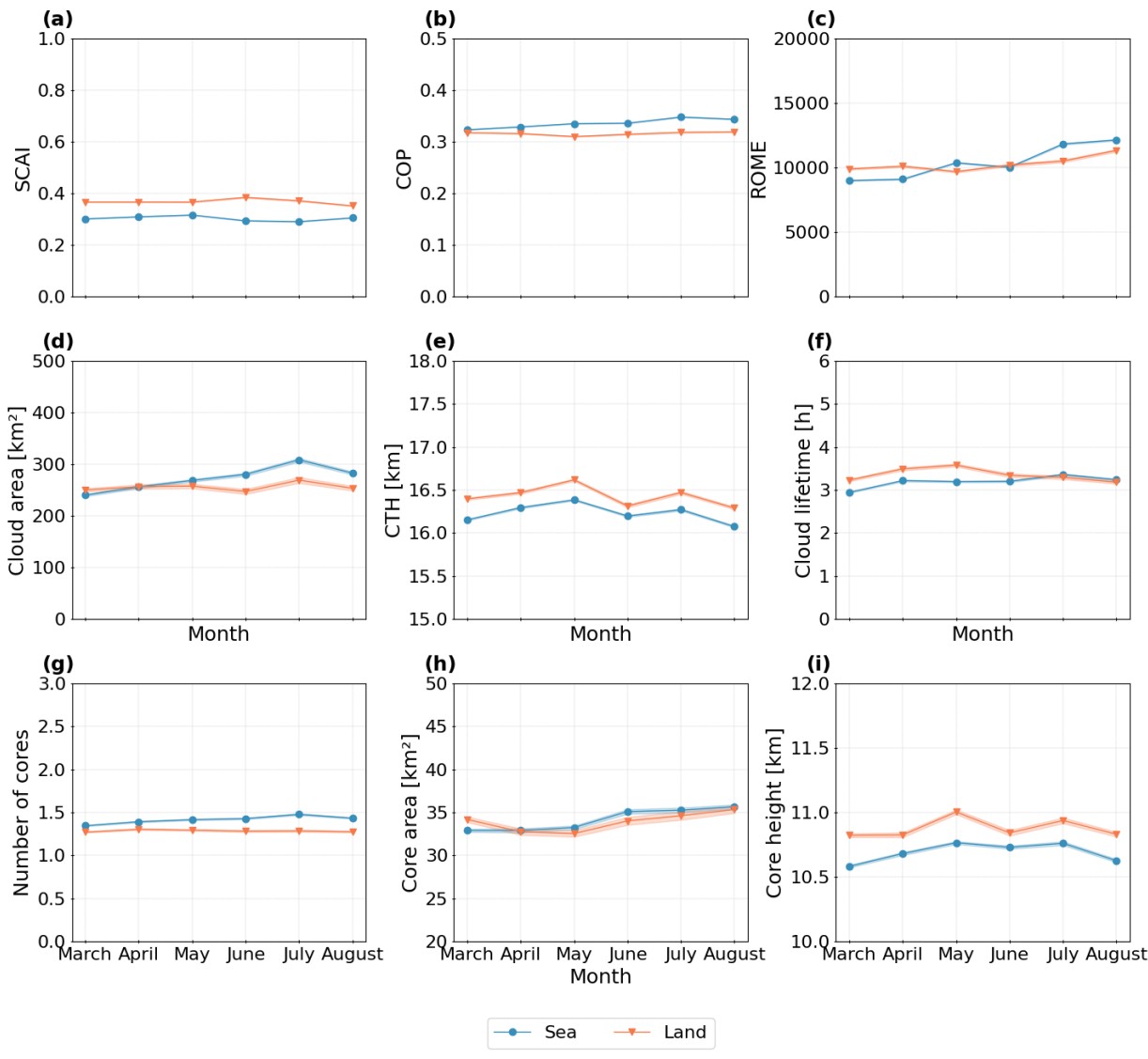

**Figure 8.** Monthly changes of the organisation indices and cloud and core properties between March and August 2019. We show **(a)** the SCAI, **(b)** the COP, **(c)** the ROME, **(d)** the cloud area, **(e)** the CTH, **(f)** the cloud lifetime, **(g)** the number of cores, **(h)** the convective core size, and **(f)** the core height grouped by the surface type. Line plots show the mean value (solid line) with a confidence interval of 95 %.

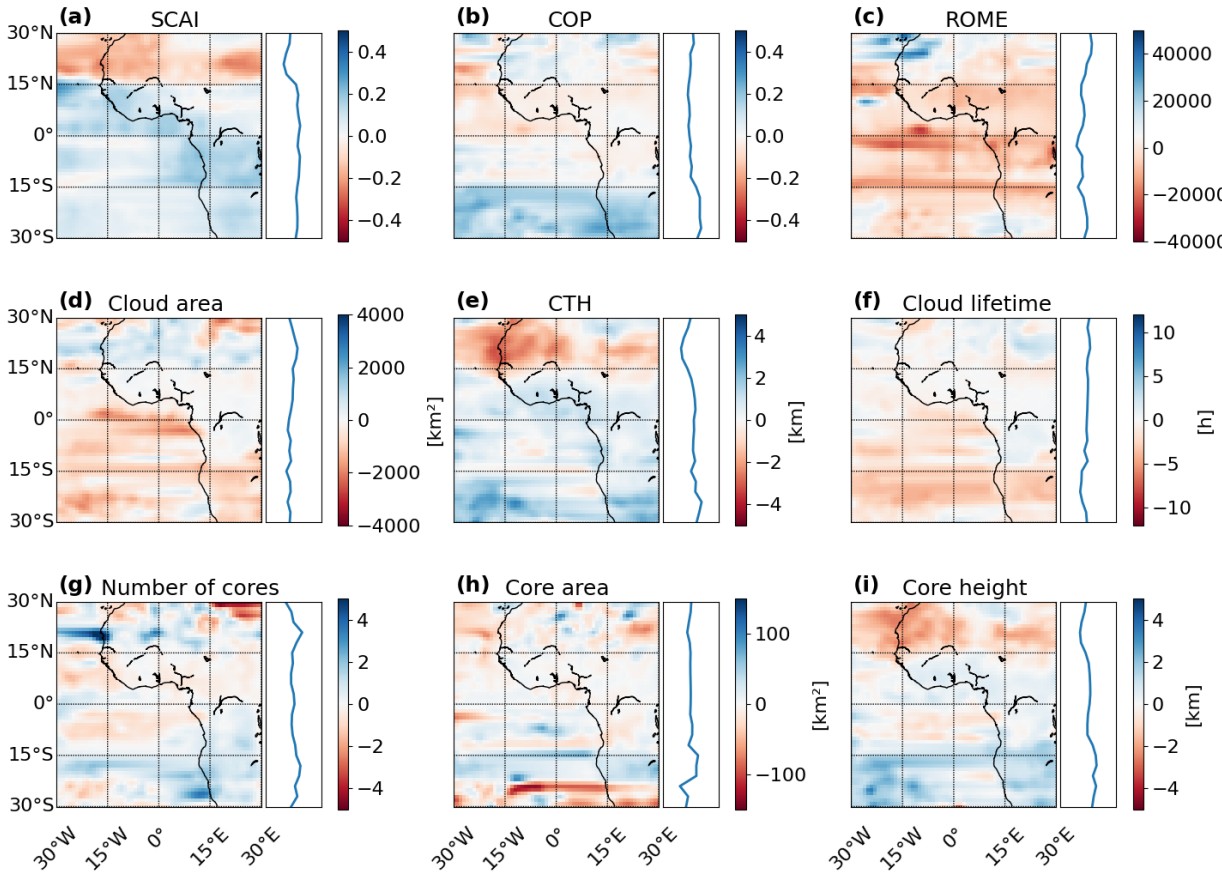

**Figure 9.** Changes between boreal spring (MAM, n = 212,984) and summer (JJA, n = 141,089) showing the average differences for MAM minus JJA. The plot shows the spatial distribution in the AOI interpolated on a 3° x 3° grid (left) and the average value for each latitude between 30° N and 30° S (right). Values are derived for **(a)** SCAI, **(b)** COP, **(c)** ROME, **(d)** the cloud area, **(e)** the CTH, **(f)** the cloud lifetime, **(g)** the number of cores, **(h)** the convective core size, and **(f)** the core height.

**Table 4.** Spearman's R for the SCAI, COP, and ROME against the cloud area, the cloud lifetime, the CTH, the number of cores, the convective core size, and the core height. The table shows the correlation coefficient for boreal spring (MAM) and summer (JJA) and the difference (MAM - JJA).

| | SCAI | | | COP | | | ROME | | |
| --- | --- | --- | --- | --- | --- | --- | --- | --- | --- |
| | MAM | JJA | Difference | MAM | JJA | Difference | MAM | JJA | Difference |
| Cloud area | -0.06 | -0.13 | 0.07 | 0.12 | 0.12 | 0.00 | 0.09 | 0.07 | 0.02 |
| Cloud lifetime | -0.01 | -0.12 | 0.11 | 0.05 | 0.10 | -0.05 | 0.08 | 0.11 | -0.03 |
| CTH | 0.24 | 0.12 | 0.11 | 0.04 | 0.07 | -0.03 | 0.04 | 0.07 | -0.03 |
| Number of cores | -0.04 | -0.08 | 0.05 | 0.07 | 0.09 | -0.02 | 0.05 | 0.05 | 0.0 |
| Core area | -0.04 | -0.04 | 0.00 | 0.02 | 0.01 | 0.01 | 0.01 | -0.03 | 0.02 |
| Core height | 0.12 | -0.01 | 0.11 | 0.05 | 0.08 | -0.03 | 0.05 | 0.07 | -0.02 |

the cloud lifetime, CTH, and core height, and a decrease to the cloud area and core area. However, these shifts are small, with changes up to 0.11 (SCAI vs. cloud lifetime, CTH, and core height). Despite apparent spatial patterns and temporal shifts in convective cloud organisation and structure as seen in Figs. 8 and 9, statistical relationships remain overall weak. These weak correlations suggest that relations may be affected by additional factors which were not integrated in our analysis, such as the large-scale circulation, interannual variations (caused by, e.g., El Niño-Southern Oscillation (ENSO)), or local topography.

### 4.4 Investigating effects of convective organisation

To identify regional patterns of convective organisation and their effects on cloud properties, we adopt a percentile-driven approach. There exist no universally defined thresholds to distinguish between weak and strong convective organisation. In response, we compute the 10th, 25th, 75th, and 90th percentiles based on the distribution of each organisation index (SCAI, COP, and ROME) using the cloud tracks between March to August 2019 (Table 5). These percentiles serve as thresholds to classify the data into subsets of weak and strong convective organisation, as induced by the interpretation of the indices: strong organisation may be related to low SCAI and high COP/ROME, weak organisation to high SCAI and low COP/ROME (Biagioli and Tompkins, 2023; Semie and Bony, 2020). Following, regions of strong convective organisation are defined as cloud tracks with an index value below the 10th percentile for SCAI or above the 90th percentile for COP and ROME. Conversely, regions of weak organisation correspond to values that lie above the 90th percentile for SCAI or below the 10th percentile for COP and ROME. To identify spatial and temporal patterns of convective organisation, we create two subsets from all data points in the dataset, whereas one represents the 10 % strongest convective organisation (Q10 for SCAI; Q90 for COP and ROME, hereafter: P90), and the other representing the 10 % weakest convective organisation (Q90 for SCAI; Q10 for COP and ROME, hereafter: P10). These may represent so-called "hotspots". We also define the interquartile range (IQR, values between the

**Table 5.** Percentiles for the organisation indices (SCAI, COP, ROME) derived from the time series between March and August 2019. The table contains the percentiles Q10, Q25, Q75, and Q90 which are used as thresholds to filter subsets of strong or weak convective organisation.

|      | Q10       | Q25       | Q75        | Q90        |
|------|-----------|-----------|------------|------------|
| SCAI | 0.165     | 0.224     | 0.418      | 0.528      |
| COP  | 0.237     | 0.278     | 0.381      | 0.443      |
| ROME | 3260.327  | 5652.496  | 14695.356  | 22659.608  |

25th–75th percentile) to represent a baseline, which is used to contrast the spatial distribution of average organisation against the identified hotspot regions.

### 4.4.1 Characteristics of percentile-based subsets

We filter the dataset by the percentiles from Table 5 to create the subsets of weak (P10) and strong (P90) convective organisation. Both subsets include 84,132 samples. Our analysis reveals that the frequency and location of convective clouds — and their strength of organisation — are not evenly distributed spatially or temporal. The majority of cloud tracks was detected between 10° S and 20° N (Figure 2). However, we observe distinct temporal and land–sea contrasts reflected in both P90 and P10. During March–May (MAM), strong convective organisation (P90) is more prevalent over land in the southern hemisphere and over ocean regions between 10°–30° S and 5° S–10° N. From June–August (JJA), P90 occurrences shift northward, peaking over land between 10° S–5° N and over ocean between 5°–15° N. A persistent local minimum appears around 0°–5° N in both seasons (Figure 10, a, c). In contrast, weak convective organisation (P10) is rare north of 15° N in boreal spring and south of 15° S in JJA. In MAM, it is more frequent over land from 10° S–10° N and over ocean between 0°–10° N. In JJA, we see an overall northward shift of the distribution (Figure 10, b, d).

Comparing the surface types of all cloud tracks and both percentile subsets, we observe a higher proportion of clouds over the ocean than over land for all datasets. However, there are differences within the surface-type distribution for the organisation-based subsets: when comparing all three datasets (all cloud tracks, P90, P10), strong convective organisation occurs about 5–15 % more frequently over the ocean, whereas the proportion of cloud systems with a weak convective organisation is about 10–15 % higher over land (Figure 11, a). P10 clouds are generally associated with fewer cores and shorter lifetimes than both P90 and the full dataset. They may be associated to a higher proportion to single-core clouds (15 % higher than P90) and clouds with a lifetime between 0–3 h (30 % higher than P90). We observe more clouds from P90 with a cloud lifetime of more than 3 hours. However, the longest lifetimes in the dataset may be found for clouds not connected to the percentile subsets (Figure 11, b, d). Clouds were detected slightly more frequently in MAM than JJA. In March, the proportion is especially high for P10 (15 % higher than P90). In contrast, occurrences of P90 are less common in MAM and increase in JJA (10 % higher than P10). These findings may indicate an increase of strong convective organisation during boreal summer (Figure 11, c).

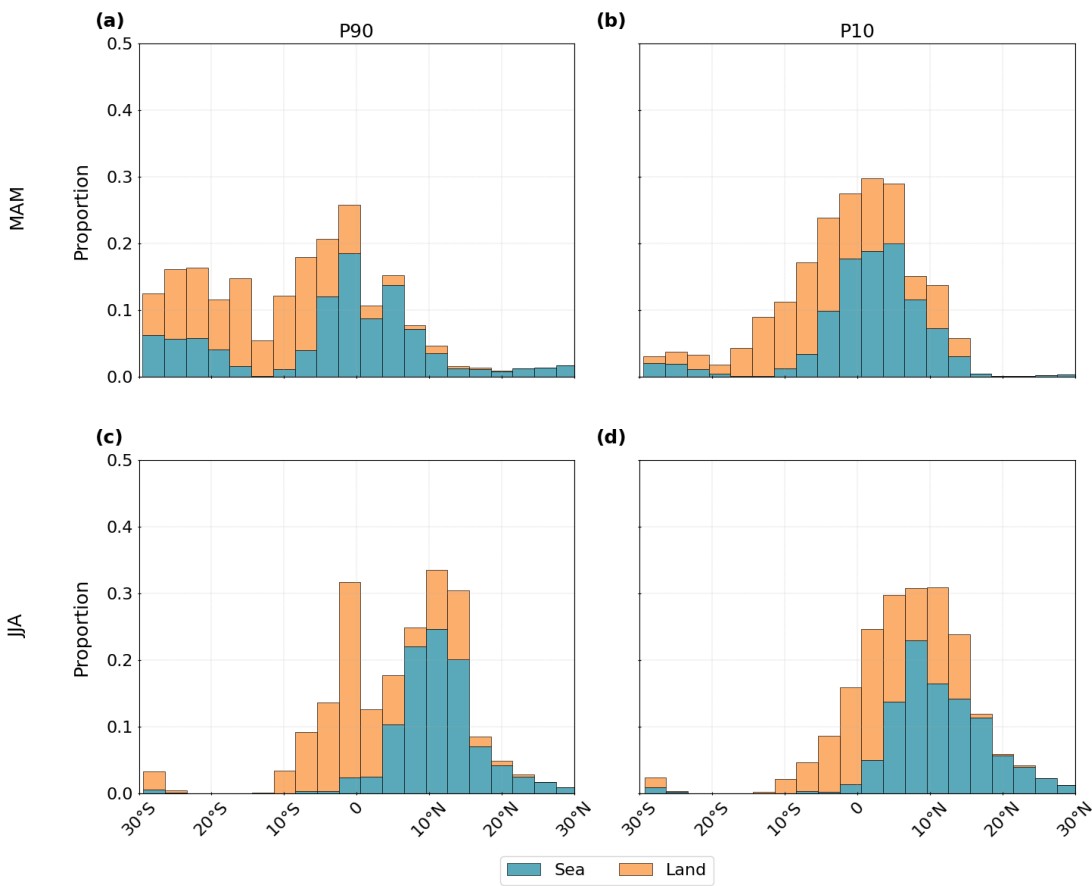

**Figure 10.** Distribution of detected clouds grouped in 3° intervals between 30° S and 30° N. The histograms show the proportions of cloud tracks grouped by the surface type (land, sea) for **(a)** the 10 % strongest convective organisation (P90, n = 84,132) and **(b)** the 10 % weakest convective organisation (P10, n = 84,132) between March–May (MAM) and June–August (JJA).

### 4.4.2   Relationship between organisation subsets and cloud properties

To explore how the relationship between cloud and core properties differs for weak (P10) and strong (P90) convective organisation, we compare the correlation coefficients between all cloud tracks and the two subsets. As noted in Sect. 4.2, SCAI tends

to correlate negatively with cloud properties, while COP and ROME show positive associations. For all cloud tracks, correlations between the indices range from -0.08 to 0.26 (Figure 7). Figure 12 highlights that inter-index and intra-cloud property correlations are stronger than those between indices and cloud properties. Here, COP and ROME exhibit moderate to strong positive correlation, while COP and SCAI are moderately negatively correlated (Figure 12, a). SCAI and ROME show a weak to moderate inverse relationship. Among cloud and core properties, the strongest positive correlation is between cloud area and number of cores, followed by CTH and core height. The number of cores, core area, and core height are also moderately

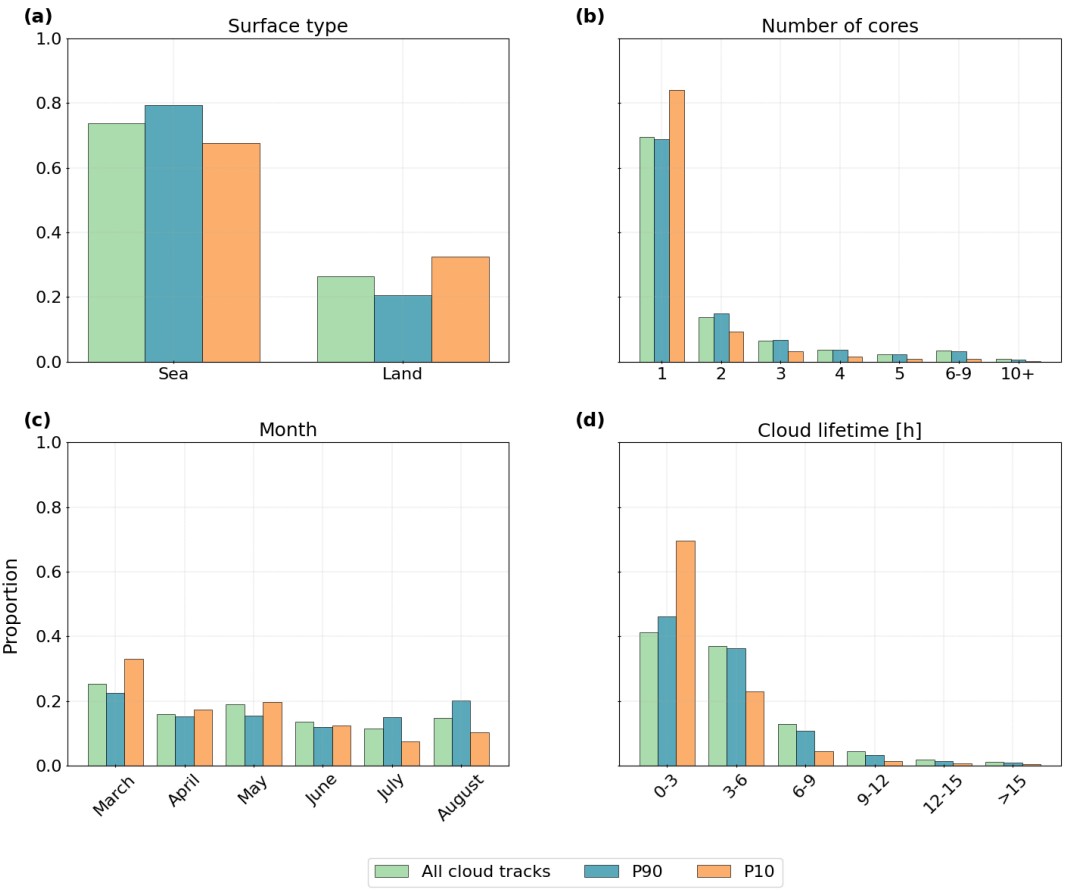

**Figure 11.** Comparison of the distributions between all cloud tracks, the 10 % strongest convective organisation (P90), and the 10 % weakest convective organisation (P10) for **(a)** the surface type derived from a land-sea mask, **(b)** the number of cores, **(c)** the month, and **(d)** the cloud lifetime after first detection.

correlated. Cloud lifetime, however, shows only weak to moderate associations with these properties. In the P90 subset, all three indices are positively correlated - a departure from the expected negative SCAI–COP/ROME relationship seen for all cloud tracks. Correlations between cloud and core properties in P90 remain largely similar to the full dataset, though some relationships (e.g., between COP/ROME and core height or area) strengthen slightly (Figure 12, b). In P10, we find similar property-to-property correlations, though the strength varies more. The strongest correlation remains between the number of cores and core area and the core and cloud height (Figure 12, c). For all data, we remain to find the strongest correlation between SCAI for the indices and CTH for the cloud/core properties. Uniquely, SCAI and ROME show a high positive correlation in both P10 and P90, despite being theoretically opposed in their interpretation of convective organisation (Section 3, Section 4.1). This apparent contradiction underscores the complexity of the indices, particularly when filtered by percentiles.

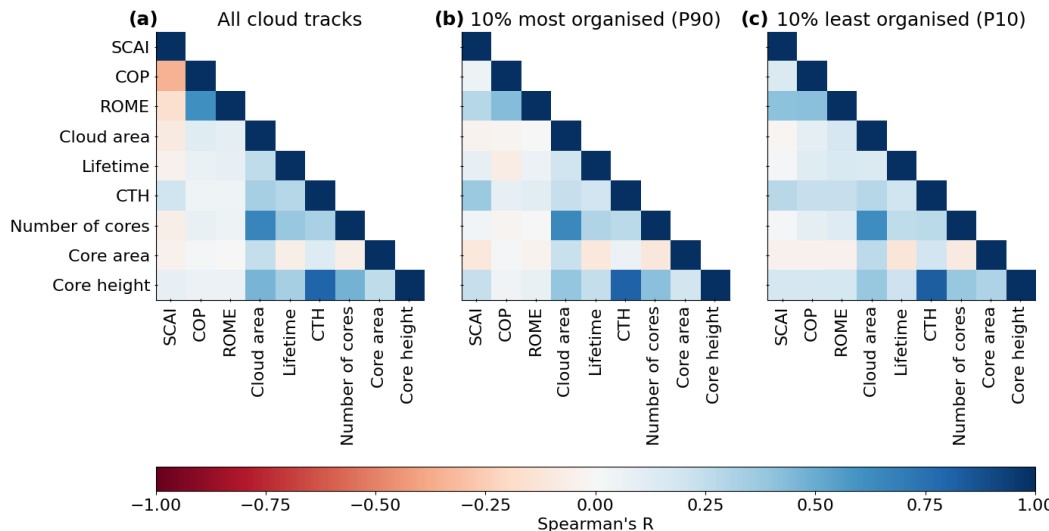

**Figure 12.** Correlation matrix for the organisation indices, cloud and core properties. We calculate Spearman's R to quantify the correlation coefficient on a scale between -1 and 1 for **(a)** the whole dataset of cloud tracks (n = 354,073), **(b)** the 10 % most organised clouds (P90, n = 84,132), and **(c)** the 10 % least organised clouds (P10, n = 84,132).

To assess whether differences between datasets are statistically significant, we compare parameter distributions for all cloud tracks, P90, and P10 subsets. We apply Welch's t-test, which may be more robust for unequal sample sizes (Derrick and White, 2016). For instance, we complement this with Cohen's D to estimate the effect size as small (< 0.2), medium (0.2–0.5), or large (higher than 0.8) (Cohen, 2013). The organisation indices show statistically significant differences across the three subsets, with large effect sizes for SCAI, COP, and ROME. Here, the effect size is largest between all data and P10 (Figure 13, a–c). Cloud and core properties exhibit more nuanced differences. The cloud area shows the largest effect size between all data and P10, while the CTH shows higher differences between all data and P90 (Figure 13, d, e). Compared to all cloud tracks, P90 clouds tend to be larger, with lower CTH, slightly shorter lifetimes, and slightly less, larger, and lower cores. P10 clouds are smaller, with a higher CTH, shorter lifetimes, fewer cores, and a larger core area and lower core height than clouds in the full dataset (Figure 13, d–i). For the number of cores, we find very low differences between all data and P90. As seen in Fig. 2, single-core clouds dominate the dataset. This skewness may affect statistics - in particular of data in P10 - which are heavily weighted toward fewer cores. Core area is larger in P90 and P10, whereas core height is lower in P90 and P10. However, for the core area, we observe only very small differences between the subsets (Figure 13, f–i).

While we detect statistically significant differences between percentile-based subsets and the dataset with all cloud tracks, the effect sizes for cloud and core properties remain mostly small to moderate. Our results indicate that strong convective organisation (low SCAI, high COP and ROME) tends to co-occur with larger cloud and core areas, slightly less and lower

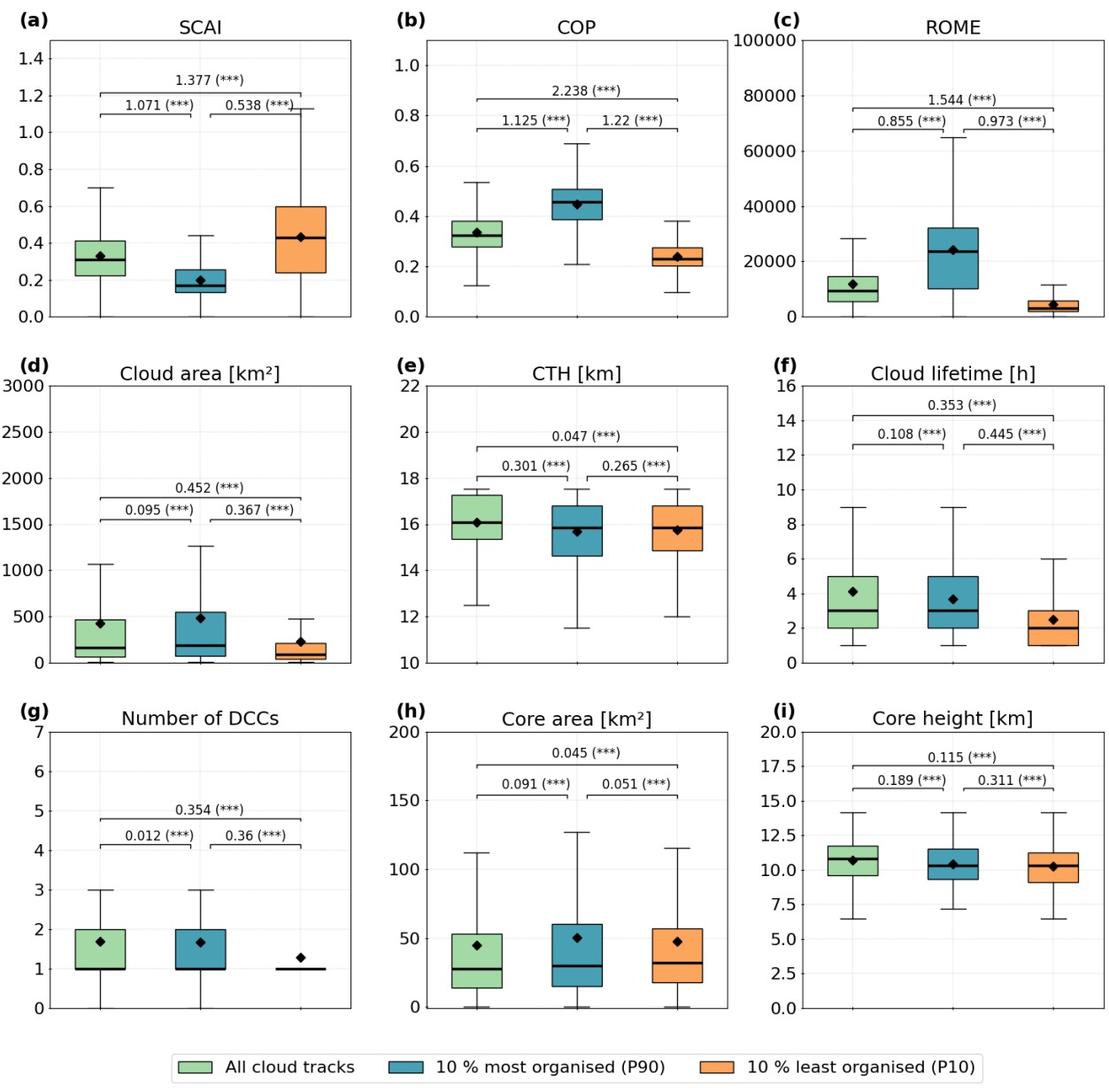

**Figure 13.** Boxplot showing the distribution of the convective organistaion indices, cloud and core properties for the whole dataset (n = 354,073), the 10 % most organised clouds (P90, n = 84,132), and the 10 % least organised clouds (P10, n = 84,132). We show the distribution of **(a)** the SCAI, **(b)** the COP, **(c)** the ROME, **(d)** the cloud area, **(e)** the CTH, **(f)** the cloud lifetime, **(g)** the number of cores, **(h)** the convective core area, and **(i)** the core height. The boxplot contains the median (bold black lines) and the arithmetic mean (black diamonds). Annotations depict the effect size measured by Cohen's D and the p-value derived from Welch's t-test (not significant (ns): p higher than 0.1, significant with *: $p < 0.1$, **: $p < 0.05$, ***: $p < 0.01$).

**Table 6.** Summary of differences between cloud and core properties for all cloud tracks against the percentile-based classification of weak (P10) and strong (P90) convective organisation. The table contains the arithmetic mean of all properties for the three datasets. We show in which direction the subset mean differs from all tracks (Direction) and the effect strength (Cohen's D) for all tracks compared to P90 or P10.

| | All tracks | | P90 | | | P10 | |
| | Arithmetic mean | Direction | Arithmetic mean | Cohen's D | Direction | Arithmetic mean | Cohen's D |
|---|---|---|---|---|---|---|---|
| SCAI | 0.331 | - | 0.198 | large | + | 0.433 | large |
| COP | 0.334 | + | 0.449 | large | - | 0.240 | large |
| ROME | 11858.977 | + | 24274.897 | large | - | 4413.081 | large |
| Cloud area | 421.225 | + | 485.431 | small | - | 223.537 | medium |
| CTH | 16.077 | - | 15.696 | medium | - | 15.755 | small |
| Cloud lifetime | 4.119 | - | 3.691 | small | - | 2.501 | medium |
| Number of cores | 1.774 | - | 1.740 | small | - | 1.305 | medium |
| Core area | 44.976 | + | 50.074 | small | + | 47.636 | small |
| Core height | 10.704 | - | 10.428 | small | - | 10.266 | small |

cores, and slightly shorter lifetimes. The highest effect sizes may be found for the CTH, core height, and cloud lifetime. Weak organisation (high SCAI, low COP and ROME) is associated with smaller clouds, lower CTH, fewer cores, a smaller core area, lower core height, and shorter lifetimes. Here, we observe the highest effect sizes for the cloud area, number of cores, and cloud lifetime (Table 6). These findings - and the differences between the two percentile-based subsets - suggest that different

aspects of cloud and core morphology may contribute to the strength of convective organisation.

### 4.4.3   Spatial distribution of percentiles

To identify how convective organisation may be spatially distributed for each of the three organisation indices (SCAI, COP, ROME), we filter the dataset of all cloud tracks by the percentiles (Q10, Q90) and we map the frequency of cloud occurrences across the area of interest (AOI) between 30° N–30° S and 30° W – 30° E. The data is interpolated using a 3° × 3° grid and

smoothed with a Gaussian filter (sigma = 0.5). In addition to the percentiles Q10 and Q90, we visualise the interquartile range (IQR; 25th–75th percentile) for each organisation index. Frequency values (0–140 per grid cell) are colour-coded to represent absolute counts.

    As shown in Sects. 4.1 and 4.2, high SCAI values - indicating weak convective organisation - are typically concentrated near the equator. In MAM, low SCAI values (Q10) occur over the equatorial Atlantic Ocean and land/sea areas south of 15°

S. High values (Q90) appear over equatorial Africa (0°–15° N), especially in rainforest zones, and Cameroon. The IQR peaks near the equator, particularly over the Ivory Coast, Guinea, Benin, Angola's coast, and Lake Victoria (Figure 14, a–c). In boreal summer, values shift north to 0°–15° N, with SCAI Q10 regions over the Atlantic and coastal West Africa. High SCAI (Q90) values occur in MAM over the Congo and Central African Republic. The IQR also shifts north in JJA, with hotspots over the

West African plains, Jos Plateau, and Congo River basin (Figure 15, a–c). COP exhibits weaker spatial variability than SCAI or
ROME. We detect clusters of low (Q10) COP near the equator in both seasons, over the Atlantic in MAM and across continental
Africa in JJA. For high values (Q90) in MAM, strong peaks are found along West and Central African coasts and offshore in
the Atlantic Ocean - many overlapping with regions of low COP, suggesting coexisting weak and strong organisation (Figure
14, d–f). In boreal summer, peaks of high COP (Q90) are concentrated over the Atlantic Ocean near Cape Verde and coastal
zones between Senegal and Sierra Leone. Secondary peaks appear inland across West Africa (Figure 15, d–f). The IQR aligns
closely around the equator but shifts northward in JJA, with dominant peaks over Central Africa's rainforest and minor peaks
across the West African plains (Figures 14 b, e, 15, b, e). ROME shows greater latitudinal variability than SCAI and COP. In
MAM, low values (Q10) values focus primarily along 15°–30° W near the equator, and secondarily between 15° S–15° N.
High values (Q90) values are concentrated along the West African coast and between Cameroon and Gabon. IQR peaks are
dispersed over the equatorial rainforest and coastlines (Figure 14, g–i). In JJA, low ROME (Q10) clusters around the Congo
River and more diffusely across continental Africa. Peaks for high values of ROME (Q90) appear over the Jos Plateau, Congo
River, and Atlantic. Like COP, ROME shows overlapping regions of weak and strong organisation over rainforests and oceans.
IQR values peak between 0°–15° N and extend to coastal West Africa (Figure 15, g–i).

The spatial patterns of COP and ROME are closely aligned, with the 10th and 90th percentiles showing often spatial overlaps.
SCAI has an inverse pattern due to its opposing index scale: regions with high COP/ROME may correspond to low SCAI (and
vice versa). This inverse relationship is evident throughout the period, with all three indices exhibit consistent spatial patterns.
The IQR maps, consistent across indices, reveal a northward shift of the indices which aligns with convective cloud occurrences
during boreal summer as depicted for the percentile-based subsets in Fig. 10.

### 4.4.4 Identifying hotspots of convective organisation

In contrast to the former analysis, we examine the spatial distribution for clouds in the two subsets (P90, P10) (Section 4).
These subsets of the 10 % strongest (P90) and the 10 % weakest (P10) convective organisation may help to identify cumulative
hotspot regions averaged over the three indices. The data may allow us to analyse spatial patterns and temporal changes of
convective organisation across two seasons from boreal spring (March to May, MAM) to summer (June to August, JJA). The
occurrences are interpolated onto a 3° × 3° grid between 30° N–30° S and 30° W–30° E and smoothed using a Gaussian filter
with a kernel size of 0.5.

In MAM, the highest proportion of strong convective organisation (P90) occurs over the Atlantic Ocean, with a notable
concentration near the equator and between 15° and 30° S. Additional hotspots are found along the West African coastlines,
the Gulf of Guinea. Moreover, we observe small peaks over the equatorial rainforest, Angola, and parts of the Sahel. Overall,
most of the data points for the 10 % strongest convective organisation during boreal spring are located south of the equator
(Figure 16, a). Weak convective organisation during MAM displays two primary clusters. The first is located over the equatorial
Atlantic Ocean, particularly between 15° and 30° W. The second spans continental Africa, where more dispersed peaks emerge
between Cameroon and the Congo River. Across the belt from 15° N to 15° S, the frequency of the 10 % weakest convective
organisation is generally high (Figure 16, b). In boreal summer, the spatial distribution of strong convective organisation shifts

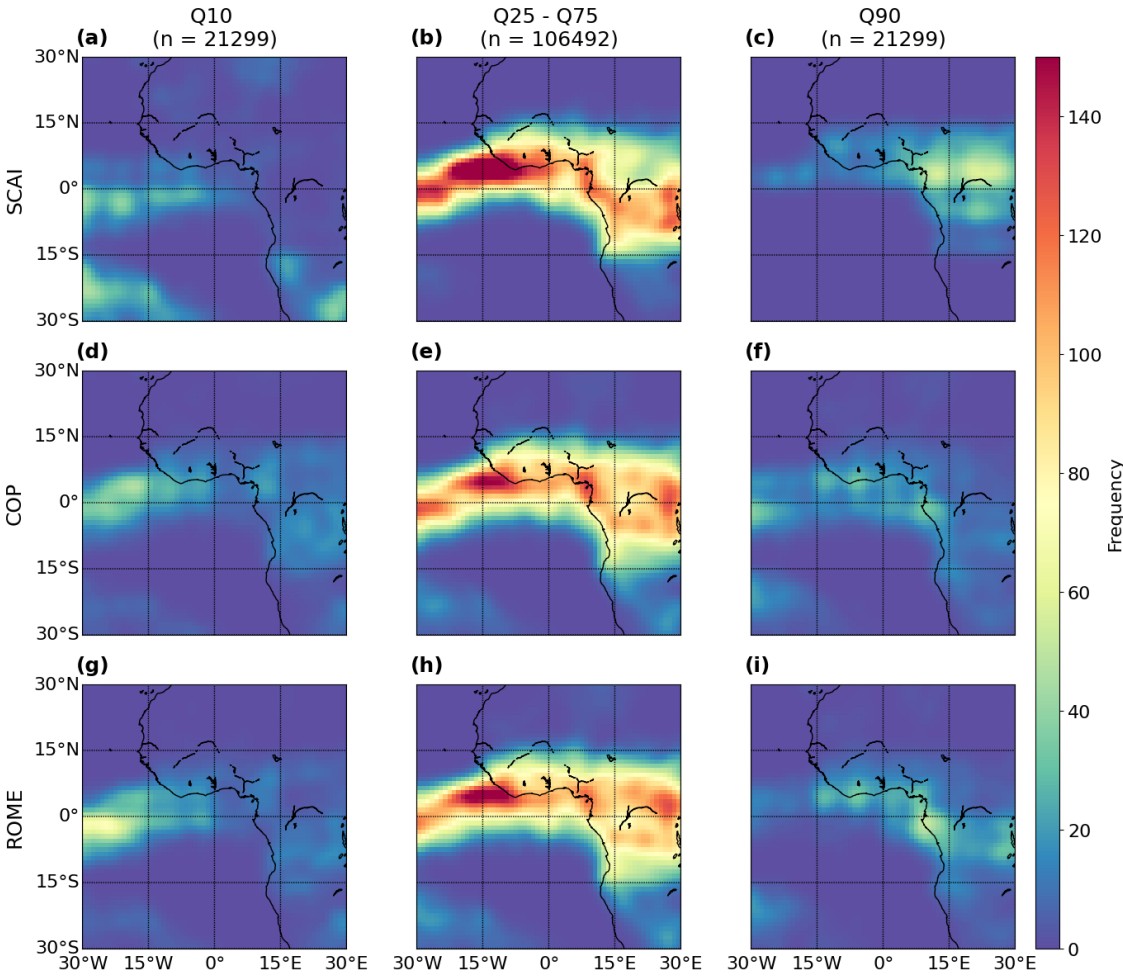

**Figure 14.** Spatial distribution of the percentiles Q10, Q25–Q75, and Q90 for the convective organisation indices **(a–c)** SCAI, **(d–f)** COP, and **(g–i)** ROME between March and May (MAM, n = 212,984). The values represent the frequency distribution interpolated on a 3° x 3° grid.

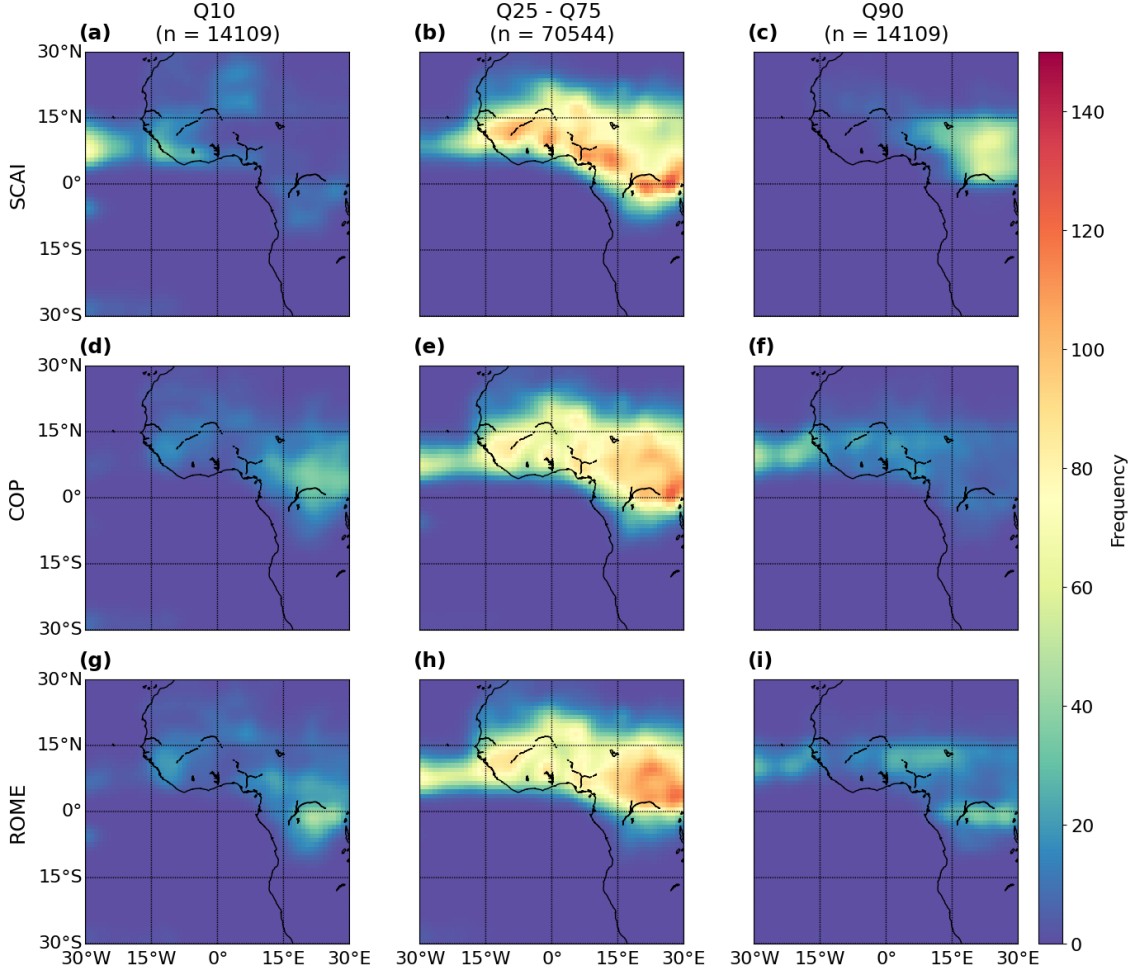

**Figure 15.** Spatial distribution of the percentiles Q10, Q25 - Q75, and Q90 for the convective organisation indices **(a–c)** SCAI, **(d–f)** COP, and **(g–i)** ROME between June and August (JJA, n = 141,089). The values represent the frequency distribution interpolated on a 3° x 3° grid.

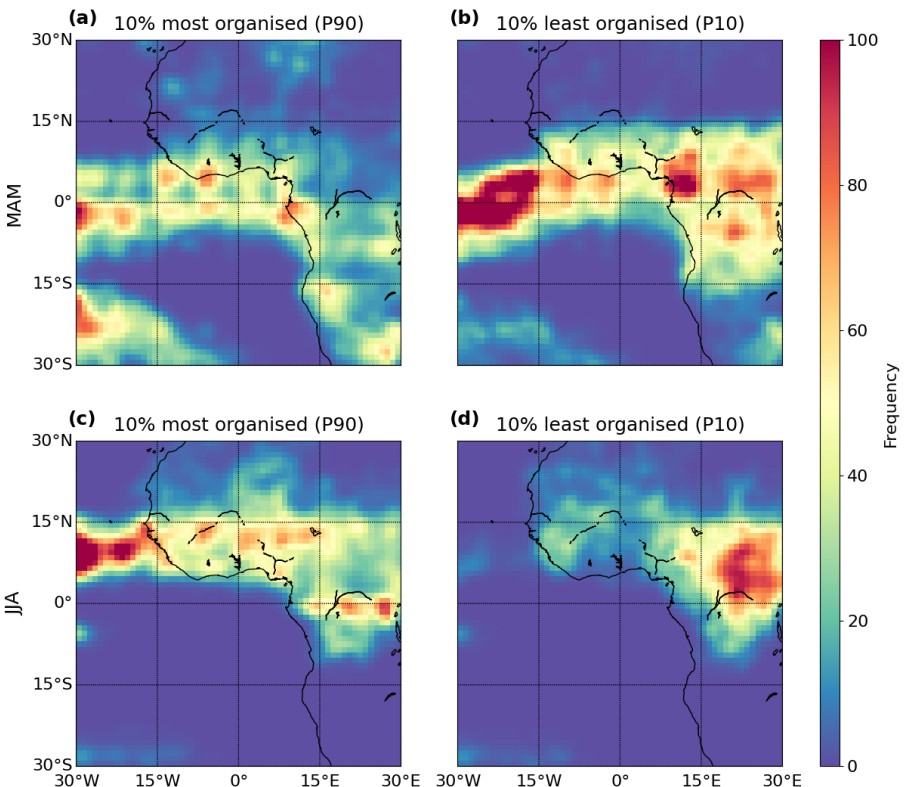

**Figure 16.** Spatial distribution of convective organisation based on an aggregation of the percentiles for SCAI, COP, and ROME in boreal spring (MAM, upper row) and summer (JJA, lower row). Clouds are grouped as the **(a) & (c)** strong organised or **(b) & (d)** weak organised using Q10 lowest (SCAI) or Q90 (COP, ROME). The values represent the frequency distribution interpolated on a 3° x 3° grid.

northward. Regions with a frequent occurrence of strong convective organisation emerge over the Atlantic Ocean and become more widespread across the West African plains, including areas around the Niger and Congo rivers (Figure 16, c). Weak organisation, on the other hand, is concentrated primarily over continental Africa, especially between 15° and 30° E, with a peak located just north of the Congo River (Figure 16, d).

As suggested in Sect. 4.4.3, we observe overlapping regions of weak and strong convective organisation throughout the period. In MAM, this overlap is evident over both ocean and land, whereas in JJA, it is mainly confined to continental Africa. Overall, cloud occurrences and spatial patterns suggest a shift between MAM and JJA which is consistent to the imbalance between cloud tracks over land and ocean observed in Fig. 11. In boreal spring, strong convective organisation is more frequently observed over the ocean, while weak organisation is distributed across both land and sea. By boreal summer, strong organisation becomes more prominent over land, and weak organisation is largely confined to the African continent. This migration of convective hotspots appears consistent with the northward movement of the ITCZ, as described in Atiah et al. (2023).

# 5 Discussion

## 5.1 Summary of key findings

Our analysis reveals that convective cloud occurrence and convective organisation vary considerably across both space and time. While the study spans only six months and does not provide a full climatology, the results highlight spatial and temporal changes of convective organisation during the period. Notably, the frequency of the 10 % strongest convective organisation increases during the boreal summer months (June to August, JJA), particularly north of the equator (Figure 11, Figure 16, c–d). In boreal spring (March to May, MAM), it may occur more frequently south of the equator (Figure 16, a–b). Between March and May, we observe a higher concentration of strongly organised convection over the Atlantic Ocean, primarily between 0° and 30° S. This peak shifts northward to between 0° and 15° N in JJA, with additional hotspots appearing over the equatorial rainforest and West Africa. Meanwhile, weakly organised convection tends to dominate over the Atlantic Ocean in boreal spring and shifts to continental Africa in JJA. These findings suggest a broader northward movement of convective cloud occurrences throughout the study period (Section 4.2, Section 4.4).

Correlations between convective organisation indices (SCAI, COP, ROME) and cloud/core properties suggest generally weak to medium relationships for all cloud tracks. SCAI is negatively correlated to the properties, except for the CTH and core height, while COP and ROME show positive correlations except for the core area. In all cases, the coefficients remain below 0.3 (Section 4.2). These correlations partly change for the 10 % strongest and 10 % weakest convective organisation. Within these subsets, we find the highest corelation coefficient between SCAI and the CTH for clouds with a strong convective organisation. However, the relationships between the cloud and core properties remain similar over all subsets (Section 4.4.2). In contrast, we observe pronounced changes in both the indices and the associated cloud characteristics along the period and across the AOI. These changes reflect a high variability for average values, though the correlation strength remains limited (Section 4.3). The distribution of cloud and core properties within identified hotspot regions differ from those observed in the full dataset of all cloud tracks. We analyse the effect size using Cohen's D to reveal how organisation strength may influence cloud characteristics. Compared to all cloud tracks, the cloud systems of the 10 % strongest organisation tend to have larger cloud and core areas, a lower CTH and core height, a shorter lifetime, and a lower number of convective cores. In contrast, weaker convective organisation may be typically associated with smaller clouds and larger cores, fewer cores, shorter lifetimes, and lower vertical extent. Strong convective organisation differs the most from all cloud tracks regarding the CTH and from weak organisation regarding the cloud lifetime. Between weak convective organisation and all cloud tracks, we identify the cloud area to have the highest effect size (Section 4.4.2). Hence, the cloud area appears to be more important to identify weak convective organisation, whereas strong convective organistaion may be stronger driven by the CTH. Despite these differences in the distribution of cloud and core properties, we detect partly the same direction for correlations in case of strong and weak convective organisation, highlighting the complexity of involved processes.

## 5.2 Spatio-temporal drivers of organisation

Our results show that convective organisation tends to be stronger for cloud properties typically associated with large convective systems containing multiple core regions, such as MCSs (Stubenrauch et al., 2023). In line with Brüning and Tost (2025), we observe that cloud area, lifetime, cloud top height (CTH), core area, and core height all grow with the number of convective cores (Figure 12). While multiple cores may enhance cloud longevity, promote cloud area growth, and strengthen vertical updrafts, the number of cores may also be a key factor in determining the strength of convective organisation. Interestingly, our findings contrast with Takahashi et al. (2017), as we observe stronger convective organisation - reflected in higher COP and ROME values and lower SCAI values - more frequently over the ocean. Over continental Africa, spatial patches of weak convective organisation appear in both seasons (Section 4.2, Section 4.4.4). However, the difference between land and ocean remain small and may partly stem from an uneven distribution of cloud tracks (Figure 2).

Spatial patterns of convective hotspots show differences over land and ocean. Around the equator, we observe a great share of cloud systems with a weak and strong organisation. This spatial overlap occurs between March–May, in particular over the ocean, and between June–August, especially over continental Africa (Section 4.4.4). Overall, the distribution of convective organisation varies notably between hemispheres and between equatorial and tropical zones (Section 4.3). These differences may be driven by a mix of local surface features (Vondou, 2012), monsoonal dynamics (Futyan and Genio, 2007), and topographic influences such as katabatic flows (Nicholson, 2018). Although our study reveals distinct geographical patterns, isolating the role of topography will require more targeted analysis.

We also detect a link between convective core occurrence and organisation that may follow the northward migration of the ITCZ in boreal summer. As the ITCZ shifts, it may alter regional circulation, surface energy balance, and moisture availability — particularly influencing cloud development over the northern Sahel and southern Sahara, as observed by, e.g., the spatial distribution of SCAI between June and August (Section 4.4.3). These changes may be associated with increased humidity, reduced subtropical subsidence, and deeper ascent within the tropical rainbelt (Fontaine and Philippon, 2000). Together with strengthened meridional pressure gradients (Lavaysse et al., 2009), they may contribute to the occurrence of large convective systems with multiple cores. This observation may be reflected in our results as a northward displacement of convective clouds and an increase in cloud area, core area, and core number over continental Africa in July and August (Section 4.2, Section 4.3). While our findings highlight the variability of convective organisation, the limited six-month time frame prevents a climatological interpretation. Extending this analysis across multiple years may provide deeper insights into the annual cycle of convective organisation and help refine operational forecasting and early-warning systems (Pendergrass, 2020).

## 5.3 Uncertainties and limitations

Our analysis may offer additional insights into the spatio-temporal distribution of convective organisation in the tropics. However, overall statistical relationships between convective organisation indices and cloud properties remain weak as we observe mostly small to medium effect sizes and low to moderate correlation coefficients (Section 4.2, Section 4.4.2). They highlight the complexity of quantifying convective organisation across space and time. Although our study may help to map patterns

of convective organisation across the AOI, gaining a deeper understanding of the underlying processes may require incorporating additional cloud parameters — such as cloud radiative properties - or associated precipitation rates (e.g., Stauffer and Wing (2024); Stubenrauch et al. (2023)). Moreover, addressing the imbalance between land and ocean cloud occurrences could strengthen the robustness of our findings. Currently, the cloud track distribution is skewed, with a heavy concentration near the equator. Notably, all indices indicate overlapping occurrences of both weak and strong organisation within the same regions — particularly over the Atlantic Ocean and continental Africa in boreal spring, and over the Congo River basin in summer. These spatial overlaps may obscure clearer statistical signals (Section 4.4).

Our dataset describes the three-dimensional structure of the tracked clouds, which may enable segmentation of cloud and core regions across horizontal and vertical dimensions at each point in time. Still, it is constrained by the performance of the ML model and the underlying tracking algorithm. Based on evaluations from Brüning et al. (2024) and Brüning and Tost (2025), the ML-predicted radar reflectivities exhibit a mean error of 2.99 dBZ. While suitable for building contiguous 3D cloud fields, the predictions struggle to accurately represent shallow cumulus and cirrus clouds — limitations inherited from the CloudSat CPR (Sassen and Wang, 2008). Incorporating higher-resolution satellite data or ground-based radar could enhance prediction accuracy. Other sources of uncertainty include the chosen thresholds for the detection algorithm and the skewed distributions underlying our percentile-based classifications of convective organisation. The indices themselves are sensitive to cloud object count (SCAI) or area (COP, ROME), which may affect spatial patterns, especially since equatorial convective clouds tend to be smaller and more frequent than those near the tropics (Section 2.5, Section 3.1). Additional uncertainties involve the influence of the terrain on cloud organisation (Biagioli and Tompkins, 2023). Future research could benefit from using combined indices or integrating temporal and spatial factors into a unified metric for 3D data. Our current method uses a moving-window, grid-based approach (Section 3.2), differing from past studies that partitioned the AOI into equal-area subsets (e.g., Tobin et al. (2012); Stubenrauch et al. (2023); Retsch et al. (2020)). While a moving window may reduce noise from small-scale fluctuations, its kernel size is manually chosen. To address this, we plan to explore unsupervised clustering techniques such as the Density Based Spatial Clustering of Applications with Noise (DBSCAN) (Ester et al., 1996) or the extended Hierarchical Density-Based Spatial Clustering of Applications with Noise (HDBSCAN) (Campello et al., 2013) as a more data-driven alternative. Zuo et al. (2022) successfully applied DBSCAN to identify cloud clusters in 3D radar data, while Kim et al. (2023) used the approach to derive precipitation probabilities from geostationary satellites. In future work, we aim to test whether such algorithms may reliably quantify convective organisation across space and time.

## 6   Conclusions

This study explores the spatial and temporal patterns of convective organisation in tropical West Africa using ML-based 3D radar reflectivities. We focus on the relationship between convective organisation, cloud structure, and core properties, using three organisation indices to statistically identify regional hotspots through a percentile-based classification.

Our analysis reveals that convective organisation tends to be slightly stronger over the ocean. However, differences between the indices over different surface types and along the period remain low and average around 10–15 %. We observe a consid-

erable spatial variability and a temporal shift in the distribution of strong convective organisation which appears linked to the northward migration of the ITCZ. From March to August, COP and ROME values increase while SCAI decreases, especially in the northern hemisphere, indicating an enhanced spatial clustering of convective clouds. Our regional analysis shows that the most cloud systems with a strong convective organisation during boreal spring (March–May) are concentrated over the Atlantic Ocean and coastal West Africa, predominantly in the southern hemisphere. In boreal summer (June–August), these hotspots shift inland toward the equatorial rainforest, West African Plains, and Sahel region. Notably, both weak and strong convective organisation frequently co-occur in the same regions, complicating statistical interpretation and underscoring the complexity of convective systems.

While correlations between organisation indices and cloud or core properties are generally weak to moderate, we observe that clouds with the 10 % strongest convective organisation tend to have larger cloud areas, lower cloud top and core heights, and more less but larger cores than the average cloud trajectory. In contrast, the 10 % weakest convective organisation are associated with smaller cloud areas, shorter lifetimes, fewer but larger cores, and a lower cloud and core height. Differences in CTH appear to be the most important for identifying cases of strong convective organisation. In contrast, the cloud area, cloud lifetime and number of cores appear to be a driver of weak convective organisation. Despite these findings, observed relationships and spatial patterns vary notably across the indices. The indices themselves often yield opposing results, reflecting their individual sensitivities and limitations. This variability is further influenced by the characteristics of the ML-based dataset. As the current study relies on 2D indices, developing a 3D organisation metric could provide a more accurate and holistic view. In summary, our findings highlight substantial variability in convective organisation across time and space. Given its influence on extreme weather, understanding these variations - and the mechanisms behind them - is crucial for improving climate risk assessments and forecasting capabilities in West Africa and beyond.

*Code and data availability.* The level 2B-GEOPROF CloudSat data used in this study are available at the CloudSat Data Processing Center at CIRA/Colorado State University and can be retrieved from http://www.CloudSat.cira.colostate.edu/order-data (CloudSat Data Processing Center, 2024). The Meteosat SEVIRI level 1.5 data used in this study is freely and openly available via the EUMETSAT Data Store at https://navigator.eumetsat.int/product/EO-:EUM:DAT:MSG:HRSEVIRI (EUMETSAT Data Services, 2024). The dataset of convective cloud tracks and organisation indices used in this study is available at the following repository: https://zenodo.org/records/14724869 (Brüning, 2025b). The material used to prepare this paper, including code used to perform analysis and that needed for the preparation of figures, is archived at https://zenodo.org/records/15607483 (Brüning, 2025a).

*Author contributions.* S.B and H.T. designed the study. S.B developed the code for performing the analysis and visualisation. S.B. and H.T. contributed to analysing and evaluating spatio-temporal patterns of convective cloud organisation. S.B. and H.T. wrote the draft of the paper. All authors have read and agreed to the published version of the manuscript.

*Competing interests.* The authors declare that they have no conflict of interest.

*Acknowledgements.* This work was supported by the project "Big Data in Atmospheric Physics (BINARY)", funded by the Carl Zeiss Foundation (grant P2018-02-003), and the Max Planck Graduate Center with the Johannes Gutenberg University of Mainz (MPGC). We thank EUMETSAT for providing access to the Meteosat SEVIRI imager data and the Cooperative Institute for Research in the Atmosphere, CSU, for providing access to the CloudSat 2B-GEOPROF data.

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
