# Peer review of "A machine learning-based perspective on deep convective clouds and their organisation in 3D. Part II: Spatial-temporal patterns of convective organisation"

_EGUsphere, 2025_

## Referee Comment (RC1)

Review of "A machine learning-based perspective on deep convective clouds and their organisation in 3D. Part II: Spatial-temporal patterns of convective organization"
By Sarah Brüning and Holger Tost
MS No.: egusphere-2025-376

**Recommendation**

Minor revisions

**General comments**

This study presents the spatial and temporal patterns of convective organization based on a three-dimensional cloud field constructed using a machine learning technique that combines data from horizontal distribution from geostationary satellite MeteoSat-11 and vertical profiles from Cloud-Sat Cloud Radar. The area of interest (AOI) spans from 30°S to 30°N and 30°W to 30°E, where parameters such as cloud area, cloud top height, cloud lifetime, the number of deep convective clouds, core size, and the area ratio between the core and the anvil are analyzed.

Detailed results illustrate seasonal and geographic variations in organization indices and related quantities. The newly constructed dataset provides unique insights into convective organization within the AOI, making it a valuable resource for understanding these characteristics. Therefore, the paper could be published only after clarifying the following results.

The reviewer's main request is to clarify the description of the results. For example, the term "convective activity" is used without clearly specifying its characteristics, leading to confusion rather than clarification. Although the manuscript presents various findings on geographic distributions and seasonal evolution, it would benefit to present a summarizing analysis that highlights the key conclusions. In particular, the differences in convective organization between sea and land, as well as between the Northern and Southern Hemispheres (Figs. 14 and A1), remain unclear and should be clarified.

Moreover, many of the key results are not exclusively derived from the unique three-dimensional dataset with continuous object tracking. Ideally, the analysis should emphasize aspects uniquely obtainable from this dataset. It was disappointing that the authors chose indices of convective organization that could be derived from 2D imagery alone. The authors should clearly summarize the

advantages of their unique dataset and highlight how it advances our understanding of convective organization.

The characteristics of the dataset, especially the 3D objects obtained through the machine learning method, should be described more clearly in this paper, even if detailed explanations are provided in Part I. In particular, it remains unclear whether the identified 3D objects are smoothly connected over time.

**Specific comments**

L26, "convective organisation (or aggregation)": The authors should distinguish "convective organization" and "convective aggregation" by giving their definitions.

L30, "The spatial distribution of convective clouds is not arbitrary.": This sentence is unclear.

L37, "several regions": It is unclear. What types of "regions" are meant in this context?

L44, "So far, the models show convective organisation increases with a warming climate": Wing et al. (2020) also showed a change in convective organization with warming in RCEMIP.

L58-60, "At the same time, food security and a high climate risk exposeWest Africa to multiple threats (Berthou et al., 2019). Changing atmospheric conditions could intensify those hazards.": We know that these points are important but not specifically related to the current research. These sentences should be moved to the final section, instead of the introduction, or removed.

L104-105, "It is characterised by lower temperatures and a strong vertical ascent, which we identify by an extensive vertically contiguous 105 layer and a high radar reflectivity (e.g., Igel et al. (2014); Takahashi et al. (2017)).": Vertical ascent is not directly analyzed by the proposed method, neither in Igel et al. (2014) nor in Takahashi et al. (2017). This sentence should be modified.

L173-174: Please describe the methodology of the moving windows in more detail. What types of iterations are applied to calculate the indices?

L179, "the frequency distribution shows an overlap of lower index values for all indices": This sentence is unclear. What does this mean by an overlap?

L230-232: What is the meaning of "a diverging convective activity"? In Fig. 7a, we cannot see where the maximum cloud area is. Where is "a lower cloud lifetime" in Fig. 7c?

L244, 245: What does "convective activity" mean in the sentences, and which figure shows this? Which figure and location show "a higher convective activity comes with a lower area ratio, a higher number of DCCs, a larger cloud and core area"?

L254, "the convective organisation is overall weaker around the equator": In 4.1, "convective organization" is not clearly defined and is not specifically described. The definition of organization must be clarified.

L255, "Their impact on large-scale patterns of organisation is limited compared to MCSs": The meaning of this sentence is unclear.

L263, "(Figure 9, a-b,g-h,i-j)": Fig.9-j does not exist.

L264: "Figure 7" should be "Figure 6". Figure 7 does not show seasonal distributions.

L274: What does "convective activity" indicate here?

L274-279: In Figure 7, which index is used to define a most or least organized group?

L303-304, "Overall, organised clouds (P90) come along a larger cloud anvil area, a longer cloud lifetime, a lower CTH, a lower area ratio, and more and larger DCCs": It is unclear from Figure 13 to see these relationships. The relationships between the indices and these properties should be directly compared. A part of the comparison between the indices and the number of DCC is shown in Fig. 4g-i.

L320-336: The authors describe notable characteristics which can be seen from Fig. 14. However, some points are not convincing from the figure. Please check whether the description is consistent with the figures. For example, in L328-330, I

cannot see a noticeable narrowing in summer (JJA) for ROME. The differences in the effect size are significant according to the numbers in the Figure, but not clearly visible.

L350, L359-360, "the microphysical cloud properties": Cloud anvil area is a cloud macrophysical characteristic rather than a microphysic property. It is true that cloud microphysics affect cloud anvil area through the balance between sedimentation and the outflow, the present study does not examine microphysical cloud properties specifically.

L361-362, "For continental cloud clusters in the northern hemisphere, we find more distinct results regarding the relationship between DCCs and the degree of organization": This is one of the noticeable results discovered in this paper. However, this conclusion is indirectly shown by the figures in 4.1.2. The authors are suggested to show a more direct analysis showing the conclusion.

L468: Spell out "JAS".

---

## Author Comment (AC1)

**Final response**
**(Manuscript "EGUSPHERE-2025-376")**
**Reviewer 1**

Sarah Brüning and Holger Tost

May 23, 2025

**1 Introductory remarks**

We would like to thank the reviewer for taking your time to give us constructive and detailed feedback. In the following sections, we address all comments. For every comment, you can find (1) the comment, (2) the author's response (both with lines from initial manuscript), and (3) the author's changes in the revised manuscript. The lines given for **author's changes in the manuscript** refer to the lines in the **revised manuscript**.

Based on all referee comments, we focus the revision on improving the clarity of the written expression, in particular the description of our methods and the results. While the methods used in the manuscript did not change, we aim to focus on describing the data and results more directly. This includes, in particular, the distribution of organisation indices, their connection to cloud and core properties, and differences for the percentile-based subsets of organisation. As the period of six months may to too short to derive a distinct seasonality, we revise the description of hemisphere- or season-based subsets and replace the term "seasonal" changes with "monthly" changes which we observed during the period. Moreover, we changed the abbreviation "DCC" to "cores". Throughout the manuscript, we replace terms like "convective activity" with a more direct description of the cloud properties (cloud area, core number). We changed the figure labels in the following way:

- 1 -> 1

- 2 -> 3

- 3 -> 4

- 4 -> 5

- 5 -> Removed

- 6 -> Removed

- 7 -> 6

- 8 -> 9

- 9 -> 14

- 10 -> 15

- 11 -> 16

- 12 -> 8

- 13 -> Removed

- 14 -> 13

- New: 2 (Summary of cloud tracks: Latitude, Surface type, Month, Lifetime, Daytime of first detection, number of cores)

- New: 7 (2D histogram of average relationship between organisation indices and cloud/core properties)

- New: 10 (Summary of percentile-based subsets P90 and P10: Spatial distribution grouped by surface type (land,ocean) and month (March-May,June-August))

- New: 11 (Summary of percentile-based subsets P90 and P10: Surface type, number of cores, month, lifetime)

- New: 12 (Correlation matrix for organisation indices, cloud properties, and core properties for all cloud tracks, P90, and P10)

**2  General comments**

- **Comment**: The reviewer's main request is to clarify the description of the results. For example, the term "convective activity" is used without clearly specifying its characteristics, leading to confusion rather than clarification. Although the manuscript presents various findings on geographic distributions and seasonal evolution, it would benefit to present a summarizing analysis that highlights the key conclusions. In particular, the differences in convective organization between sea and land, as well as between the Northern and Southern Hemispheres (Figs. 14 and A1), remain unclear and should be clarified.
  **Author's response**: Thank you for your comment. We revised the manuscript focusing on improving the written written to clarify the description of results. In the original manuscript, we used the term "convective activity" to describe either the occurrence of convective clouds and their associated properties, such as the cloud area and number of cores; however, we revise the text to address observed characteristics more directly. A summarizing analysis highlighting the key conclusions will be added in Sect. 5.1. Based on the reviewer comments, we revise the analyses throughout the manuscript to focus more on a direct comparison of convective organisation indices and the cloud/core properties than a seasonal/hemispheric division (as the depicted period may only contain six months from one year of data).
  **Author's changes in the manuscript**: Please see Sect. 4 (Results) and Sect. 5.1 (Summary of key findings) of the revised manuscript.

- **Comment**: Moreover, many of the key results are not exclusively derived from the unique three-dimensional dataset with continuous object tracking. Ideally, the analysis should emphasize aspects uniquely obtainable from this dataset. It was disappointing that the authors chose indices of convective organization that could be derived from 2D imagery alone. The authors should clearly summarize the advantages of their unique dataset and highlight how it advances our understanding of convective organization.
  **Author's response**: All key results can be obtained from the cloud trajectories in ML-based 3D dataset. In the revised version, we add an extended description of the data and derived cloud and core properties to provide more clear information on how we retrieve the properties used for analysis (Sect. 2.4). We agree that using a organisation index adapted for 3D data may provide further insights. However, to our best knowledge, there exists no 3D index - unfortunately, designing such a novel index was out of scope for our study. In contrast, the aim of our study is rather to apply established indices and combine the information with cloud and core properties derived from our 3D dataset (e.g., core size, core height). Here, we see a major advantage of our approach, which is a simultaneous retrieval of horizontal (area) and vertical (height) cloud and core properties with a high spatial and temporal resolution, and a broad coverage over land and sea. Otherwise, receiving these data from 2D imagery or ground-based radar may be particularly challenging - moreover, compared to studies using only CloudSat, we may detect a considerably higher number of clouds. When applying the framework to a climatological time series, we may contribute to close data gaps in current research.
  **Author's changes in the manuscript**: We add a more detailed description of the satellite data, the machine-learning approach, the detection framework, and associated limitations in Sect. 2.1-2.5 (Data) and Sect. 5 (Discussion).

- **Comment**: The characteristics of the dataset, especially the 3D objects obtained through the machine learning method, should be described more clearly in this paper, even if detailed explanations are provided in Part I. In particular, it remains unclear whether the identified 3D objects are smoothly connected over time.

**Author's response**: We add a more detailed description (with revised visual overview) of the workflow for the machine learning method and the framework to detect convective clouds and cores in Sect. 2 of the revised manuscript. Here, we address characteristics of the satellite data, the ML algorithm, and the linking step in more detail: in Sect. 2.2, you may find that the predicted data has a temporal resolution of 15 minutes, between which the linking step smoothly connects the cloud objects.

**Author's changes in the manuscript**: Please see Sect. 2.1 (Satellite data), Sect. 2.2 (3D cloud field reconstruction), and 2.3 (Detection and tracking of convective clouds and cores) in the revised manuscript for a more detailed description of the dataset.

**3 Specific comments**

In the next section, we shortly address your specific comments. Changes will be added in the revised manuscript.

- **L26**: "convective organisation (or aggregation)": The authors should distinguish "convective organization" and "convective aggregation" by giving their definitions.
  **Author's response**: Thank your the comment, we add a more clear definition and revise the written expression in the manuscript.
  **Author's changes in the manuscript**: See, e.g, Sect. 1, lines 25-31: "Although the term convective organisation has become increasingly popular in climate research, it is often used vaguely. Mapes and Neale (2011) broadly summarise organisation as "non-randomness in meteorological fields in convecting regions". This definition induces a clustering of deep convective cells which is ubiquitous in the atmosphere, particularly in the tropics. However, the underlying mechanisms remain insufficiently understood (Muller and Bony, 2015). While convective organisation is difficult to quantify in observational data, idealised model configured in radiative-convective equilibrium (RCE) could demonstrate a large-scale clustering of convective clouds which is known as self-aggregation of convection (e.g, Held et al. (1993); Wing et al. (2017))."

- **L30**: "The spatial distribution of convective clouds is not arbitrary.": This sentence is unclear.
  **Author's response**: We agree and change the text in the revised manuscript.
  **Author's changes in the manuscript**: We removed the sentence in the revised manuscript.

- **L37**: "several regions": It is unclear. What types of "regions" are meant in this context?
  **Author's response**: The term "regions" points toward the convective cores within a MCS. In the revised manuscript, we change the text to be more clear.
  **Author's changes in the manuscript**: We removed the sentence in the revised manuscript.

- **L44**: "So far, the models show convective organisation increases with a warming climate": Wing et al. (2020) also showed a change in convective organization with warming in RCEMIP.
  **Author's response**: Thank you pointing this out, we add the reference pointing to RCEMIP studies.
  **Author's changes in the manuscript**: See above, lines 25-31: "Although the term convective organisation has become increasingly popular in climate research, it is often used vaguely. Mapes and Neale (2011) broadly summarise organisation as "non-randomness in meteorological fields in convecting regions". This definition induces a clustering of deep convective cells which is ubiquitous in the atmosphere, particularly in the tropics. However, the underlying mechanisms remain insufficiently understood (Muller and Bony, 2015). While convective organisation is difficult to quantify in observational data, idealised model configured in radiative-convective equilibrium (RCE) could demonstrate a large-scale clustering of convective clouds which is known as self-aggregation of convection (e.g, Held et al. (1993); Wing et al. (2017))." and lines 34-40: "Self-aggregation increases with the size and proximity of convective clouds and affects the radiative feedback, large-scale circulation, and moisture distribution in the vicinity of a cloud cluster (Hartmann et al., 1984). For instance, an idealised model setup shows that an aggregated state consists of a single moist region surrounded by dry regions. Moreover, the feedback between convection, surface fluxes, and radiation further drives aggregation (Tobin et al., 2012). Research shows that self-aggregation may increase with a warming climate (Wing et al., 2020). However, there remain uncertainties connected to a large model spread (Bläckberg and Singh, 2022)."

- **L58-60**: "At the same time, food security and a high climate risk expose West Africa to multiple threats (Berthou et al., 2019). Changing atmospheric conditions could intensify those hazards.": We know that these points are

important but not specifically related to the current research. These sentences should be moved to the final section, instead of the introduction, or removed.

**Author's response**: We agree and remove the sentence from the manuscript.

**Author's changes in the manuscript**: We removed the sentence in the revised manuscript.

- **L104-105**: "It is characterised by lower temperatures and a strong vertical ascent, which we identify by an extensive vertically contiguous 105 layer and a high radar reflectivity (e.g., Igel et al. (2014); Takahashi et al. (2017)).": Vertical ascent is not directly analyzed by the proposed method, neither in Igel et al. (2014) nor in Takahashi et al. (2017). This sentence should be modified.

  **Author's response**: You are right, we will rewrite this section to be more clear.

  **Author's changes in the manuscript**: Please find the updated description of the cloud and core detection framework in Sect. 2.1-2.3. and add in lines 139-142: "We use the ML-based predictions of the radar reflectivity as input data for the detection framework. While radar reflectivity does not directly measure vertical velocity, it may provide information for detecting hydrometeors associated with convective cloud development (Luo et al., 2008)."

- **L173-174**: Please describe the methodology of the moving windows in more detail. What types of iterations are applied to calculate the indices?

  **Author's response**: We add a more detailed description in Sect. 3.2, together with a visualisation of the approach in Fig. 3.

  **Author's changes in the manuscript**: We update the description of the moving windows in Sect. 3.2, lines 261-270: "To assess regional variability in convective organisation, we refrain from computing organisation indices over the entire domain. Instead, the AOI is partitioned into overlapping $3° \times 3°$ grid cells (e.g., Semie and Bony (2020); Tobin et al. (2012)). Given that the spatial extent and number of convective cloud elements affect the resulting index values, it may be beneficial to mitigate artifacts arising from cloud systems intersecting grid boundaries. In response, we implement a moving-window approach. The initial window is anchored at the northwestern corner of the AOI (27°–30° N, 27°–30° W) and is incrementally shifted by 1° in both the zonal and meridional directions (Figure 3). For each time step, the spatial organisation indices (SCAI, COP, and ROME) are computed within a $3° \times 3°$ window. To enhance statistical robustness and reduce sensitivity to window placement, we calculate a local mean across adjacent overlapping windows, assigning the averaged value to the central grid cell. This approach may reduce boundary-related discontinuities and contribute towards a more stable representation of convective structure, particularly in regions where cloud systems span multiple windows (Jin et al., 2022)."

- **L179**: "the frequency distribution shows an overlap of lower index values for all indices": This sentence is unclear. What does this mean by an overlap?

  **Author's response**: The distribution of all indices (SCAI, COP, ROME) indicates that lower values occur more often than high values. We change the text in the revised manuscript.

  **Author's changes in the manuscript**: We revise the description of the results in Sect. 4.1 and remove the sentence. In the revised mansucrit, see lines 277-284: "Figure 4 (a) shows that SCAI values predominantly range between 0 and 1, with a peak concentration between 0.2–0.4. Oceanic regions have a slightly higher frequency of SCAI values lower than 0.4, whereas values higher than 0.4 are more common over land. This finding may suggest SCAI detects stronger convective organisation over water. COP values are mainly distributed between 0.1 and 0.75, with the highest density between 0.2–0.45. Over the ocean, values above 0.4 are more frequent, whereas over land, lower values dominate — again pointing to stronger convective organisation over the ocean (Figure 4, b). ROME displays a right-skewed distribution, with most values falling below 15,000. Differences between land and ocean are minor compared to SCAI or COP (Figure 4, c). Overall, the results may indicate a marginally stronger convective organisation over oceanic regions, with ROME showing the weakest land–sea contrast."

- **L230-232**: What is the meaning of "a diverging convective activity"? In Fig. 7a, we cannot see where the maximum cloud area is. Where is "a lower cloud lifetime" in Fig. 7c?

  **Author's response**: In the original manuscript, we meant to describe a concurrent existence of clouds with a single core and multiple cores (which may be observed in Fig. 5 of the old manuscript). In the revised manuscript, we change the figures and text to be more clear and describe the relationship between organisation indices and cloud/core properties more directly. In Figure 7 (now: 6) (a), we show the average cloud area

associated to cloud tracks occurring in the respective 3° x 3° subset of the grid. We change "lower" to "shorter" lifetime and revise the description of the figure.

**Author's changes in the manuscript**: Please see Sect. 4.2 (Spatial patterns and statistical relationships). Spatial patterns observed in Figure 6 are described in lines 306-324.

- **L244,245**: What does "convective activity" mean in the sentences, and which figure shows this? Which figure and location show "a higher convective activity comes with a lower area ratio, a higher number of DCCs, a larger cloud and core area"?

  **Author's response**: Here, we used the term "convective activity" to refer to the cloud properties (cloud area) and the number of cores associated to a convective cloud, whereas a cloud with more cores often comes along a larger area. In the revised manuscript, we focus to describe the relationship between organisation indices and cloud/core properties directly and revise the text to be more clear.

  **Author's changes in the manuscript**: Figure 8 is now Figure 9. The description can be found in Sect. 4.3 (Temporal variability of cloud properties and organisation indices), lines 351-367. Moreover, we add an analysis of temporal changes in the correlation coefficient over land and sea in lines 368-377.

- **L254**: "the convective organisation is overall weaker around the equator": In 4.1, "convective organization" is not clearly defined and is not specifically described. The definition of organization must be clarified.

  **Author's response**: The definition of organisation can be found in Sect. 1. In Sect. 4.1 of the revised manuscript, we add a more detailed description of the indices to introduce a differentiation between weaker and stronger convective organisation. Following, lower values of SCAI (or higher values COP or ROME) may be associated to an enhanced clustering of convective clouds, indicating a stronger convective organisation. In contrast, higher values of SCAI (lower values COP or ROME) correspond to a more dispersed spatial distribution of convective clouds. Hence, they induce a weaker convective organisation.

  **Author's changes in the manuscript**: See for the definition of convective organisation Sect. 1, lines 25-27: "Although the term convective organisation has become increasingly popular in climate research, it is often used vaguely. Mapes and Neale (2011) broadly summarise organisation as "non-randomness in meteorological fields in convecting regions" and Sect. 3.1, lines 210f.: "Convective organisation describes the contrast between convective cells randomly distributed in space and time from those clustering together inducing a stronger convective organisation (Pendergrass, 2020)."

- **L255**: "Their impact on large-scale patterns of organisation is limited compared to MCSs": The meaning of this sentence is unclear.

  **Author's response**: In the original manuscript, we wanted to state that large MCSs with multiple cores may be more frequently associated to convective organisation, compared to convective cells with a single core. Since we do not classify convective regimes, such as MCSs, we revise the manuscript to be more clear.

  **Author's changes in the manuscript**: We removed the sentence from the revised manuscript.

- **L263**: "(Figure 9, a-b,g-h,i-j)": Fig.9-j does not exist.

  **Author's response**: Thank you for the remark, we change the figure description accordingly.

  **Author's changes in the manuscript**: Description and label of updated Figure 9 (now Fig. 14) can be found in Sect. 4.4.3 (Spatial distribution of percentiles), lines 454-479.

- **L264**: "Figure 7" should be "Figure 6". Figure 7 does not show seasonal distributions.

  **Author's response**: You are right, we change the text in the revised manuscript.

  **Author's changes in the manuscript**: We remove Figure 6 in the revised manuscript. A description of the spatial distribution of temporal differences in MAM and JJA can be found in Figure 9 (Sect. 4.3, 351-367).

- **L274**: What does "convective activity" indicate here?

  **Author's response**: Originally, the sentence corresponds to clouds with multiple core regions. We remove the term "convective activity" and describe instead e.g., the number of cores to make the text more clear (see Introductory Remarks).

  **Author's changes in the manuscript**: For a revised analysis between cloud properties and convective organisation, please see ,e.g., Sect. 4.4.2, lines 445-453: "While we detect statistically significant differences between percentile-based subsets and the dataset with all cloud tracks, the effect sizes for cloud and core properties remain mostly small to moderate. Our results indicate that strong convective organisation (low SCAI, high

COP and ROME) tends to co-occur with larger cloud and core areas, slightly less and lower cores, and slightly shorter lifetimes. The highest effect sizes may be found for the CTH, core height, and cloud lifetime. Weak organisation (high SCAI, low COP and ROME) is associated with smaller clouds, lower CTH, fewer cores, a smaller core area, lower core height, and shorter lifetimes. Here, we observe the highest effect sizes for the cloud area, number of cores, and cloud lifetime (Table 6). These findings - and the differences between the two percentile-based subsets - suggest that different aspects of cloud and core morphology may contribute to the strength of convective organisation."

- **L274-279**: In Figure 7, which index is used to define a most or least organized group?

  **Author's response**:The analysis is based on combing the subsets filtered by the percentiles of all indices (SCAI, COP, ROME) to identify hotspots of convective organisation. This includes the 10th percentile of SCAI and the 90th percentile of COP and ROME, which are used as thresholds to filter the dataset of all cloud tracks to reveal only cloud locations associated to the 10 % strongest or weakest convective organisation. We add a more detailed explanation in the revised manuscript.

  **Author's changes in the manuscript**: Please see Sect. 4.4, lines 379-390: "To identify regional patterns of convective organisation and their effects on cloud properties, we adopt a percentile-driven approach. There exist no universally defined thresholds to distinguish between weak and strong convective organisation. In response, we compute the 10th, 25th, 75th, and 90th percentiles based on the distribution of each organisation index (SCAI, COP, and ROME) using the cloud tracks between March to August 2019 (Table 5). These percentiles serve as thresholds to classify the data into subsets of weak and strong convective organisation, as induced by the interpretation of the indices: strong organisation may be related to low SCAI and high COP/ROME, weak organisation to high SCAI and low COP/ROME (Biagioli and Tompkins, 2023; Semie and Bony, 2020). Following, regions of strong convective organisation are defined as cloud tracks with an index value below the 10th percentile for SCAI or above the 90th percentile for COP and ROME. Conversely, regions of weak organisation correspond to values that lie above the 90th percentile for SCAI or below the 10th percentile for COP and ROME. To identify spatial and temporal patterns of convective organisation, we create two subsets from all data points in the dataset, whereas one represents the 10 % strongest convective organisation (Q10 for SCAI; Q90 for COP and ROME, hereafter: P90), and the other representing the 10 % weakest convective organisation (Q90 for SCAI; Q10 for COP and ROME, hereafter: P10).". In Sect. 4.4.4, we visualize the spatial distribution of these subsets (P90, P10), see lines 486-489: "In contrast to the former analysis, we examine the spatial distribution for clouds in the two subsets (P90, P10) (Section 4). These subsets of the 10 % strongest (P90) and the 10 % weakest (P10) convective organisation may help to identify cumulative hotspot regions averaged over the three indices. The data may allow us to analyse spatial patterns and temporal changes of convective organisation across two seasons from spring (March to May, MAM) to summer (June to August, JJA)."

- **L303-304**: "Overall, organised clouds (P90) come along a larger cloud anvil area, a longer cloud lifetime, a lower CTH, a lower area ratio, and more and larger DCCs": It is unclear from Figure 13 to see these relationships. The relationships between the indices and these properties should be directly compared. A part of the comparison between the indices and the number of DCC is shown in Fig. 4g-i.

  **Author's response**: Thank you for your comment, we will add a more detailed analysis of the relationship between convective organisation and cloud and core properties in the revised manuscript. These analyses can be found in Sect. 4.2 (for overall correlations between organisation indices and cloud/core properties) and 4.4 (for the comparison of percentile-based subsets to all cloud tracks).

  **Author's changes in the manuscript**: See Sect. 4.2, lines 325-334, and Sect. 4.4.2, lines 415-452.

- **L320-336**: The authors describe notable characteristics which can be seen from Fig. 14. However, some points are not convincing from the figure. Please check whether the description is consistent with the figures. For example, in L328-330, I cannot see a noticeable narrowing in summer (JJA) for ROME. The differences in the effect size are significant according to the numbers in the Figure, but not clearly visible.

  **Author's response**: Thank you for your remark, we revise the figure. Based on the reviewer comments, we add new analysis to investigate differences between subsets of organisation and focus less on seasonal differences (as the dataset covers only a six-month period). Hence, Figure 14 (13 in revised manuscript) shows a more direct comparison between the distribution of the organisation indices, cloud properties, and core properties for all cloud tracks, the 10 % strongest convective organisation, and the 10 % weakest convective organisation. These

analyses may provide insights on how cloud/core properties are related to convective organisation.

**Author's changes in the manuscript**: See Sect. 4.4.2 (Relationship between organisation subsets and cloud properties), lines 415-452, for a revised analysis of the relationship between organisation and cloud/core properties.

- **L350, L359-360**: "the microphysical cloud properties": Cloud anvil area is a cloud macrophysical characteristic rather than a microphysic property. It is true that cloud microphysics affect cloud anvil area through the balance between sedimentation and the outflow, the present study does not examine microphysical cloud properties specifically.

  **Author's response**: You are right, we change the term to "microphysical cloud properties" to "cloud properties" throughout the text in the revised manuscript.

  **Author's changes in the manuscript**: Introduced in Sect. 2.5, lines 186-189: "We filter the cloud trajectories to exclude possibly non-convective tracks from the analysis. For that purpose, we employ three criteria: (a) One or more core regions for at least 15 minutes, (b) radar reflectivity of higher than 0 dBZ at 10 km height for at least 15 minutes, (c) minimum CTH of 10 km and maximum CBH of less than 5 km for at least 15 minutes. While we do not require the convective clouds to have a CTH higher than 10 km at every time step during their trajectory, we discard trajectories that never reach the CTH threshold. After filtering the dataset, we receive 375,000 uniquely labeled 3D cloud objects, each190 associated with a continuous time trajectory and structural information about cloud and core properties (Figure 1, b)."

- **L361-362**: "For continental cloud clusters in the northern hemisphere, we find more distinct results regarding the relationship between DCCs and the degree of organization": This is one of the noticeable results discovered in this paper. However, this conclusion is indirectly shown by the figures in 4.1.2. The authors are suggested to show a more direct analysis showing the conclusion.

  **Author's response**: Thank you for your comment. Since we found overall only small differences between oceanic and continental clouds, we aim to focus on a more direct analysis of the relationship between convective organisation and cloud properties in the revised manuscript. Based on the reviewer comments, we modify the division into subsets for each hemisphere; however, we describe changes that may be associated to the shift of the ITCZ in boreal summer. Revised analyses may be found in Sect. 4.4.2, 4.4.3, and 4.4.4. A discussion of potential drivers and influences on organisation may be found in Sect. 5.2.

  **Author's changes in the manuscript**: Please see, e.g., Sect. 4.4.3, lines 460-467: "As shown in Sects. 4.1 and 4.2, high SCAI values - indicating weak convective organisation - are typically concentrated near the equator. In spring, low SCAI values (Q10) occur over the equatorial Atlantic Ocean and land/sea areas south of 15° S. High values (Q90) appear over equatorial Africa (0°–15° N), especially in rainforest zones, and Cameroon. The IQR peaks near the equator, particularly over the Ivory Coast, Guinea, Benin, Angola's coast, and Lake Victoria (Figure 14, a–c). In summer, values shift north to 0°–15° N, with SCAI Q10 regions over the Atlantic and coastal West Africa. High SCAI (Q90) values occur in spring over the Congo and Central African Republic. The IQR also shifts north in summer, with hotspots over the West African plains, Jos Plateau, and Congo River basin (Figure 15, a–c).", Sect. 4.4.4, lines 499-504: "n summer, the spatial distribution of strong convective organisation shifts northward. Regions with a frequent occurrence of strong convective organisation emerge over the Atlantic Ocean and become more widespread across the West African plains, including areas around the Niger and Congo rivers (Figure 16, c). Weak organisation, on the other hand, is concentrated primarily over continental Africa, especially between 15° and 30° E, with a peak located just north of the Congo River (Figure 16, d)" or Sect. 5.2, lines 559-567: "We also detect a link between convective core occurrence and organisation that may follow the northward migration of the ITCZ in boreal summer. As the ITCZ shifts, it may alter regional circulation, surface energy balance, and moisture availability — particularly influencing cloud development over the northern Sahel and southern Sahara, as observed by, e.g., the spatial distribution of SCAI between June and August (Section 4.4.3). These changes may be associated with increased humidity, educed subtropical subsidence, and deeper ascent within the tropical rainbelt (Fontaine and Philippon, 2000). Together with strengthened meridional pressure gradients (Lavaysse et al., 2009), they may contribute to the occurrence of large convective systems with multiple cores. This observation may be reflected in our results as a northward displacement of convective clouds and an increase in cloud area, core area, and core number over continental Africa in July and August (Section 4.2, Section 4.3)."

- **L468**: Spell out "JAS".

  **Author's response**: Changed abbreviation "JAS" to "Journal of Atmospheric Sciences".

**Author's changes in the manuscript**: Changes can be found, e.g., in lines 650, 681, or 685.

---

## Author Comment (AC2)

**Final response**
**(Manuscript "EGUSPHERE-2025-376")**
**Reviewer 2**

Sarah Brüning and Holger Tost

May 23, 2025

**1   Introductory remarks**

We would like to thank the reviewer for taking your time to give us constructive and detailed feedback. In the following sections, we address all comments. For every comment, you can find (1) the comment, (2) the author's response (both with lines from initial manuscript), and (3) the author's changes in the revised manuscript. The lines given for **author's changes in the manuscript** refer to the lines in the **revised manuscript**.

Based on all referee comments, we focus the revision on improving the clarity of the written expression, in particular the description of our methods and the results. While the methods used in the manuscript did not change, we aim to focus on describing the data and results more directly. This includes, in particular, the distribution of organisation indices, their connection to cloud and core properties, and differences for the percentile-based subsets of organisation. As the period of six months may to too short to derive a distinct seasonality, we revise the description of hemisphere- or season-based subsets and replace the term "seasonal" changes with "monthly" changes which we observed during the period. Moreover, we changed the abbreviation "DCC" to "cores". Throughout the manuscript, we replace terms like "convective activity" with a more direct description of the cloud properties (cloud area, core number). We changed the figure labels in the following way:

- 1 -> 1

- 2 -> 3

- 3 -> 4

- 4 -> 5

- 5 -> Removed

- 6 -> Removed

- 7 -> 6

- 8 -> 9

- 9 -> 14

- 10 -> 15

- 11 -> 16

- 12 -> 8

- 13 -> Removed

- 14 -> 13

- New: 2 (Summary of cloud tracks: Latitude, Surface type, Month, Lifetime, Daytime of first detection, number of cores)

- New: 7 (2D histogram of average relationship between organisation indices and cloud/core properties)

- New: 10 (Summary of percentile-based subsets P90 and P10: Spatial distribution grouped by surface type (land,ocean) and month (March-May,June-August))

- New: 11 (Summary of percentile-based subsets P90 and P10: Surface type, number of cores, month, lifetime)

- New: 12 (Correlation matrix for organisation indices, cloud properties, and core properties for all cloud tracks, P90, and P10)

**2 General comments**

- **Comment**: The main recommendation is to add new analyses to address the scientific objective more directly. The aim of the study is to get at the relationship between organisation and cloud properties, but very little of the analysis progresses this aim despite the opportunity to do so with the dataset developed. The focus on seasonal, land/ocean and hemispheric differences are relatively arbitrary, and not well developed or discussed. Further, with only 6 months of data, from one year, the seasonal results in the paper are of limited value as they do not capture the entire seasonal cycle or account for inter-annual variability. Instead, we suggest more focus on direct comparisons between key cloud properties and the level of organisation measured using convective indices and the tracking of convective cores.

  **Author's response**: Thank you for your comment. We revise the manuscript based on your suggestions to focus on a more direct comparison of organisation and cloud/core properties. While we do not aim to provide a climatology, which is unfortunately out of scope for our study, we rather used the seasonal/hemispheric differences to showcase the variability of derived properties. However, we agree with you and add new analyses to focus on investigating the connection between convective organisation and cloud properties in the dataset and percentile-based subsets. You may find these analyses in Sect. 4 (i.e., Figure 7, Figure 10-12) of the revised manuscript.

  **Author's changes in the manuscript**: We an analysis of the relationship between organisation indices, cloud properties, and core properties in Sect. 4.2, lines 326-335: "To quantify the relationship between organisation indices and cloud properties, we compute Spearman's rank correlation coefficient R using data from all cloud tracks (Figure 7). The logarithmic distributions reveal a general skew toward low values for SCAI, ROME, cloud area, lifetime, number of cores, and core area. The correlation analysis shows that COP and ROME may be positively associated with cloud area, lifetime, CTH, number of cores, and core height (Figure 7, g–r). In contrast, SCAI is negatively correlated with all of these properties except for CTH and the core height (Figure 7, a–f). For the core area, we see a weak negative correlation to all indices. The findings suggest that stronger convective organisation may be statistically linked to larger, longer-lived cloud systems, a higher CTH and core height, and more cores. Interestingly, these statistical relationships contrast with some spatial patterns in Figure 6. For instance, while higher ROME values spatially co-occur with smaller clouds and shorter lifetimes in some regions, correlation coefficients suggest that, overall, organisation increases with cloud area and duration. However, most correlations are weak, with maximum coefficients around 0.26 between ROME and the cloud lifetime. They highlight the complex and regionally variable nature of these relationships.". In Sect. 4.3, we show how these correlations change along the period, lines 368-377: "To evaluate how the relationships between organisation indices and cloud/core properties evolve along the two seasonal subsets, we compare correlation coefficients in spring (MAM) and summer (JJA). Overall, SCAI maintains negative correlations with cloud properties, while COP and ROME remain positively correlated. The direction of correlation does not change along the period, though some coefficients vary in strength. From spring to summer, correlations between SCAI and cloud properties increase slightly - except for the CTH and core height. Correlations between COP and cloud properties predominantly increase, whereas the differences are lower than for SCAI. For ROME, we see an increase for the correlation to the cloud lifetime, CTH, and core height, and a decrease to the cloud area and core area. However, these shifts are small, with changes up to 0.11 (SCAI vs. cloud lifetime, CTH, and core height). Despite apparent spatial patterns and temporal shifts in convective cloud organisation and structure as seen in Figs. 8 and 9, statistical relationships remain overall weak (Table 4).". Please see Sect. 4.4.2 (lines

415-452) for a detailed comparison of the correlations between all cloud tracks and the percentile-based subsets of the 10 % strongest and 10 % weakest convective organisation.

- **Comment**: The hemispheric differences emphasised in the text may mainly result from the large oceanic anomaly (-15 S, 0 E; Figures 6f, 7acdef, 8abcd). But this tends to be a region of large-scale descent covered mainly by low level stratocumulus, rather than deep convection. Is it possible that systems here are being misidentified as DCCs (eg., mid-latitude cyclones or atmospheric rivers)? Given the small sample size (Figures 5,6), average properties here may be unreliable. The authors should address this, and focus on distinct convective regimes rather than hemispheric land/ocean divisions.

  **Author's response**: Thank you for your comment. We have checked the data by evaluating the vertical depth of cloud layer for the identified objects. Here, we found an error in the code that led to the misidentification of potentially low level stratocumulus as deep convection, in particular in the southern hemisphere. We fixed the error and revised the figures and their descriptions. You may find the updated results in the revised manuscript. Moreover, we removed Figures 5 and 6 aiming to focus on a direct evaluation of observed cloud/core properties in the domain (as done it Figure 7, now Figure 6).

  **Author's changes in the manuscript**: Please see Sect. 4.2 (Spatial patterns and statistical relationships), lines 306-324 and Sect. 4.3 (Temporal variability of cloud properties and organisation indices), lines 351-367.

- **Comment**: The quality of written expression needs improvement. In particular, the authors frequently make claims that are not evidenced by the results. Comparable spatial patterns alone do not justify inferred relationships. In addition, there are issues with referencing throughout. Citations are frequently used as evidence for a methodological choice or finding that are merely examples of comparable works. Please clarify why a statement needs referencing, or merely state the result.

  **Author's response**: Thank you for your comment, we revised the referencing and written expression throughout the text to focus on a more clear description of the results. We add analysis to directly show the relationships between organisation indices and cloud/core properties (Figs. 7, 12, 13).

  **Author's changes in the manuscript**: Pleas see, e.g., Sect. 4.2, lines 325-335 for the description of the relationship between organisation indices and cloud/core properties: "To quantify the relationship between organisation indices and cloud properties, we compute Spearman's rank correlation coefficient R using data from all cloud tracks (Figure 7). The logarithmic distributions reveal a general skew toward low values for SCAI, ROME, cloud area, lifetime, number of cores, and core area. The correlation analysis shows that COP and ROME may be positively associated with cloud area, lifetime, CTH, number of cores, and core height (Figure 7, g–r). In contrast, SCAI is negatively correlated with all of these properties except for CTH and the core height (Figure 7, a–f). For the core area, we see a weak negative correlation to all indices. The findings suggest that stronger convective organisation may be statistically linked to larger, longer-lived cloud systems, a higher CTH and core height, and more cores. Interestingly, these statistical relationships contrast with some spatial patterns in Figure 6. For instance, while higher ROME values spatially co-occur with smaller clouds and shorter lifetimes in some regions, correlation coefficients suggest that, overall, organisation increases with cloud area and duration. However, most correlations are weak, with maximum coefficients around 0.26 between ROME and the cloud lifetime. They highlight the complex and regionally variable nature of these relationships."

- **Comment**: The dataset was not adequately detailed, and relevant limitations and validation were not addressed. Please justify whether the derived-radiances and cloud-tracking methods are suitable, and clarify uncertainties across the AOI, seasons, diurnal cycle, cloud regimes or lifecycle. Can mid-level cloud and cumuli arise that may instead determine the cloud area extent? How were the machine learning derived reflectivities evaluated? A summary figure of the tracked clouds would help introduce the data, and help to interpret the robustness of the organisation indices in different areas/seasons/time of day. Additionally, normalised density maps alone do not convey absolute occurrences of the data shown (Figures 5,6,9,10 and 11); this information is important for interpreting the results and should be provided, perhaps present frequency maps instead of densities.

  **Author's response**: We add a more detailed introduction for the dataset, the machine-learning algorithm, the detection framework, and the derived cloud properties in Sect. 2 (Data), Sects. 2.1-2.5. Moreover, we include a summary of the tracked clouds regarding their spatial and temporal distribution (Figure 2). You may find a discussion of the uncertainties and limitations of the machine learning-based predictions and the detection framework in Sect. 2.5 and in Sect. 5.3. In Sect. 2.5 we describe how we potentially convective cloud tracks by employing a minimum core number of one core, a CTH of 10 km, and a radar reflectivity of 0 dBZ at 10 km

height for at least 15 minutes to classify the cloud trajectory as convective. Otherwise, we exclude the trajectory from further analysis. These criteria may help to reduce the presence of mid-level cloud and cumuli in the dataset. We revise the figures 9-11 (now: Figs. 14-16) to present frequencies instead of densities.

**Author's changes in the manuscript**: See subsections of Sect. 2 (Data) for a detailed description of the satellite data fed into machine-learning model (Sect. 2.1), the machine-learning model and a 3D extrapolation of radar reflectivities (Sect 2.2.), the object-based cloud detection framework (Sect. 2.3), and the extraction of key cloud and core properties used in this study (Sect. 2.4). In Sect. 2.5, we describe how to filter possibly convective cloud tracks. Moreover, this section includes a discussion of potential limitations and provides a summary of the spatial and temporal distribution of detected cloud tracks. For the description of Figs. 14-16, see Sect. 4.4.3 and Sect. 4.4.4, e.g.. The data is introduced in Sect. 4, lines 379-394: "To identify regional patterns of convective organisation and their effects on cloud properties, we adopt a percentile-driven approach. There exist no universally defined thresholds to distinguish between weak and strong convective organisation. In response, we compute the 10th, 25th, 75th, and 90th percentiles based on the distribution of each organisation index (SCAI, COP, and ROME) using the cloud tracks between March to August 2019 (Table 5). These percentiles serve as thresholds to classify the data into subsets of weak and strong convective organisation, as induced by the interpretation of the indices: strong organisation may be related to low SCAI and high COP/ROME, weak organisation to high SCAI and low COP/ROME (Biagioli and Tompkins, 2023; Semie and Bony, 2020). Following, regions of strong convective organisation are defined as cloud tracks with an index value below the 10th percentile for SCAI or above the 90th percentile for COP and ROME. Conversely, regions of weak organisation correspond to values that lie above the 90th percentile for SCAI or below the 10th percentile for COP and ROME. To identify spatial and temporal patterns of convective organisation, we create two subsets from all data points in the dataset, whereas one represents the 10 % strongest convective organisation (Q10 for SCAI; Q90 for COP and ROME, hereafter: P90), and the other representing the 10 % weakest convective organisation (Q90 for SCAI; Q10 for COP and ROME, hereafter: P10). These may represent so-called "hotspots". We also define the interquartile range (IQR, values between the 25th–75th percentile) to represent a baseline, which is used to contrast the spatial distribution of average organisation against the identified hotspot regions."

- **Comment**: It should be clarified what advances are made from using the 3D fields and whether these justify uncertainties associated with additional data processing, as comparable cloud tracking and properties can be derived from 2D data.

  **Author's response**: The predicted 3D cloud field may enable a simultaneous analysis of horizontal cloud properties derived from 2D data (such as geostationary satellite imagery) while at the same time providing detailed information on the vertical cloud column (from which we derive the vertical extension of cloud and core). In particular for the analysis of core properties, we think our data may provide intriguing insights as most core detection methods use either 2D data from active sensors (like CPR) or passive sensors (like geostationary/polar-orbiting satellite); but no comprehensive 3D perspective. In contrast to ground-based radar, we achieve a higher spatial coverage for remote oceanic regions. Hence, we estimate our data to fill the data gap where ground-based instruments are not available. We address this in Sect. 1 and Sect. 2 of the revised manuscript.

  **Author's changes in the manuscript**: Sect. 1, lines 71-78: "Our study employs convective cloud trajectories derived from a 4D time series of contiguous 3D radar reflectivities, which we predict from a machine learning (ML)-based extrapolation of 2D satellite data (Brüning et al., 2024). We employ an object-based algorithm to detect and track convective clouds in the predicted radar reflectivity field. This perspective allows a simultaneous coverage of the horizontal (cloud and core area) and vertical (cloud and core height) properties in the AOI, including remote oceanic regions over the Atlantic Ocean. Our aim is to showcase how convective organisation is distributed in the AOI within the six-month period. Furthermore, we strive to quantify how differences in the cloud and core properties are connected to a weak or strong convective organisation." and Sect. 2.5, lines 200-207: "While this framework enables a seamless tracking of convective systems along the ML-based 4D time series, it remains subject to several limitations. The predicted data display a ML-based extrapolation of the received CloudSat CPR reflectivities. Hence, they include uncertainties connected to the ML model, such as the blurriness of predictions induced by the loss function which optimizes towards the mean. We receive few information on thin ice clouds due to a reduced sensitivity of the CloudSat CPR to ice clouds in high altitudes (Sassen and Wang, 2008). Moreover, the detection framework rests on an object-based perspective to investigate

atmospheric processes. We note the identified trajectories may underlie simplifications caused by an inherent subjectivity of the thresholds applied in the cloud detection step. Nevertheless, the approach may help to bring further insights into the structure and organisation of convective clouds."

- **Comment**: "Anvil cloud" and "anvil area" are discussed, but at no point is a cloud anvil defined. If no work was done to segment the anvil from the cloud objects tracked or ensure the cloud area extent always results from an anvil feature, all references to cloud anvils should be specified as "cloud" or "cloud area".
  **Author's response**: Thank you for the remark, we agree and change "cloud anvil area" to "cloud area" in the revised manuscript.
  **Author's changes in the manuscript**: Introduced in Sect. 2.4 (Extraction of cloud properties), lines 179-181: "The cloud area is computed from the column-wise maximum horizontal extent of the 3D cloud mask, while CTH is derived from the vertical extent."

- **Comment**: The calculation of cloud properties and spatial density distribution was not described.
  **Author's response**: We add a detailed description of derived cloud and core properties in Sect. 2 of the revised manuscript.
  **Author's changes in the manuscript**: Please see Sect. 2.4 and Table 3 for the introduction of cloud and core properties derived from the contiguous convective cloud trajectories. The description can be found in lines 176-185: "We use the labelled cloud masks to extract cloud and core properties at each point in time. Moreover, we compute average properties across the cloud's lifetime to derive distinct key properties that may characterise the trajectory. These properties include the cloud lifetime, cloud area, cloud top height (CTH), number of cores, and mean core area and height (Table 3). The cloud area is computed from the column-wise maximum horizontal extent of the 3D cloud mask, while CTH is derived from the vertical extent. For the cloud lifetime, we extract the time (in hours) between the first and last detection of each trajectory of the labelled pixels. Surface type is assigned via a binary land-sea mask and the modal value for the locations of the cloud trajectory within this land-sea mask. For clouds with one or more cores, we count the maximum number of cores associated to the trajectory. Moreover, the core area and height are derived from the column-wise maximum horizontal extent and vertical extent of the previously identified cores, similar to the cloud area and CTH". Spatial densities were replaced by frequency distributions, a description may be found in Sect. 4.2, lines 306f.: "Figure 6 presents the spatial distribution of the three organisation indices (SCAI, COP, ROME), along with associated cloud and core properties, interpolated onto a $3° \times 3°$ grid and displayed as latitudinal cross-sections."

- **Comment**: Assessing the area / core ratio does not seem to add much information to this work, additional justification of why this was included and what it shows would be welcome.
  **Author's response**: The area/core ratio was used to analyze how the relation between the cloud area size and the core area affects convective organisation. However, we replace the area ratio with the core height in the revised manuscript to include more vertical resolved properties.
  **Author's changes in the manuscript**: We removed "area ratio" and replace its with the "core height" as described in Sect. 2.4 (see above or Table 3).

**3   Specific comments**

In the next section, we shortly address your specific comments. Changes will be added in the revised manuscript.

- **L36**: "contiguous convective regions" unclear whether this refers to a convective cloud, multiple connected convective clouds, or an MCS. There are issues throughout on the use of "MCS" and "organised states" (also L260, L276). In addition, the following line regarding MCS organisation is a little unclear.
  **Author's response**: In the original manuscript, the sentence with "regions" may point towards the number of cores within a convective cloud. Since we do not classify distinct cloud regimes, we revise the text to directly address observed results (lines 260, 276). Throughout the manuscript, we focus on improving the written expression.
  **Author's changes in the manuscript**: We removed the sentence in the revised manuscript.

- **L45-46**: [discussion on convective aggregation]: Contradicts statement at L53-54. Aggregation is different to organisation, and is generally a phenomena seen in RCE models, please clarify.

**Author's response**: You are right, we revise the text and add the definition of organisation and aggregation to clarify the text in the revised manuscript.

**Author's changes in the manuscript**: Please see Sect. 1, lines 25-31: "Although the term convective organisation has become increasingly popular in climate research, it is often used vaguely. Mapes and Neale (2011) broadly summarise organisation as "non-randomness in meteorological fields in convecting regions". This definition induces a clustering of deep convective cells which is ubiquitous in the atmosphere, particularly in the tropics. However, the underlying mechanisms remain insufficiently understood (Muller and Bony, 2015). While convective organisation is difficult to quantify in observational data, idealised model configured in radiative-convective equilibrium (RCE) could demonstrate a large-scale clustering of convective clouds which is known as self-aggregation of convection (e.g, Held et al. (1993); Wing et al. (2017))." for the introduction of organisation and self-aggregation. We change lines 34-40: "Self-aggregation increases with the size and proximity of convective clouds and affects the radiative feedback, large-scale circulation, and moisture distribution in the vicinity of a cloud cluster (Hartmann et al., 1984). For instance, an idealised model setup shows that an aggregated state consists of a single moist region surrounded by dry regions. Moreover, the feedback between convection, surface fluxes, and radiation further drives aggregation (Tobin et al., 2012). Research shows that self-aggregation may increase with a warming climate (Wing et al., 2020). However, there remain uncertainties connected to a large model spread (Bläckberg and Singh, 2022)." and lines 48-50: "Providing timely forecasts and a robust climate risk assessment requires even more a correct representation of convective organisation. While satellite observations has shown that organisation within the tropics may increase overall with extreme precipitation (Semie and Bony, 2020), we have limited knowledge about convective organisation on a regional level."

- **L58**: MCSs contribute not only the majority of extreme rainfall, but all rainfall, and the stratiform component of their precipitation is important for this. Highlighting could provide more justification for the tracking of the entire anvil, not just the convective cores

   **Author's response**: Thank you for your remark, we change the text in the revised manuscript.

   **Author's changes in the manuscript**: Sect. 1, lines 52-55: "The area of interest (AOI) covers West Africa and the tropical Atlantic Ocean between 30° N–30° S and 30° W–30° E and lies within the Inter-Tropical Convergence Zone (ITCZ). Here, the environmental conditions favour the development of deep convective clouds, which are often associated to heavy rain (Takahashi et al., 2023)."

- **L71**: "DCC" used as an acronym for deep convective cores, while it is more commonly used as an acronym for deep convective clouds. Would recommend referring to the convective cores simply as "cores" as used in some figure headings

   **Author's response**: We change "DCCs" to "cores" in the revised manuscript.

   **Author's changes in the manuscript**: See Sect. 1, lines 64-66: "In Part 1 of this sequence of papers, we derived contiguous trajectories of convective clouds and their deep convective core regions (hereafter: cores) in 15-minute intervals for a six-month period between March to August 2019 (Brüning and Tost, 2025)."

- **L83**: This talks about "cloud development", but the lifecycle of tracked convective clouds is not assessed in this manuscript. Is this meant to refer to the distribution of observed cloud properties?

   **Author's response**: We agree, the cloud life-cycle is not covered in this study. The sentence refers to the cloud properties derived from the trajectories. We change the text in the revised manuscript to be more clear.

   **Author's changes in the manuscript**: Sect. 1, lines 64-69: "In Part 1 of this sequence of papers, we derived contiguous trajectories of convective clouds and their deep convective core regions (hereafter: cores) in 15-minute intervals for a six-month period between March to August 2019 (Brüning and Tost, 2025). In this study, we examined cloud and core properties of tropical convection and the life-cycle of single-core and multi-core convective clouds. In this paper, we aim to complement the findings by an in-depth analysis of spatio-temporal patterns of convective organisation. Moreover, we aim to investigate the connection between convective organisation and cloud properties within the AOI."

- **L86, L350, L359, L406**: No microphysical properties of clouds (i.e. Effective radius, droplet number density etc.) are discussed in this paper. Instead, the authors may be referring to the bulk or macrophysical properties of clouds

   **Author's response**: Thanks for the remark; we will change "microphysical cloud properties" to "cloud properties" in the revised manuscript.

**Author's changes in the manuscript**: See, e.g., line 69: "Moreover, we aim to investigate the connection between convective organisation and cloud properties within the AOI.".

- **L93**: Do you include the 3.9 micron channel (channel 4) from SEVIRI? This channel includes a contribution from reflected solar IR, so has a different response between day time and nighttime. Has the consistency of the input data been verified across the diurnal cycle?

  **Author's response**: Yes, the channel at 3.9 µm is included in the model training. While we evaluated the model performance in our previous paper, we did not specifically focus on the consistency of the input data across the diurnal cycle. However, we did not detect substantial differences of the data quality. We add a more detailed description of the machine learning model in Sect. 2.

  **Author's changes in the manuscript**: Sect. 2.1 introduces the satellite data used for the machine-learning model which is described in more detail in Sect. 2.2; in Sect. 2.5, we provide a summary of the detected clouds and discuss some limitations of the machine-learning model and the object-based detection algorithm.

- **L101**: The resolution is 3km at nadir, however this increases further away from the sub-satellite point. Should the comment on the vertical resolution say that it is that of Cloudsat CPR, not SEVIRI?

  **Author's response**: The predicted 3D data have a horizontal resolution of 3 km at nadir, which is similar to MSG SEVIRI. A description may be found in Sect. 2.2. We address a reduction of the model performance which may be connected to a decreasing spatial resolution towards the poles and further uncertainties in Sect. 2.5 and 5.3 of the revised manuscript. You are right, the vertical resolution is derived from the CloudSat CPR which is used for validation of the model. We change the text in the revised manuscript to be more clear.

  **Author's changes in the manuscript**: Sect. 2.2, lines 111-121: "The AOI for the reconstructed 3D cloud field spans from 60° W to 60° E and from 60° S to 60° N, corresponding to 2400 × 2400 pixels in the horizontal dimensions. MSG SEVIRI satellite imagery serves as input to the Res-UNet model, setting the horizontal resolution of the 3D data to 3 km × 3 km. Initially, we used 11 spectral channels covering the visible, near-infrared, and thermal-infrared ranges. However, the visible channels were excluded in this study to enable daylight-independent predictions (Tables 1 and 2). The training data consist of 128 × 128 pixel patches of MSG SEVIRI imagery, spatially and temporally aligned with CloudSat overpasses. Each training sample includes a diagonal CPR cross-section. To address the resolution mismatch between MSG SEVIRI and CloudSat, the CPR data are downsampled to match the horizontal resolution of MSG SEVIRI pixels. To mitigate the strong class imbalance between cloudy and cloud-free conditions, cloud-free samples are limited to a maximum of 10 % of the training data. The model is trained on nine months of data and validated on a separate three-month period. It is optimised to reconstruct CloudSat-like 3D reflectivity volumes with a horizontal resolution of 100 × 100 pixels and 90 vertical levels." and lines 129-132: "Visual inspection confirms that no artifacts are present at tile boundaries, indicating seamless reconstruction across the domain. However, model accuracy tends to decrease with increasing distance from the MSG SEVIRI nadir."

- **L102**: onwards (discussion of cloud tracking): The detection starts with a very low radar reflectivity threshold of -15dBz, which will detect all clouds, not just convective cloud features. Is any restriction on cloud height applied, or will this also detect low level liquid clouds? How are scenes with multiple layers clouds handled?

  **Author's response**: Based on former studies, we chose the low threshold of -15 dBZ to track clouds contiguously beginning from early stages of their life-cycle. However, this threshold alone may lead to the detection of non-convective clouds. That is why we filter the cloud trajectories after the detection and linking step. To determine a potentially convective track, we apply a minimum cloud height (> 10 km) and radar reflectivity (> 0dBZ). Trajectories that do not pass these criteria are discarded from the dataset. However, scenes with multiple cloud layers are difficult to handle. In our study, unfortunately, we may not account for their occurrence. In the revised manuscript, we add a more detailed description of the detection and tracking algorithm.

  **Author's changes in the manuscript**: Please see Sect. 2.3, lines 142-147: "We identify potential candidates of convective clouds within the 3D cloud field by applying a fixed radar reflectivity threshold of –15 dBZ. This threshold is used to distinguish hydrometeors from background noise in the radar reflectivity data (Marchand et al., 2008). Although moderately restrictive, this threshold is intended to capture the full spatio-temporal evolution of convective clouds throughout their life cycle, thereby supporting the formation of contiguous trajectories (Esmaili et al., 2016)." and Sect. 2.5, lines 185-190: "We filter the cloud trajectories to exclude possibly non-convective tracks from the analysis. For that purpose, we employ three criteria: (a) One or more core regions for at least 15 minutes, (b) radar reflectivity of higher than 0 dBZ at 10 km height for at least 15

minutes, (c) minimum CTH of 10 km and maximum CBH of less than 5 km for at least 15 minutes. While we do not require the convective clouds to have a CTH higher than 10 km at every time step during their trajectory, we discard trajectories that never reach the CTH threshold. After filtering the dataset, we receive 375,000 uniquely labeled 3D cloud objects, each associated with a continuous time trajectory and structural information about cloud and core properties (Figure 1, b)."

- **L108**: only one threshold was used to define the cloud boundary, you apply some filtering to elongated clouds, but does this mean that in other cases some organised systems are counted as one cloud object?
  **Author's response**: In our study, we allow cloud systems to contain several core regions - which characterises the system itself as more organised. We only split up these cloud clusters, if their shape appears more elongated than rounded. However, the analysis of the organisation indices (which we employ to investigate spatial organisation) uses a moving-window approach that may take the neighborhood of indivdiual objects into account.
  **Author's changes in the manuscript**: We revised Sect. 2.3, lines 154-157: "Subsequently, we analyse the morphology of each cloud to determine whether any structures might represent a merger of multiple cloud systems. Each cloud's shape is characterised using the best-fitting ellipse, and we compute the aspect ratio — that is, the ratio of the major to the minor axis length. If the major axis is more than 75 % longer than the minor axis, we split the identified cloud into separate objects for further analysis."

- **L110**: "convective updraft", data for an "updraft" (i.e. vertical velocity) is not used. Is this instead based on a radar reflectivity threshold?
  **Author's response**: You are right, we do not assess the "updraft". We employ the radar reflectivity as a proxy to detect moist convection. We revise the manuscript to be more clear.
  **Author's changes in the manuscript**: Sect. 2.3, lines 139-140: "We use the ML-based predictions of the radar reflectivity as input data for the detection framework. While radar reflectivity does not directly measure vertical velocity, it may provide information for detecting hydrometeors associated with convective cloud development (Luo et al., 2008)."

- **L111**: please clarify how the local extrema are used to detect core objects. Is there any contiguity of these required in time, and does using the local extrema to define cores result in there always being >= 1 core at each time? If a cloud has many cores over its lifetime but they don't co-occur in time, is it still reported as having multiple cores?
  **Author's response**: We add a more detailed description of the detection algorithm in the revised manuscript. Like the clouds, the cores are detected for each time step separately, by a combined perspective of the with radar reflectivity values and the vertical depth of potential cores along the vertical columns of the labelled cloud mask. Each cloud may contain multiple cores. While there may be time steps along the trajectory where we detect no core, clouds with cores that do not co-occur in time are reported as single-core (since we derive the maximum core number occurring simultaneously).
  **Author's changes in the manuscript**: See Sect. 2.3, lines 163-174: "We aim to detect convective cores for each cloud object at every time step throughout its life cycle. For this purpose, we use the previously generated labeled 3D cloud mask. Core centroids are identified by locating local maxima in a combined metric that incorporates both smoothed radar reflectivity and the vertical extent of a contiguous potential core layer. Specifically, we calculate the mean radar reflectivity for each vertical cloud column, and determine the height of the core layer by counting the number of pixels with reflectivity values greater than 0 dBZ located above 5 km altitude. To fill isolated gaps in otherwise vertically continuous cores, we expand the threshold from 0 dBZ to –5 dBZ in columns that contain at least one pixel exceeding 0 dBZ (Luo et al., 2008; Igel et al., 2014). We then combine both indicators —average reflectivity and potential core vertical depth — for each pixel associated with a cloud label, resulting in a 2D layer where we search for local maxima. If at least one local maximum is detected, the corresponding locations are considered candidate core centroids. If no local maxima are found — for example, if no columns contain pixels above 0 dBZ at altitudes higher than 5 km — the cloud is recorded as having zero cores for that time step. Otherwise, we use a 3D watershed segmentation algorithm to delineate the core volumes surrounding each centroid, allowing for multiple cores to exist within a single cloud at the same time."

- **L118**: In the introduction (L48) you state that observational studies have been limited as there is a "low frequency of events most relevant for aggregation", is your 6 months of data sufficient?
  **Author's response**: The data used in this study may not provide a climatology of convective organisation; for this purpose, a long-term time series should be analyzed. However, we rather aim to show-case the machine

learning-based data and observed relationships between detected cloud properties and organisation indices. We revise the manuscript to be more clear about our objective and the limitations.

**Author's changes in the manuscript**: Please see Sect. 1, lines 67-77: "In this paper, we aim to complement the findings by an in-depth analysis of spatio-temporal patterns of convective organisation. Moreover, we aim to investigate the connection between convective organisation and cloud properties within the AOI. For this purpose, we quantify convective organisation at each point in time by employing three organisation indices. The goal is to derive spatial patterns of organisation and compare their spatio-temporal variability (Biagioli and Tompkins, 2023). Our study employs convective cloud trajectories derived from a 4D time series of contiguous 3D radar reflectivities, which we predict from a machine learning (ML)-based extrapolation of 2D satellite data (Brüning et al., 2024). We employ an object-based algorithm to detect and track convective clouds in the predicted radar reflectivity field. This perspective allows a simultaneous coverage of the horizontal (cloud and core area) and vertical (cloud and core height) properties in the AOI, including remote oceanic regions over the Atlantic Ocean. Our aim is to showcase how convective organisation is distributed in the AOI within the six-month period. Furthermore, we strive to quantify how differences in the cloud and core properties are connected to a weak or strong convective organisation." and Sect. 5.3, lines 571-600, for the discussion of limitations.

- **Section 3.1**: It would be useful to have a final paragraph comparing the pros and cons of each organisation metrics and in which situations they are more or less reliable, rather than isolated comments or their capabilities which are difficult to compare

    **Author's response**: Thank you for your suggestion, we revise Sect. 3.1 to add a comparison of the indices.

    **Author's changes in the manuscript**: Sect. 3.1, lines 247-259: "While SCAI and COP are easy to compute, the calculation of ROME is less convenient. Since it has been designed to retrieve information from radar reflectivities, we include the index in our study. In contrast to SCAI and COP, ROME may also be computed when only a single object is present. As evaluated by, e.g., Mandorli and Stubenrauch (2024) and Biagioli and Tompkins (2023), each index has its own strengths and weaknesses. SCAI is insensitive to the size of the objects and mainly dominated by the variability in the number of clouds. However, it is less affected by shifts in time and space which induce high fluctuations of the index values, e.g, due to changes in the resolution of the input image or between two consecutive time steps. In contrast, the calculation of COP includes the object area. While COP correctly increases with the proximity and size, it is sensitive to noise caused in a domain with only a few objects. The index is correlated to the image resolution and shows a high variability for consecutive time steps. While ROME is more noise-safe and independent of the dataset resolution, it strongly connects to the object size. Compared to SCAI and COP, ROME shows a lower variability along consecutive time steps and it is less sensitive to the proximity of objects. Despite these limitations, we employ these indices that have been applied before in our studies to retrieve comparable results. However, building an adapted methodology for assessing convective organisation may benefit future research."

- **L131**: Please state why these three indices were chosen over others

    **Author's response**: We aim to investigate organisation based on known metrics rather than designing a new index. Here, we chose the three indices that are easy to compute, comparable to previous studies, and adapted for the input data (radar reflectivity).

    **Author's changes in the manuscript**: Sect. 3.1, lines 211-215: "While there exist various organisation indices to quantify the spatial clustering, each index alone may not sufficiently characterise convective organisation (Stubenrauch et al., 2023). Instead, all indices have specific limitations, such as a sensitivity to the mean cloud area or to the number of individual objects. In response, we chose a combination of three organisation indices (SCAI, COP, ROME). All indices are designed to work on 2D data."

- **L139-140**: Please clarify how shifts in time and space occur and why they matter. Does the calculation of convective organisation indices take into account multiple time steps, or is the calculation independent per time step?

    **Author's response**: The shifts in space and time refer to the input data (i.e., when using multiple sources of imagery with a different resolution) and the variability between time steps (which may occur in our study). As we calculate the indices for each time step of 15 minutes, temporal shifts may occur and affect the results.

    **Author's changes in the manuscript**: See lines 250-252: "SCAI is insensitive to the size of the objects and mainly dominated by the variability in the number of clouds. However, it is less affected by shifts in time and space which induce high fluctuations of the index values, e.g, due to changes in the resolution of the input image

or between two consecutive time steps."

- **L152**: "continuous convective regions", a "convective region" has not been defined and this is unclear

  **Author's response**: In the original manuscript, we meant to describe detected cloud objects with at least one core. We change the text in the revised manuscript to be more clear.

  **Author's changes in the manuscript**: Sect. 3.1, lines 235-237: "The index considers the average size, proximity, and size distribution of convective clouds. Initially, it was designed to analyse radar observations. However, it also worked well with other data (Bläckberg and Singh, 2022). "

- **L177**: I suggest you keep consistent the ordering of the indices, which you introduced as (SCAI, COP, ROME) but here and in the figures list as (COP, SCAI, ROME). I would also suggest that, for readability, SCAI is not placed in the middle of the two that have the same direction for increasing organisation.

  **Author's response**: Thank you for your suggestion; we will change the indices to be listed as (SCAI, COP, ROME) throughout the manuscript.

  **Author's changes in the manuscript**: See, e.g., Sect. 4.1 lines 274f.: "This section analyses the spatial and temporal distributions of the three convective organisation indices: SCAI, COP, and ROME."

- **L178**: The phrasing of the statement about values of COP and ROME vs SCAI is confusing. I would recommend rephrasing along the lines of "By design, COP and ROME produce larger values for more organised domains, while SCAI instead results in smaller values"

  **Author's response**: Thank you for your suggestion, we will revise the text to be more clear.

  **Author's changes in the manuscript**: Lines 274-276: "Lower SCAI values (or higher COP and ROME values) are indicative of enhanced convective clustering, reflecting stronger spatial organisation. Conversely, high SCAI (low COP or ROME) values correspond to more scattered convective structures, implying weaker organisation (Biagioli and Tompkins, 2023)."

- **L184**: Can the metrics be compared to each directly in this manner?

  **Author's response**: We are aware that the metrics appear on different scales, i.e., ROME. However, we aim to compare rather their relative changes and the proportions of the distributions.

  **Author's changes in the manuscript**: We revised the description of Fig. 4 in lines 277-284: "Figure 4 (a) shows that SCAI values predominantly range between 0 and 1, with a peak concentration between 0.2–0.4. Oceanic regions have a slightly higher frequency of SCAI values lower than 0.4, whereas values higher than 0.4 are more common over land. This finding may suggest SCAI detects stronger convective organisation over water. COP values are mainly distributed between 0.1 and 0.75, with the highest density between 0.2–0.45. Over the ocean, values above 0.4 are more frequent, whereas over land, lower values dominate — again pointing to stronger convective organisation over the ocean (Figure 4, b). ROME displays a right-skewed distribution, with most values falling below 15,000. Differences between land and ocean are minor compared to SCAI or COP (Figure 4, c). Overall, the results may indicate a marginally stronger convective organisation over oceanic regions, with ROME showing the weakest land–sea contrast."

- **L18-188**: By "the diurnal cycle is opposed" are you referring to lower SCAI values meaning more organised as opposed to higher values for COP/ROME?

  **Author's response**: We aim to describe the opposed scaling of the metrics. We rewrite the text to be more clear.

  **Author's changes in the manuscript**: Please see Sect. 4.1, lines 274-276 for the interpretation of the indices' scales and lines 285-294 for a revised description of the diurnal cycle: "Figure 5 compares the diurnal cycle, changes to core numbers, and latitudinal averages of the indices over land and ocean within the 30° S–30° N domain. For SCAI, we find predominantely lower values over land throughout the day. The diurnal cycle exhibits minima between 09:00–12:00 UTC and 21:00–00:00 UTC, particularly over land. SCAI increases between 00:00–06:00 UTC and 12:00–21:00 UTC (Figure 5, a). COP shows a weaker temporal variability than SCAI but with values consistently suggesting higher organisation over the ocean (Figure 5, d). Diurnal variations in SCAI and COP reach up to 10 % of the indices' scales. ROME shows daytime (06:00–18:00 UTC) and nocturnal (00:00–03:00 UTC) peaks over land and mostly nocturnal peaks (21:00–06:00 UTC) over the ocean (Figure 5, g). Collectively, the indices indicate maximum convective organisation occurs over land in the afternoon and over the ocean in the night and early morning; minima occur at night over land and from noon to afternoon over the ocean."

- **L189**: (on diurnal cycle of organisation metrics): Yes, but isolated convection tends to only exist for a short period around the diurnal maximum, whereas more organised convection lasts for longer throughout the diurnal cycle, so it is not contradictory that observed convection is less organised during the diurnal maximum.
  **Author's response**: Thank you for that remark, we revise text in that section of the manuscript.
  **Author's changes in the manuscript**: Please see Sect. 4.1, lines 285-294, for the revised description of the diurnal cycle of the organisation metrics.

- **L190**: Most studies have found convective maximum over the ocean to occur during early morning hours, but the diurnal differences are much smaller than over land
  **Author's response**: We agree and revise the text.
  **Author's changes in the manuscript**: Please see Sect. 4.1, lines 285-294, for the revised description of the diurnal cycle of the organisation metrics.

- **L193-195**: and Figure 4 d,e,f: I interpret this as only small latitudinal differences in COP and ROME, excluding a peak in the south around -20 degrees, yet a large weakening in organisation reported by SCAI in the equatorial region. I don't see that the variability differences between land and ocean are significant. I would like the commentary to instead explain the strong weakening reported by SCAI, and whether this relates to data sensitivity due to the high number of clouds found in this region.
  **Author's response**: Thank you for your remarks, we revise the description in the text to be more clear. The weakening reported by SCAI near the equator may be due to the high number of clouds we have detected in this region (see Fig. 2 in the revised manuscript for the spatial distribution of cloud tracks). In contrast, we found considerably less clouds near the tropics. As SCAI is mainly dominated by the number of objects, the distribution of the cloud tracks may be responsible for the spatial variability of SCAI.
  **Author's changes in the manuscript**: Please find a revised version of Fig. 5 and its description in Sect. 4.1, lines 297-304: "Latitudinally, all indices show stronger organisation near the equator, although the spatial variability differs for the three indices. As SCAI is sensitive to object numbers, a higher frequency of detected clouds near the equator and less clouds near the borders of the AOI may contribute to the variability of the index (Figure 2, Figure 5, c). COP varies less with latitude, whereas we observe slightly higher values between $20°$ S–$20°$ N (Figure 5, f). For ROME, we find the highest variability between latitudinal averages and surface types with peaks over land between $20°$ S and the equator, and over oceanic regions near the equator and between $20°$–$30°$ S (Figure 5, i). Compared to other regions in the domain, the results show a considerably stronger convective organisation over the southern Atlantic Ocean ($30°$ S) for SCAI and ROME."

- **Fig 3**: It would be clearer to plot these distributions as proportions, rather than frequencies, to allow easier comparison between land and sea. Also, it would be good to increase the number of bins to show more detail in the distributions
  **Author's response**: Thank you for your suggestion to improve the figure. We add a revised version in the manuscript.
  **Author's changes in the manuscript**: See new Figure 4 containing the distribution (proportions) for SCAI, COP, and ROME (Sect. 4.1).

- **L198**: Can the convective core detection be used here to resolve this issue rather than calculating organisation purely from the MCS cloud shields?
  **Author's response**: We think this is an interesting suggestion which may help to resolve issues connected to clouds with a large shield. However, basing the calculation of the indices on the cores alone may increase temporal and spatial shifts of the indices as the tracked clouds are not required to have a core region during every time step of their life-cycle (as we also include clouds that show a reinvigoration after the initially present core regions dissolve). Nevertheless, we appreciate this idea to be interesting to try in a further step.
  **Author's changes in the manuscript**: -

- **L203**: How are the spatial densities calculated? Is it using a gaussian kernel estimation?
  **Author's response**: For the spatial densities we use a gaussian kernel estimation to derive densities for gridded data - which are the properties of the tracked clouds- interpolated on the grid with a resolution of $3°$ x $3°$. However, in the revised manuscript, we change the figures to contain frequencies instead of densities.
  **Author's changes in the manuscript**: We removed (old) Fig. 5 and 6 from the manuscript; instead please see (new) Fig. 6 for the average values of organisation indices, cloud properties, and core properties interpolated on

a 3° x 3° grid; the description of the figure can be found in Sect. 4.2 (Spatial patterns and statistical relationships of organisation indices), lines 306-324.

- **L204-205 and 210**: "isolated convective cells", "highly clustered systems" and "clustered systems", the link between the number of cores and the "clustering" is not defined or justified. The assumption is that one detected core means that the cloud is isolated, but this may not be the case. Further a system with multiple cores may be isolated. Could the authors provide additional detail in the text justifying the categories used here?

  **Author's response**: We agree the terminology may be confusing, we focus on improving the written expression to be more clear in the revised manuscript.

  **Author's changes in the manuscript**: We revised the description of spatial patterns in Sect. 4.2 (Spatial patterns and statistical relationships of organisation indices), lines 306-325. Here, we describe the properties more directly, e.g., lines 307-310: "Distinct regional patterns emerge across the AOI, highlighting potential links between convective organisation and cloud structure. Near the equator - particularly over continental Africa - higher SCAI values may coincide with a smaller cloud area, elevated cloud top height (CTH), and taller convective cores."

- **Fig 4**: The layout of the figures could be rotated to make comparisons clearer, e.g. change COP, SCAI and ROME values to be along one row each, and make all diurnal cycle plots one column etc. (i.e. current positions a,b,c would move to a,d,g and so forth)

  **Author's response**: Thank you for your suggestion to improve the figure, we add a revised version in the manuscript.

  **Author's changes in the manuscript**: Please see (new) Figure 5 for a revised version of (old) Figure 4.

- **Fig 4e** Why does SCAI reduce so much towards the northern and southern edges of the domain?

  **Author's response**: The indices values (SCAI, COP, and ROME) may be affected by the low number of cloud objects identified at the edges of the AOI (Fig. 2). In these regions we find less but larger clouds that may affect the indices.

  **Author's changes in the manuscript**: We changed Figure 5 to contain bar plots for showing differences in the spatial distribution between organisation indices over land and sea. A description of the distribution of cloud tracks can be found in Sect. 2.5, a discussion of the results in Sect 5.3, lines 589-591: "The indices themselves are sensitive to cloud object count (SCAI) or area (COP, ROME), which may affect spatial patterns, especially since equatorial convective clouds tend to be smaller and more frequent than those near the tropics (Section 2.5, Section 3.1)."

- **L214** Should this read less than 1000km2? The description of how systems are binned by area is clearer in the caption of figure 5 than in the text.

  **Author's response**: The sign got reversed here and should point to a cloud area that is less than 1000km2.

  **Author's changes in the manuscript**: See above, we removed the figure and revised the description of the spatial distribution in Sect. 4.2.

- **L214**: why define moderate cloud area as (mean, 10 x mean)? The skewed distribution of anvil cloud area results in a mean that is greater than the median, so the majority of clouds will end up in the smallest category.

  **Author's response**: We agree and modify the section in the manuscript to omit these figures.

  **Author's changes in the manuscript**: See above, figure was removed in the revised manuscript.

- **L222-223**: "we generally find highly clustered systems to be accompanied by a larger cloud anvil size and vice versa", this claim not supported. These plots mainly show that most of the tracked clouds occur in the equatorial region. Also the number of cores a cloud has has not been directly compared with the cloud area, which would be interesting to include in the discussion.

  **Author's response**: We agree and revise Sect. 4 to include more direct analyses for the relationship between organisation indices and cloud/core properties.

  **Author's changes in the manuscript**: Please see Sect. 4.2, lines 325-335 for the relationship between organisation indices and cloud/core properties: "To quantify the relationship between organisation indices and cloud properties, we compute Spearman's rank correlation coefficient R using data from all cloud tracks (Figure 7). The logarithmic distributions reveal a general skew toward low values for SCAI, ROME, cloud area, lifetime, number of cores, and core area. The correlation analysis shows that COP and ROME may be positively associated with cloud

area, lifetime, CTH, number of cores, and core height (Figure 7, g–r). In contrast, SCAI is negatively correlated with all of these properties except for CTH and the core height (Figure 7, a–f). For the core area, we see a weak negative correlation to all indices. The findings suggest that stronger convective organisation may be statistically linked to larger, longer-lived cloud systems, a higher CTH and core height, and more cores. Interestingly, these statistical relationships contrast with some spatial patterns in Figure 6. For instance, while higher ROME values spatially co-occur with smaller clouds and shorter lifetimes in some regions, correlation coefficients suggest that, overall, organisation increases with cloud area and duration. However, most correlations are weak, with maximum coefficients around 0.26 between ROME and the cloud lifetime. They highlight the complex and regionally variable nature of these relationships." and Sect. 4.4.2, lines 415-452 for a comparison of the relationship organisation indices and cloud/core properties, and cloud and core properties in the whole dataset, for the 10 % strongest convective organisation, and the 10 % weakest convective organisation.

- **Figure 6**: I interpret this figure differently. I find the differences between the distribution of small and medium cloud areas to be small, and not very meaningful. I think the main result here is that the largest clouds predominantly occurred over land, and not in the equatorial belt. This may also relate to the number of samples in this partition, which should be stated.
  **Author's response**: In the revised manuscript, we remove this figure. Instead, we focus on plotting the average values of cloud and core properties and their connection to the organisation indices more directly.
  **Author's changes in the manuscript**: Please see Sect. 4.2 for a revised description of spatial patterns of cloud/core properties in Figure 6 (lines 306-324).

- **L224**: "convective activity", not defined or assessed. Could the estimated radar reflectivity be used to provide a measure of convective intensity?
  **Author's response**: In our study, we used the predicted radar reflectivity to detect convective clouds - hence, it may provide a measure of convective intensity. However, we revise the manuscript to address the extracted cloud and core properties more directly.
  **Author's changes in the manuscript**: The description of (new) Fig. 6, which contains the spatial distribution of average cloud/core properties, can be found in lines 306-324.

- **L225-226**: I disagree partially with this description. A "reduced cloud lifetime, "enhanced area ratio" and "fewer DCCs" is not characteristic of the equator region, but rather characteristic of most of the domain when compared against the large anomalous region over the Atlantic (15S 0E)
  **Author's response**: We revise the description of figure to be more clear.
  **Author's changes in the manuscript**: As we found an error in the data, which possibly caused the large anomalous region in the southern hemisphere, we include an updated version of the figure (see Figure 6 and Figure 9). For a revised description of Figure 6, please see lines 306-324.

- **L227**: I don't think this reference is really appropriate here, as only SCAI reported less organisation over the equator in your dataset?
  **Author's response**: We revise the text in that section of the manuscript.
  **Author's changes in the manuscript**: We revised the text and removed or updated references where beneficial. Please see lines 306-324 for a revised version.

- **L229-230**: The reference here is unnecessary without a link between your results and the previous study.
  **Author's response**: We revise the text in that section of the manuscript.
  **Author's changes in the manuscript**: See above. We revised the text and checked the references used in the section. Please see lines 306-324 for a revised version.

- **L250**: Why is the number of systems included in the 10th and 90th percentile bins unequal?
  **Author's response**: Thank you for pointing this out. We checked the code and found an error while filtering the subsets. In the revised version of the manuscript, P90 and P10 contain the same number of samples (which, however, may vary when filtering between seasons or surface types due to the imbalance of detected clouds over land/sea and MAM/JJA).
  **Author's changes in the manuscript**: Please see Sect. 4.4, lines 379-392 for the description of the percentile-based subsets of strong/weak convective organisation: "To identify regional patterns of convective organisation and their effects on cloud properties, we adopt a percentile-driven approach. There exist no universally defined

thresholds to distinguish between weak and strong convective organisation. In response, we compute the 10th, 25th, 75th, and 90th percentiles based on the distribution of each organisation index (SCAI, COP, and ROME) using the cloud tracks between March to August 2019 (Table 5). These percentiles serve as thresholds to classify the data into subsets of weak and strong convective organisation, as induced by the interpretation of the indices: strong organisation may be related to low SCAI and high COP/ROME, weak organisation to high SCAI and low COP/ROME (Biagioli and Tompkins, 2023; Semie and Bony, 2020). Following, regions of strong convective organisation are defined as cloud tracks with an index value below the 10th percentile for SCAI or above the 90th percentile for COP and ROME. Conversely, regions of weak organisation correspond to values that lie above the 90th percentile for SCAI or below the 10th percentile for COP and ROME. To identify spatial and temporal patterns of convective organisation, we create two subsets from all data points in the dataset, whereas one represents the 10 % strongest convective organisation (Q10 for SCAI; Q90 for COP and ROME, hereafter: P90), and the other representing the 10 % weakest convective organisation (Q90 for SCAI; Q10 for COP and ROME, hereafter: P10). These may represent so-called "hotspots". We also define the interquartile range (IQR, values between the 25th–75th percentile) to represent a baseline, which is used to contrast the spatial distribution of average organisation against the identified hotspot regions."

- **L256**: The relationship between number of convective cores and convective organisation could be shown much more clearly by showing a scatter plot of the two to show how correlated they are, rather than comparing spatial distributions
  **Author's response**: Thank you for that suggestion, we include a 2D histogram and compute the Spearman correlation coefficient in the revised manuscript to visualise the relationship.
  **Author's changes in the manuscript**: Please see the 2D histogram in Sect. 4.2 (Figure 7) showing the relationship between cloud/core properties and convective organisation. You may find its description in lines 325-335: "To quantify the relationship between organisation indices and cloud properties, we compute Spearman's rank correlation coefficient R using data from all cloud tracks (Figure 7). The logarithmic distributions reveal a general skew toward low values for SCAI, ROME, cloud area, lifetime, number of cores, and core area. The correlation analysis shows that COP and ROME may be positively associated with cloud area, lifetime, CTH, number of cores, and core height (Figure 7, g–r). In contrast, SCAI is negatively correlated with all of these properties except for CTH and the core height (Figure 7, a–f). For the core area, we see a weak negative correlation to all indices. The findings suggest that stronger convective organisation may be statistically linked to larger, longer-lived cloud systems, a higher CTH and core height, and more cores. Interestingly, these statistical relationships contrast with some spatial patterns in Figure 6. For instance, while higher ROME values spatially co-occur with smaller clouds and shorter lifetimes in some regions, correlation coefficients suggest that, overall, organisation increases with cloud area and duration. However, most correlations are weak, with maximum coefficients around 0.26 between ROME and the cloud lifetime. They highlight the complex and regionally variable nature of these relationships" and the description of the correlation matrix comparing all cloud tracks and the percentile-based subsets P90 and P10 in Sect. 4.4.2, lines 415-431.

- **L258**: Figure 8e shows core size to decrease near the equator in a large region
  **Author's response**: We revise Section 4 including the figure and its description. The new version shows that the number of cores and the core area decrease over the equator during summer, whereas we detect an increase in large parts of near-equator regions, in particular > 5°N and < 5°S.
  **Author's changes in the manuscript**: Please see Sect. 4.3, lines 351-367 for a revised description of the changes between March-May and June-August.

- **L264**: The patterns in figures 9 and 10 appear much more complex than discussed here. In particular, ROME appears very different during JJA, and appears to show the opposite locations for more organised convection than COP. Why is this?
  **Author's response**: We add an updated version of the figures in the revised manuscript. Here, we see that patterns of ROME are more similar to SCAI and COP. Moreover, we add a detailed description of the patterns.
  **Author's changes in the manuscript**: See Sect. 4.4.3, lines 454-484 for the detailed description of observed spatial patterns for the 10th and 90th percentiles of SCAI, COP, and ROME.

- **L269**: The description of how the convective indices are aggregated is unclear, particularly given the differences between them shown in fig. 10
  **Author's response**: Figure 11 displays the locations of the percentiles of all three indices which are used to

identify hotspot regions. For that purpose, we filter the data set with all cloud tracks to receive two subsets of the 10 % index values associated to weakest (P90 for SCAI and P10 for COP and ROME) and strongest (P10 for SCAI and P90 for COP and ROME) convective organisation. Each subset contains the data of all three indices and is used to assess the average distribution of weak and strong organisation. We add a more detailed description in the revised manuscript.

**Author's changes in the manuscript**: See above (Sect. 4.4, lines 379-392) for the description of the percentile-based subsets. The results for the hotspot regions - examined by the spatial distribution for clouds in the two subsets of the 10 % strongest (P90) and the 10 % weakest (P10) convective organisation may be found in Sect. 4.4.4, lines 486-510.

- **L270**: "averaged of the AOI", I believe you mean "averaged zonally"
  **Author's response**: Thank you for your remark, we change the manuscript to be more clear.
  **Author's changes in the manuscript**: Please see Sect. 4.4.4, lines 498f.: "Across the belt from 15° N to 15° S, the frequency of the 10 % weakest convective organisation is generally high (Figure 16, b)."

- **L272**: It would be clearer to show this directly. An additional plot, plotting the average of different cloud properties against organisation would show whether this is the case. In addition, this statement refers to vertical cloud core properties, but only CTH has been assessed. It would be interesting to see how e.g. convective core height and anvil height vary as well
  **Author's response**: We add additional analyses to describe the relationship between organisation indices and cloud/core properties. Moreover, we change the analysis to include the core height instead of area ratio.
  **Author's changes in the manuscript**: Sect. 2.4, Table 3, we replace "area ratio" by "core height". Moreover, we include new analyses to investigate the relationships and statistical differences between all cloud tracks, the 10 % strongest (P90), and the 10 % weakest (P10) convective organisation: Fig. 7 for the 2D histogram organisation indices vs. cloud/core properties (Sect. 4.2), Fig. 12 for the correlation matrix for all tracks, P90, and P10 (Sect. 4.2.2.), and Fig. 13/ Table 6 for the effect sizes between properties associated to all tracks, P90, and P10 (sect. 4.2.2).

- **L273-274**: The claim here is not well supported, it would be clearer to show this directly. An additional plot, plotting the average of different cloud properties against organisation would show whether this is the case
  **Author's response**: Thank you for the suggestion; we add new analysis in the revised manuscript which describe the parameter's relationship more directly.
  **Author's changes in the manuscript**: Please see the comment above for the additional analysis in the revised manuscript. Based on these analyses, we summarize our key findings Sect. 5.1, lines 523-539: "Correlations between convective organisation indices (SCAI, COP, ROME) and cloud/core properties suggest generally weak to medium relationships for all cloud tracks. SCAI is negatively correlated to the properties, except for the CTH and core height, while COP and ROME show positive correlations except for the core area. In all cases, the coefficients remain below 0.3 (Section 4.2). These correlations partly change for the 10 % strongest and 10 % weakest convective organisation. Within these subsets, we find the highest corelation coefficient between SCAI and the CTH for clouds with a strong convective organisation. However, the relationships between the cloud and core properties remain similar over all subsets (Section 4.4.2). In contrast, we observe pronounced changes in both the indices and the associated cloud characteristics along the period and across the AOI. These changes reflect a high variability for average values, though the correlation strength remains limited (Section 4.3). The distribution of cloud and core properties within identified hotspot regions differ from those observed in the full dataset of all cloud tracks. We analyse the effect size using Cohen's D to reveal how organisation strength may influence cloud characteristics. Compared to all cloud tracks, the cloud systems of the 10 % strongest organisation tend to have larger cloud and core areas, a lower CTH and core height, a shorter lifetime, and a lower number of convective cores. In contrast, weaker convective organisation may be typically associated with smaller clouds and larger cores, fewer cores, shorter lifetimes, and lower vertical extent. Strong convective organisation differs the most from all cloud tracks regarding the CTH and from weak organisation regarding the cloud lifetime. Between weak convective organisation and all cloud tracks, we identify the cloud area to have the highest effect size (Section 4.4.2). Hence, the cloud area appears to be more important to identify weak convective organisation, whereas strong convective organistaion may be stronger driven by the CTH."

- **L276**: is "MCS" here referring to overall organisation (which is inconsistent with earlier usage)? If not, statement is not supported as no assessment of the cloud types in this regime has been performed

**Author's response**: Refers to clouds with a large cloud area; we revised the written expression to be more clear.
**Author's changes in the manuscript**: Sect. 4.4.4 (lines 486-510) for the description of the spatial patterns. We move the analysis of these findings to Sect. 5.2, e.g., lines 542-550: "Our results show that convective organisation tends to be stronger for cloud properties typically associated with large convective systems containing multiple core regions, such as MCSs (Stubenrauch et al., 2023). In line with Brüning and Tost (2025), we observe that cloud area, lifetime, cloud top height (CTH), core area, and core height all grow with the number of convective cores (Figure 12). While multiple cores may enhance cloud longevity, promote cloud area growth, and strengthen vertical updrafts, the number of cores may also be a key factor in determining the strength of convective organisation. Interestingly, our findings contrast with Takahashi et al. (2017), as we observe stronger convective organisation - reflected in higher COP and ROME values and lower SCAI values - more frequently over the ocean. Over continental Africa, spatial patches of weak convective organisation appear in both seasons (Section 4.2, Section 4.4.4). However, the difference between land and ocean remain small and may partly stem from an uneven distribution of cloud tracks (Figure 2)."

- **L297**: I wouldn't describe this as a linear decrease as it appears more like a rapid drop from May to June
  **Author's response**: We agree and revise the description of the figure.
  **Author's changes in the manuscript**: Please see Sect. 4.3 for temporal variations of organization indices and cloud/core properties, e.g., lines 336-350: "The previous analysis suggests the overall correlation between convective organisation indices and cloud/core properties is generally weak. In this section, we aim to capture changes in convective behaviour along the period that may help to explain observed patterns. For this purpose, we filter the dataset into two subsets between March to May (MAM, n = 212,984) and June to August (JJA, n = 141,089). Here, we analyse monthly means over land and ocean (Figure 8). Overall, differences between land and ocean typically span up to 10 % of each index's dynamic range (Figure 4). For the monthly changes, most variables do not exhibit a linear trend. SCAI, COP, and the number of cores remain relatively stable, while the CTH and core height vary non-monotonically (Figure 8, a, b, e, g, i). SCAI generally decreases over the ocean and increases slightly over land until June, returning to near-March values by August (Figure 8, a). COP displays similar changes over land, while over the ocean, it increases marginally throughout the period (Figure 8, b). ROME exhibits the strongest variability, increasing over both surface types, especially over the ocean (Figure 8, c). Notably, average CTH, cloud lifetime, and core height are consistently higher over land, whereas cloud and core areas are larger over the ocean, particularly from May to August (Figure 8, d–f, h–i). The number of cores remains fairly constant across the time series (Figure 8, g). Over the ocean, we observe a steady increase in cloud and core area and a decrease in CTH. Core height peaks in May and July, followed by a decline in August. Over land, temporal changes are less pronounced, though the core area shows a slight dip until May and then rises again by August."

- **L299**: I don't think the trends are clear enough to say that there is a relative increase, just that organisation tends to be higher over the ocean
  **Author's response**: We add a revised description of the seasonal changes in Sect. 4.2 of the manuscript.
  **Author's changes in the manuscript**: See the comment above (lines 336-350) for a revised description of monthly changes along the period (March-August).

- **Figure 13**: Unclear that the percentiles refer to organisation metrics. Are the percentiles calculated overall or individually for each month? If the latter, then this could cause differences not because the DCC properties are changing for a given level of organisation, but because the average organisation is changing and hence the percentiles are at different organisation values
  **Author's response**: The percentiles are computed along the whole time series, not on a monthly base. We update the figure and its description in the revised manuscript in Sect. 4.4.
  **Author's changes in the manuscript**: We removed old Figure 13, instead please find monthly changes of cloud/core properties together with changes of the organisation indices in Figure 8, lines 328-341.

- **L303**: Number of DCCs doesn't appear to trend over time, while core size increases except for SH-P90.
  **Author's response**: In most cases, variability between the months is indeed higher than any linear trend, we revise the text to be more clear.
  **Author's changes in the manuscript**: See Sect. 4.3, lines 336-350 for an updated description of monthly changes of the organisation indices and cloud/core prperties.

- **L315**: While Welsh's t-test can better handle distributions with different sizes and variances, it does not account for skewness in the distributions.
  **Author's response**: Thank you for that remark, we change the text in the revised manuscript.
  **Author's changes in the manuscript**: Sect. 4.4.2, lines 432-434: "To assess whether differences between datasets are statistically significant, we compare parameter distributions for all cloud tracks, P90, and P10 subsets. We apply Welch's t-test, which may be more robust for unequal sample sizes (Derrick and White, 2016)."

- **Figure 14**: It's difficult to interpret how meaningful these results are. As they only represent the 90th percentile, it is difficult to see how much they differ from the general population of DCCs. It would be helpful to plot the distributions of all observed systems, as well as the 90th percentiles, to show how the properties of the most organised systems differ
  **Author's response**: We revise the figure and add new analyses in Sect. 4.4 which focus on a more direct comparison between all cloud tracks, the strong convective organisation, and weak convective organisation.
  **Author's changes in the manuscript**: See Sect. 4.4.2 (Figs. 12-13 and Table 6), lines 415-452, which may show how the properties of the most and least organised systems differ from all cloud tracks.

- **L352**: There was not much discussion of the core/area ratio.
  **Author's response**: We agree and replace the area ratio by the core height to analyse vertical core properties.
  **Author's changes in the manuscript**: See Sect. 2.4 (and Table 3), lines 176-184: "We use the labelled cloud masks to extract cloud and core properties at each point in time. Moreover, we compute average properties across the cloud's lifetime to derive distinct key properties that may characterise the trajectory. These properties include the cloud lifetime, cloud area, cloud top height (CTH), number of cores, and mean core area and height (Table 3). The cloud area is computed from the column-wise maximum horizontal extent of the 3D cloud mask, while CTH is derived from the vertical extent. For the cloud lifetime, we extract the time (in hours) between the first and last detection of each trajectory of the labelled pixels. Surface type is assigned via a binary land-sea mask and the modal value for the locations of the cloud trajectory within this land-sea mask. For clouds with one or more cores, we count the maximum number of cores associated to the trajectory. Moreover, the core area and height are derived from the column-wise maximum horizontal extent and vertical extent of the previously identified cores, similar to the cloud area and CTH."

- **L361**: The claim here is that grouping by organisation allows significant differences to be shown between different regions and different seasons, but are the differences statistically significant for all levels of organisation?
  **Author's response**: We add new analyses in Sect. 4.4 to compare differences between these levels of organisation (Fig. 11-13, Table 6).
  **Author's changes in the manuscript**: See Sect. 4.4.2 for a statistical analysis of the differences between all cloud tracks, the most organised 10 % systems, and the least organised 10 % systems (lines 415-452). In the revised manuscript, we focus less on hemispheric/seasonal differences, instead we show the relationship between organisation and cloud/core properties. A discussion of these results may be found in Sect. 5.1 (Summary of key findings) and Sect. 5.2 (Spatio-temporal drivers of organisation).

- **L366**: Agreed, and I believe with your dataset you have the opportunity to progress this, which would greatly improve the value of this and future works. For instance, by focusing on more considered comparison between cloud properties and the organised sate (as in Figure 13), or by considering key synoptic or flow regimes separately rather than the somewhat arbitrary partitioning along hemisphere or latitude, or by making better use of 4D radiances with vertical cloud and core properties and development information. What additional properties would be useful for future studies?
  **Author's response**: Thank you for your detailed suggestions on how to improve our manuscript. Instead of centering the analyses on hemispheric and seasonal differences, we focus the revised manuscript to address the relationship between organisation indices and cloud and core properties more directly. The new analyses can be found throughout Sect. 4 (Results). While we use only properties derived from the predicted radar reflectivites, analysing further cloud bulk properties (e.g., vertical mass flux, the excess temperature, or the water content) or cloud radiative properties (e.g., outgoing longwave radiation) may be interesting for the analysis. However, we consider the quantification of convective organisation to be of uttermost importance to reduce uncertainties connected to current approaches.
  **Author's changes in the manuscript**: Based on all reviewers suggestions, we add new analyses to analyse the

relationship between between cloud properties and organisation indices for all cloud tracks (Figure 7), to identify changes of these relationships along the depicted period (Table 4), and to compare different organisational states (differentiating between all cloud tracks, the 10 % most organised, and the 10 % least organised clouds, Figures 11-13). With these analyses (which may be found in Section 4.2, 4.3, and 4.4.2), we aim to assess how cloud/core properties affect convective organisation. In Sect. 5.1 and Sect. 5.2, you may find a discussion of our findings, e.g., lines 531-540: "The distribution of cloud and core properties within identified hotspot regions differ from those observed in the full dataset of all cloud tracks. We analyse the effect size using Cohen's D to reveal how organisation strength may influence cloud characteristics. Compared to all cloud tracks, the cloud systems of the 10 % strongest organisation tend to have larger cloud and core areas, a lower CTH and core height, a shorter lifetime, and a lower number of convective cores. In contrast, weaker convective organisation may be typically associated with smaller clouds and larger cores, fewer cores, shorter lifetimes, and lower vertical extent. Strong convective organisation differs the most from all cloud tracks regarding the CTH and from weak organisation regarding the cloud lifetime. Between weak convective organisation and all cloud tracks, we identify the cloud area to have the highest effect size (Section 4.4.2). Hence, the cloud area appears to be more important to identify weak convective organisaion, whereas strong convective organistaion may be stronger driven by the CTH. Despite these differences in the distribution of cloud and core properties, we detect partly the same direction for correlations in case of strong and weak convective organisation, highlighting the complexity of involved processes"

- **L384-385**: To achieve this, I recommend also considering the above in your analysis work. There were a few missed opportunities in this, mainly descriptive, work. Using a better measure of organisation alone will not elucidate the relationships you seek. Further, assessment of the tracked cloud distribution and sample size in space and time (diurnal and seasonal) will help to clarify which of your results are most robust, and which are subject to most uncertainty in the organisation metrics.
  **Author's response**: Thank you for your suggestions. We agree that an adaptive measure of organisation alone will not solve the issues; however, our aim is to apply known measures and to analyse their results to the 3D cloud and core properties in our dataset. To discuss the robustness of the findings, we add a more detailed description of the cloud tracks in Sect. 2.5 and revise the discussion about limitations and uncertainties in Sect. 5.3.
  **Author's changes in the manuscript**: Please see Sect. 2.5 (Fig. 2), lines 191-199, and Sect. 4.4.1, lines 395-413, for the spatial and temporal distribution of cloud tracks (for all clouds, and percentile-based subsets). Limitations are discussed in Sect. 5.3, lines 571-581, e.g.: "Moreover, addressing the imbalance between land and ocean cloud occurrences could strengthen the robustness of our findings. Currently, the cloud track distribution is skewed, with a heavy concentration near the equator."

**4 Technical corrections**

- **L84**: both "Section" and "Sect." are used in this passage. Recommend "sec." as an acronym for section instead
  **Author's response**: The usage of "Section" and "Sect." follows the manuscript composition guideline of ACP (Citation: "The abbreviation "Sect." should be used when it appears in running text and should be followed by a number unless it comes at the beginning of a sentence").
  **Author's changes in the manuscript**: Based on the ACP guideline, we suggest the text in Sect. 1, lines 78-79 remains as follows: "We have divided this article into five further sections. In Sect. 2, we describe the dataset used in this study. Section 3 presents an overview of metrics employed to quantify convective organisation."

- **L143**: COP accounts for the areas of both objects i and j as per eq. 2
  **Author's response**: Thanks for the correction.
  **Author's changes in the manuscript**: Please see Sect. 3.1, lines 228-230: "COP uses the number of objects (N), the area of the i-th object (Ai) and the j-th object (Aj), and the distance between the centroids of the i-th and the j-th object (dij). It adds the characteristic domain size (L) and the total image size (L2)."

- **L186**: "zonal changes" should be "zonal mean changes" or "latitudinal variations" or similar
  **Author's response**: Thank you for your comment, we change the text.
  **Author's changes in the manuscript**: Sect. 4.1, lines 297f.: "Latitudinally, all indices show stronger organisation near the equator, although the spatial variability differs for the three indices."

- **L210**: typo "custered" -> clustered
  **Author's response**: We correct that typo.
  **Author's changes in the manuscript**: Removed sentence from manuscript.

---

## Referee Report (RR1)

Review of "A machine learning-based perspective on deep convective clouds and their organisation in 3D. Part II: Spatial-temporal patterns of convective organization"
By Sarah Brüning and Holger Tost
MS No.: egusphere-2025-376

**Recommendation**
Minor revisions

**General comments**
The reviewer appreciates the authors' efforts to improve the manuscript. The previous comments primarily concerned clarifications of the methodology, interpretations, and descriptions. The authors have substantially revised the manuscript, and the reviewer finds that the earlier concerns have been adequately addressed.

For this second review, the reviewer again provides comments focused on clarification, as the manuscript has undergone significant modifications since the previous version. The reviewer considers the paper acceptable for publication, provided that the points specified below are adequately addressed.

**Specific comments**
L5-7, "Our analysis emphasises that the most important distinction between all detected clouds and strong convective organisation may relate to a larger cloud area, a lower cloud top and core height, and a shorter lifetime.Weak. Weak convective organisation tends to occur with smaller clouds with fewer cores, and a shorter lifetime.":
The first sentence is unclear about which characteristics apply to "all detected clouds" versus "strong convective organisation". It is also unclear whether the two sentences are consistent with each other. Please revise to clarify the intended comparison.

L10, "over the remote Atlantic Ocean": The phrase is ambiguous. Please specify the geographical extent or latitudes of the area referred to.

L30: The most standard reference to "self-aggregation of convection" is

Bretherton, C. S., Blossey, P. N., & Khairoutdinov, M. (2005) An energy-balance analysis of deep convective self-aggregation above uniform SST. Journal of the Atmospheric Sciences, 62(12), 4273–4292. https://doi.org/10.1175/JAS3614.1

The authors should also note the earlier reference to the radiative convective equilibrium simulation than Held et al. (1993) is

Nakajima, K., & Matsuno, T. (1988) Numerical experiments concerning the origin of cloud clusters in the tropical atmosphere. Journal of the Meteorological Society of Japan, 66(2), 309–329. https://doi.org/10.2151/jmsj1965.66.2_309

Caption of Figure 1, The first line: "we show how the derive a contiguous 3D cloud field from 2D data by a machine learning-based extrapolation" should be "⋯ how to derive ⋯"

L147, "To reduce noise, we first apply a Gaussian filter with a sigma value of 0.5 to smooth the input data.": Please describe the spatial scale of the Gaussian filter. What effective scale is used for the analysis?

L181, "land-sea mask": Contiguous convective clouds consist of multiple points and may generally include both land and sea grid points. How can the land or sea type of clouds be classified?

L188, "at least 15 min": Does this require at least two time steps (i.e., more than 15 minutes), or is a single snapshot (i.e., 15 minutes) sufficient?

L195, "Approximately 75 % of cloud tracks occur over ocean, with land-based tracks comprising the remaining 25 %": What is the ratio of land and sea areas within AOI? Does the ratio 75% larger than the area fraction of sea within AOI?

Figure 4: It is difficult to read from the figure whether the contribution over land is larger or smaller than that over the sea. Moreover, it is unclear whether the "all" category represents the sum of the land and sea contributions. Could you please clarify what "all" refers to?

L352-368: It is unclear whether the "differences" mentioned in line 352 refer to the values of JJA minus those of MAM. The description in this paragraph does not appear to be consistent with Fig. 9. For example, the statement "ROME displays

more complex behavior along the period, with an overall increase around the AOI in JJA" (lines 356–358) seems inconsistent with Fig. 9(c), which clearly shows an increase in ROME across the entire domain.

L352: The term "summer" is not an appropriate abbreviation to refer to JJA. "Boreal summer" or simply "JJA" is more suitable.

L369: Please refer to "Table 4" at the beginning of the paragraph.

L378, "additional regional factors": What factors would affect the results? The large-scale circulation is an example.

L405, "more common": The contribution over land is smaller than that over the ocean in all cases. The phrase "more common" is unclear and may be misleading.

L598: Please spell out "DBSCAN" and "HDBSCAN" and provide appropriate references for both.

---

## Author Response (AR2)

**Final response - minor revisions**
**(Manuscript "EGUSPHERE-2025-376")**

Sarah Brüning and Holger Tost

July 2, 2025

**1   Introduction**

We would like to thank the reviewer again for providing insightful feedback. In this response, we aim to address all raised concerns. For every comment, you can find (1) the comment, (2) the author's response (both with lines from submitted manuscript), and (3) the author's changes in the revised manuscript. The lines given for **author's changes in the manuscript** refer to the lines in the **revised manuscript**.

**2   General comments**

- **Comment**: The reviewer appreciates the authors' efforts to improve the manuscript. The previous comments primarily concerned clarifications of the methodology, interpretations, and descriptions. The authors have substantially revised the manuscript, and the reviewer finds that the earlier concerns have been adequately addressed. For this second review, the reviewer again provides comments focused on clarification, as the manuscript has undergone significant modifications since the previous version. The reviewer considers the paper acceptable for publication, provided that the points specified below are adequately addressed.
  **Author's response**: Thank you for your feedback on our revised manuscript and our proposed modifications. We strive to clarify all points raised in your review.

**3   Specific comments**

:

- **L5-7**: L5-7, "Our analysis emphasises that the most important distinction between all detected clouds and strong convective organisation may relate to a larger cloud area, a lower cloud top and core height, and a shorter lifetime.Weak. Weak convective organisation tends to occur with smaller clouds with fewer cores, and a shorter lifetime.": The first sentence is unclear about which characteristics apply to "all detected clouds" versus "strong convective organisation". It is also unclear whether the two sentences are consistent with each other. Please revise to clarify the intended comparison.
  **Author's response**: We agree that these sentences may seem unclear and revise the text.
  **Author's changes in the manuscript**: Lines 5-8: "Our findings highlight how cloud properties may interact with organisation. Hence, strong organisation tends to occur with larger cloud areas, lower cloud tops and core heights, and shorter lifespans compared to the average convective system. In contrast, weak organisation may be associated with smaller clouds, fewer cores, but similarly shorter lifespans."

- **L10**: L10, "over the remote Atlantic Ocean": The phrase is ambiguous. Please specify the geographical extent or latitudes of the area referred to.
  **Author's response**: The sentence should refers to Atlantic Ocean south of 15°S. We revise the text.
  **Author's changes in the manuscript**: Lines 10-11: "From March to May, patches of strong convective organisation emerge along the African coastlines and over the southern Atlantic Ocean."

- **L30**: L30: The most standard reference to "self-aggregation of convection" is Bretherton, C. S., Blossey, P. N., & Khairoutdinov, M. (2005) An energy-balance analysis of deep convective self-aggregation above uniform SST. Journal of the Atmospheric Sciences, 62(12), 4273–4292. The authors should also note the earlier reference to the radiative convective equilibrium simulation than Held et al. (1993) is Nakajima, K., & Matsuno, T. (1988) Numerical experiments concerning the origin of cloud clusters in the tropical atmosphere. Journal of the Meteorological Society of Japan, 66(2), 309–329.
  **Author's response**: Thank for your comment, we will revise the text to contain additional, earlier references.
  **Author's changes in the manuscript**: Sect. 1, lines 29-34: "While convective organisation is difficult to quantify in observational data, idealised model configured in radiative-convective equilibrium (RCE) could demonstrate a large-scale clustering of convective clouds which is known as self-aggregation of convection (e.g, Nakajima and Matsuno (1988); Held et al. (1993); Wing et al. (2017)). Following Bretherton et al. (2005), it occurs on a timescale between days and weeks and describes the transition of an approximately random distribution of convective cells into convecting and non-convecting regions that grow upscale over time."

- **Figure 1**: Caption of Figure 1, The first line: "we show how the derive a contiguous 3D cloud field from 2D data by a machine learning-based extrapolation" should be "... how to derive ..."
  **Author's response**: Thank you for your correction, the text will be changed.
  **Author's changes in the manuscript**: Figure 1: "In (a), we show how to derive a contiguous 3D cloud field from 2D data by a machine learning-based extrapolation (Brüning et al., 2024)."

- **L147**: L147, "To reduce noise, we first apply a Gaussian filter with a sigma value of 0.5 to smooth the input data.": Please describe the spatial scale of the Gaussian filter. What effective scale is used for the analysis?
  **Author's response**: Following the definition of the Gaussian image filter, we employ an effective scale that comprises half a standard deviation (sigma = 0.5) for the given input image (across the AOI of 1200 x 1200 x 90 pixels), containing the 3D radar reflectivity field with values between -25 to 20 dBZ.
  **Author's changes in the manuscript**: Sect. 2.3, lines 147-149: "To reduce noise, we first apply a smoothing Gaussian image filter with an effective scale of half a standard deviation (sigma = 0.5) on the 3D radar reflectivity field."

- **L181**: L181, "land-sea mask": Contiguous convective clouds consist of multiple points and may generally include both land and sea grid points. How can the land or sea type of clouds be classified?
  **Author's response**: We are aware that clouds may travel across both land and sea grid points along their life-cycle. Although it may not account for these real world conditions, we apply a simplifying approach to reduce the complexity within the dataset. For this purpose, we use the modal location of the grid points (either over land or sea) to classify the whole cloud. We note this classification displays only an approximation. We revise the description in this paragraph to be more clear.
  **Author's changes in the manuscript**: Sect. 2.4, lines 183-185: "Each cloud track is classified as either marine (sea) or continental (land) using a binary land-sea mask. For this purpose, we determine the most frequent (modal) surface type across all grid points along the cloud trajectory. While this method does not capture changes in surface type throughout the cloud's life-cycle, it may provide insights on the effect of the most frequently occurring surface type."

- **L188**: L188, "at least 15 min": Does this require at least two time steps (i.e., more than 15 minutes), or is a single snapshot (i.e., 15 minutes) sufficient?
  **Author's response**: Refers to at least one time step of 15 minutes which is required for the occurrence of cores, and the CTH/CBH/reflectivity thresholds. We rephrase the sentence to be more clear.
  **Author's changes in the manuscript**: Sect. 2.5, lines 190ff.: "We filter the cloud trajectories to exclude possibly non-convective tracks from the analysis. For that purpose, we employ three criteria occurring for at least a single timestep of 15 minutes: (a) One or more core regions, (b) radar reflectivity of higher than 0 dBZ at 10 km height, and (c) minimum CTH of 10 km and maximum CBH of less than 5 km."

- **L195**: L195, "Approximately 75% of cloud tracks occur over ocean, with land-based tracks comprising the remaining 25 %": What is the ratio of land and sea areas within AOI? Does the ratio 75% larger than the area fraction of sea within AOI?
  **Author's response**: Thank you for your comment; the ratio of land / sea within the AOI comprises about 35% to 65%, pointing out a shift of about 10 % towards marine convection, compared to the land-sea distribution.

The distribution is shown in detail in Part 1 of this paper sequence - we will revise the manuscript to provide these information and the associated reference.

**Author's changes in the manuscript**: Sect. 2.5, lines 199-201: "Approximately 75 % of cloud tracks occur over ocean, with land-based tracks comprising the remaining 25 % (Figure 2, b). Compared to the land-sea distribution of grid points across the AOI, we observe a 10 % shift toward ocean for detected clouds (Brüning and Tost, 2025)."

- **Figure 4**: Figure 4: It is difficult to read from the figure whether the contribution over land is larger or smaller than that over the sea. Moreover, it is unclear whether the "all" category represents the sum of the land and sea contributions. Could you please clarify what "all" refers to?
  **Author's response**: We revise the figure to show this contribution more clearly. Here, "all" refers to "clouds over all surface types" (i.e, the whole dataset) - we will change the description.
  **Author's changes in the manuscript**: Changed "All" to "All surface types". For the visualization, we changed the plot to show bars for each class (sea, land, all surface types).

- **L352-368**: L352-368: It is unclear whether the "differences" mentioned in line 352 refer to the values of JJA minus those of MAM. The description in this paragraph does not appear to be consistent with Fig. 9. For example, the statement "ROME displays more complex behavior along the period, with an overall increase around the AOI in JJA" (lines 356–358) seems inconsistent with Fig. 9(c), which clearly shows an increase in ROME across the entire domain.
  **Author's response**: The values show MAM minus JJA, we add the information in the caption of Figure 9 and the text. Moreover, we revise the paragraph to describe the observed changes more clearly.
  **Author's changes in the manuscript**: Please see Sect. 4.3, lines 357-377 for a revised description of Figure 9.

- **L352**: L352: The term "summer" is not an appropriate abbreviation to refer to JJA. "Boreal summer" or simply "JJA" is more suitable.
  **Author's response**: We agree and change the term to "boreal summer"
  **Author's changes in the manuscript**: Sect. 4.3, line 357: "Figure 9 presents the average differences between boreal spring (MAM) and boreal summer (JJA) (MAM minus JJA), interpolated on a 3° × 3° grid and along the latitudes."

- **L369**: L369: Please refer to "Table 4" at the beginning of the paragraph.
  **Author's response**: Thank you for your comment, will be changed.
  **Author's changes in the manuscript**: Please see Sect. 4.3, lines 378ff.: "We evaluate how the relationships between organisation indices and cloud/core properties evolve along the two seasonal subsets by comparing the correlation coefficients between MAM and JJA (Table 4). (...)"

- **L378**: L378, "additional regional factors": What factors would affect the results? The large-scale circulation is an example.
  **Author's response**: We will revise the sentence and add examples - such as the large-scale circulation, interannual variability like ENSO, or the topography, which may have an effect but were not directly considered in our analysis.
  **Author's changes in the manuscript**: Sect. 4.3, lines 387-389: "These weak correlations suggest that relations may be affected by additional factors which were not integrated in our analysis, such as the large-scale circulation, interannual variations (caused by, e.g., El Niño-Southern Oscillation (ENSO)), or local topography."

- **L405**: L405, "more common": The contribution over land is smaller than that over the ocean in all cases. The phrase "more common" is unclear and may be misleading.
  **Author's response**: We agree and change the sentence .
  **Author's changes in the manuscript**: Sect. 4.4.1, lines 416-419: "Comparing the surface types of all cloud tracks and both percentile subsets, we observe a higher proportion of clouds over the ocean than over land for all datasets. However, there are differences within the surface-type distribution for the organisation-based subsets: when comparing all three datasets (all cloud tracks, P90, P10), strong convective organisation occurs about 5–15 % more frequently over the ocean, whereas the proportion of cloud systems with a weak convective organisation is about 10–15 % higher over land (Figure 11, a)."

- **L598**: L598: Please spell out "DBSCAN" and "HDBSCAN" and provide appropriate references for both.
  **Author's response**: We will add references and introduce the acronyms.

140 **Author's changes in the manuscript**: Sect. 5.3, lines 609-612: "To address this, we plan to explore unsupervised clustering techniques such as the Density Based Spatial Clustering of Applications with Noise (DBSCAN) (Ester et al., 1996) or the extended Hierarchical Density-Based Spatial Clustering of Applications with Noise (HDBSCAN) (Campello et al., 2013) as a more data-driven alternative."